# The renal lineage factor PAX8 controls oncogenic signalling in kidney cancer

Saroor A. Patel[1,13], Shoko Hirosue[1], Paulo Rodrigues[1], Erika Vojtasova[1], Emma K. Richardson[1,14], Jianfeng Ge[1], Saiful E. Syafruddin[1,2], Alyson Speed[1], Evangelia K. Papachristou[3], David Baker[4], David Clarke[5], Stephenie Purvis[5], Ludovic Wesolowski[1], Anna Dyas[1], Leticia Castillon[1,6], Veronica Caraffini[1], Dóra Bihary[1], Cissy Yong[7,8], David J. Harrison[9], Grant D. Stewart[7], Mitchell J. Machiela[10], Mark P. Purdue[10], Stephen J. Chanock[10], Anne Y. Warren[11], Shamith A. Samarajiwa[1], Jason S. Carroll[3] & Sakari Vanharanta[1,6,12] ✉

Large-scale human genetic data[1–3] have shown that cancer mutations display strong tissue-selectivity, but how this selectivity arises remains unclear. Here, using experimental models, functional genomics and analyses of patient samples, we demonstrate that the lineage transcription factor paired box 8 (PAX8) is required for oncogenic signalling by two common genetic alterations that cause clear cell renal cell carcinoma (ccRCC) in humans: the germline variant rs7948643 at 11q13.3 and somatic inactivation of the von Hippel-Lindau tumour suppressor (*VHL*)[4–6]. *VHL* loss, which is observed in about 90% of ccRCCs, can lead to hypoxia-inducible factor 2α (HIF2A) stabilization[6,7]. We show that HIF2A is preferentially recruited to PAX8-bound transcriptional enhancers, including a pro-tumorigenic cyclin D1 (*CCND1*) enhancer that is controlled by PAX8 and HIF2A. The ccRCC-protective allele C at rs7948643 inhibits PAX8 binding at this enhancer and downstream activation of *CCND1* expression. Co-option of a PAX8-dependent physiological programme that supports the proliferation of normal renal epithelial cells is also required for *MYC* expression from the ccRCC metastasis-associated amplicons at 8q21.3-q24.3 (ref. [8]). These results demonstrate that transcriptional lineage factors are essential for oncogenic signalling and that they mediate tissue-specific cancer risk associated with somatic and inherited genetic variants.

How genetic mutations lead to tissue-specific cancer phenotypes remains a fundamental open question in cancer biology[9]. Somatic mutations in most cancer driver genes are detected only in a minority of tumour types[1,2], and inherited cancer predisposition alleles, both common and rare, are usually associated with cancer risk in a tissue-specific manner[3]. The strong effect of tissue of origin on carcinogenesis suggests that the transcriptional networks that define normal cellular states may also be crucial for oncogenic processes[9]. Lineage transcription factors (TFs), such as SOX10 in melanoma[10,11], are often needed for cancer cell survival and proliferation[12,13]. However, whether specific interactions between lineage factors and genetic alterations are needed for the establishment of cancer-type-specific oncogenic programmes has remained unclear. Loss-of-function changes in *VHL*, which are commonly seen in ccRCC[6], are extremely rare in other cancers[1], and they lead to the constitutive stabilization of HIF1A and HIF2A (also known as EPAS1), of which HIF2A has a dominant role in ccRCC[7]. Capitalizing on the particular genetic make-up of ccRCC, we set out to investigate the effect of transcriptional lineage factors on the oncogenic phenotypes that arise downstream of cancer-associated genetic alterations.

## PAX8 and HIF2A interact on chromatin

To identify essential TFs in ccRCC, we performed pooled loss-of-function screens for TFs that support the proliferation of two metastatic ccRCC cell lines: OS-LM1 and 786-M1A. These cell lines have been extensively characterized at the molecular and phenotypic levels and display clinically relevant genetic and gene regulatory characteristics, including *VHL* mutations[14–16]. The non-ccRCC cell lines MDA-MB-231 and HeLa were also used for comparison. Two factors, PAX8 and HNF1 homeobox B (HNF1B), showed strong specificity for ccRCC cells (Fig. 1a, b), a result supported by an analysis of public genome-wide CRISPR–Cas9 and RNAi screening data[12,13,17–19] (Extended Data Fig. 1a) and by validation

[1]MRC Cancer Unit, University of Cambridge, Hutchison/MRC Research Centre, Cambridge Biomedical Campus, Cambridge, UK. [2]UKM Medical Molecular Biology Institute, Universiti Kebangsaan Malaysia, Jalan Yaacob Latiff, Bandar Tun Razak, Malaysia. [3]Cancer Research UK Cambridge Institute, University of Cambridge, Robinson Way, Cambridge, UK. [4]Quadram Institute Bioscience, Norwich Research Park, Norwich, UK. [5]Cambridge Genomics Laboratory, Cambridge University Hospitals NHS Foundation Trust, Cambridge, UK. [6]Translational Cancer Medicine Program, Faculty of Medicine, Biomedicum Helsinki, University of Helsinki, Helsinki, Finland. [7]Department of Surgery, University of Cambridge, Cambridge Biomedical Campus, Cambridge, UK. [8]Cambridge University Hospitals NHS Foundation Trust, Cambridge, UK. [9]School of Medicine, University of St Andrews, St Andrews, UK. [10]Division of Cancer Epidemiology and Genetics, National Cancer Institute, Rockville, MD, USA. [11]Department of Histopathology, Cambridge University Hospitals NHS Foundation Trust, Cambridge, UK. [12]Department of Physiology, Faculty of Medicine, University of Helsinki, Helsinki, Finland. [13]Present address: Wellcome Sanger Institute, Cambridge, UK. [14]Present address: Division of Medical Oncology, National Cancer Centre Singapore, Singapore, Singapore. ✉e-mail: sakari.vanharanta@helsinki.fi

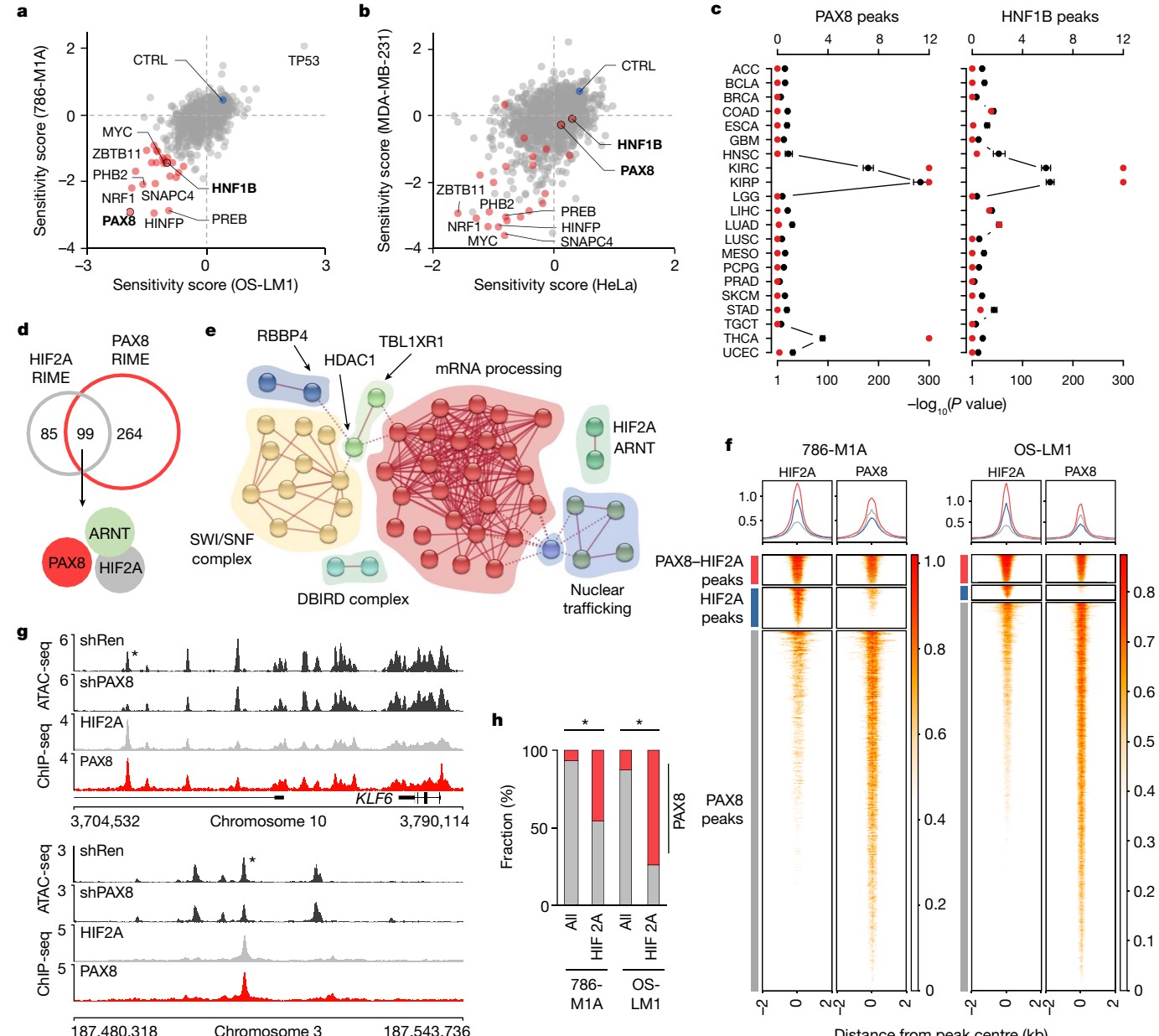

**Fig. 1 | Chromatin level interaction between the renal lineage factor PAX8 and oncogenic HIF2A in ccRCC. a,b,** Pooled CRISPR–Cas9 loss-of-function screen results of ccRCC cell lines (**a**) and non-ccRCC cell lines (**b**). Sensitivity score, $\log_2$ of the mean of the top three depleted sgRNAs per gene, two replicates per condition, at the end of the assay compared with the start of the assay. ccRCC dependencies are in red. CTRL, average of non-targeting controls. **c,** Overlap between cancer-type-specific ATAC-seq peaks in TCGA data and those with reduced accessibility after PAX8 and HNF1B depletion in ccRCC cells. Top axis, odds ratio of overlap (black), 95% confidence interval. Bottom axis, *P* value, one-sided Fisher's exact test (red). **d,** Overlap between PAX8- and HIF2A-interacting proteins as determined by RIME in 786-M1A cells. **e,** Network presentation of physical connections between 89 shared nuclear proteins from HIF2A and PAX8 interactomes. Protein names are provided in Extended Data

Fig. 4a. **f,** Heatmaps of HIF2A and PAX8 ChIP-seq signals from 786-M1A and OS-LM1 xenografts (three tumours each) across regions with strong PAX8–HIF2A co-binding (red), predominant HIF2A binding (blue) and predominant PAX8 binding (grey). Top panels show the average signal within each of the three categories in the same colours. **g,** HIF2A and PAX8 co-bound genomic regions with reduced accessibility following PAX8 depletion. Median ATAC-seq signal from 786-M1A cells expressing a control RNAi construct (shRen, $N = 6$) or PAX8-targeting RNAi constructs (shPAX8, $N = 6$). Median HIF2A and PAX8 ChIP-seq signals from 786-M1A and OS-LM1 xenografts, three tumours each. Asterisk indicates a region of interest. **h,** Fraction of PAX8 peaks (red) in all high-confidence open chromatin regions (all) and HIF2A ChIP-seq peaks in 786-M1A and OS-LM1 xenograft tumours. Asterisk indicates $P < 1.0 \times 10^{-300}$, two-sided Fisher's exact test.

experiments in several ccRCC cell lines in vitro and in vivo (Extended Data Fig. 1b–h). The combined high expression of PAX8 and HNF1B was evident in renal cancers and normal tissues of the renal epithelial lineage at different developmental stages (Extended Data Fig. 2a–h). In line with their role as renal reprogramming factors[20], inhibition of PAX8 and HNF1B reduced, but did not eliminate, chromatin accessibility at genomic loci, especially distal regulatory elements, enriched

for their predicted DNA-binding motifs and characteristic of the renal origin (Fig. 1c and Extended Data Fig. 3a–n).

In parallel, we performed rapid immunoprecipitation mass spectrometry of endogenous proteins (RIME)[21] to characterize nuclear complexes occupied by HIF2A. The protein complexes that precipitated in all four replicates of the HIF2A RIME showed strong representation of HIF2A and ARNT (Fig. 1d and Supplementary Table 1), the HIF2A

dimerization partner essential for DNA binding[22]. By contrast, the control IgG RIME experiments showed no signal for these proteins. In addition, we identified PAX8 as a member of the HIF2A nuclear interactome in three out of the four RIME experiments, but not in IgG controls. Conversely, HNF1B was identified in one out of the four HIF2A RIME replicates and in one of the IgG control experiments, which suggests that this could reflect background binding. A reciprocal experiment using PAX8 antibodies gave a strong signal for PAX8 in all four RIME replicates and identified HIF2A and ARNT as members of its nuclear interactome in three and four replicates, respectively (Fig. 1d and Supplementary Table 2). Of the 183 specific proteins identified in the same complexes with HIF2A, 99 also belonged to the same complexes with PAX8, and 89 of these represented nuclear proteins with functions in processes such as chromatin remodelling (the SWI/SNF complex) or mRNA processing (Fig. 1e and Extended Data Fig. 4a).

Chromatin immunoprecipitation with sequencing (ChIP-seq) analysis of xenografted ccRCC tumours revealed that PAX8 and HIF2A colocalized on chromatin substantially more frequently than what would be expected by chance. Specifically, 43% and 65% of the HIF2A binding sites in 786-M1A and OS-LM1 tumours, respectively, showed significant PAX8 binding (Fig. 1f–h and Extended Data Fig. 4b, c). PAX8 motifs were enriched in open chromatin regions that characterize both ccRCCs and papillary RCCs (Extended Data Fig. 3m, n), but *VHL* mutations are specific to ccRCC. In line with this, the HIF2A motif was the most significantly enriched motif in ccRCC-specific peaks from assay of transposase accessible chromatin sequencing (ATAC-seq) when compared to papillary RCCs in the The Cancer Genome Atlas (TCGA) cohort[23] (Extended Data Fig. 4d, e). The orientation of PAX8-binding and HIF2A-binding motifs in ccRCC-specific genomic regions varied, and the distance was more than expected for co-operative DNA binding[24] (Extended Data Fig. 4f). The only recurrent HIF2A–PAX8 motif orientation was related to the long terminal repeat sequence of a common endogenous retrovirus, ERV1, the expression of which has been linked to poor patient outcomes in ccRCC[25,26]. Also, although we observed HIF2A interactions with ARNT, we did not detect HIF2A–PAX8 interactions by co-immunoprecipitation (Extended Data Fig. 4g). These results demonstrate that in ccRCC, the renal lineage factor PAX8 and the oncogenic driver HIF2A interact at the chromatin level, probably through DNA and shared chromatin factor complexes.

## PAX8–HIF2A-dependent oncogene activation

In line with the strong effect on proliferation, PAX8 and HNF1B depletion in 786-M1A and OS-LM1 cells led to reduced expression of genes involved in the cell cycle, targets of E2F1 and MYC signalling (Fig. 2a, Extended Data Fig. 4h and Supplementary Tables 3 and 4). The expression of PAX8-dependent and HNF1B-dependent genes also tracked with *PAX8* and *HNF1B* expression, respectively, in fetal human kidney (Extended Data Figs. 2g and 4i, j). Notably, the hypoxia gene set was significantly downregulated in PAX8-depleted, but not in HNF1B-depleted, ccRCC cells (Fig. 2a). Restoration of VHL and consequent inhibition of HIF2A does not have a strong effect on the proliferation of ccRCC cells in vitro[7]. To identify gene regulatory elements that mediate HIF2A-driven ccRCC formation, we set out to identify transcriptional targets of HIF2A in vivo, map HIF2A-bound regulatory elements in the vicinity of these genes and target these enhancers using a CRISPRi-based loss-of-function screen in vivo.

We first developed a tumour model in which HIF2A expression could be experimentally controlled by deriving a HIF2A knockout clone from 786-M1A cells (referred to as C-M1A$^{HIF2A-/-}$) and reintroduced HIF2A in these cells using a doxycycline-dependent transgene (Extended Data Fig. 5a). C-M1A$^{HIF2A-/-}$ cells grew independently of HIF2A in vitro (Extended Data Fig. 5b), but they required the DNA-binding domain of HIF2A for tumour formation in vivo (Extended Data Fig. 5c). HIF2A depletion following doxycycline withdrawal from the diet (Extended Data Fig. 5d, e) enabled us to characterize the transcriptomic effects of HIF2A inhibition at different time points by RNA sequencing (RNA-seq) in vivo. Focusing on genes with early and sustained downregulation, we detected 205 strongly HIF2A-dependent transcripts (Extended Data Fig. 5f and Supplementary Table 5). A total of 175 HIF2A ChIP-seq peaks within a 500-kb region flanking the transcription start site of these genes were also identified, which was a significant enrichment over background (empirical $P < 0.001$ based on 1,000 permutations). We generated a pooled library of 706 single guide RNA (sgRNA) pairs that targeted these peaks and 30 positive and negative controls (Supplementary Table 6) using a tandem design previously shown to effectively inhibit enhancer function[16,27]. The library was transduced together with dCas9–KRAB[28] into a clone of 786-M1A cells that was sensitive to VHL restoration in a tumour-formation assay in vivo. We then transplanted these cells into 15 NSG mice, two tumours each, and measured sgRNA representation in established tumours (Extended Data Fig. 5g). Permutation-based tests showed that the 698 constructs with high representation in the plasmid library contained sgRNAs that were consistently depleted in tumours (Extended Data Fig. 5h). All well-represented non-targeting control constructs were recovered with high efficiency from tumours, whereas constructs targeting essential genes were frequently lost (Extended Data Fig. 5i). In addition, several constructs that targeted HIF2A-bound enhancers were depleted in tumours (Extended Data Fig. 5i). Combining data from constructs with shared target regions, we observed significant (empirical $P < 0.01$ using 10,000 permutations) depletion of constructs that targeted 21 HIF2A-bound enhancers (Fig. 2b), 16 of which showed binding of both HIF2A and PAX8 (Fig. 2c and Extended Data Fig. 5j). The strongest hit was an intergenic region at chromosome 11:69,419,632-69,420,080 (Fig. 2c). This enhancer, referred henceforth to as E11:69419, overlapped with one of the most strongly ccRCC-specific open chromatin regions in a large clinical ATAC-seq cancer dataset[23] (Fig. 2d), with clear activation in ccRCCs but not in papillary RCCs (Fig. 2e). Compared to renal cancer samples, the region covering E11:69419 showed low accessibility in other tissues, including samples representing normal renal lineage, in a large human DNAse I hypersensitivity catalogue[29] (Fig. 2f). We validated the in vivo role of E11:69419 in ccRCC formation by inhibiting it using CRISPRi with two independent sgRNA pairs (Fig. 2g).

E11:69419 is flanked by two protein coding genes, *MYEOV* and *CCND1*, and it harboured strong HIF2A and PAX8 peaks (Fig. 2c). It also overlapped with the set of genomic loci that showed reduced accessibility after PAX8 depletion in our data, and its activity has been previously linked to HIF2A[30,31]. *CCND1* encodes cyclin D1, a positive cell cycle regulator that is activated in several cancer types[32], including ccRCC, in which its expression is controlled by the VHL–HIF2A pathway[33]. *MYEOV* is poorly characterized and has only weak homology to other known proteins. On the basis of chromatin interaction data, E11:69419 interacts with the promoter regions of *MYEOV* and *CCND1* (ref. [34]). CRISPRi-mediated inhibition of E11:69419 led to downregulation of both of these genes—as determined by quantitative PCR with reverse transcription (RT–qPCR) (Fig. 2h)—but only *CCND1* was required for ccRCC cell proliferation and tumour formation in vivo (Fig. 2i and Extended Data Fig. 6a, b). Inhibition of PAX8 and HIF2A, but not HNF1B or HIF1A, reduced *CCND1* expression (Extended Data Fig. 6c–k and Supplementary Table 4). Notably, combined inhibition of PAX8 and HIF2A did not further reduce *CCND1* levels (Fig. 2j and Extended Data Fig. 6l). We did not find consistent evidence of HIF2A affecting *PAX8* expression or vice versa (Extended Data Fig. 6m, n), but *CCND1* expression correlated more strongly with *HIF2A* than *PAX8* expression in clinical ccRCC specimens (Extended Data Fig. 6o).

## PAX8 mediates inherited ccRCC risk

Genome-wide association studies (GWAS) have identified common genetic variants that are associated with RCC risk in humans, the most significant of which is rs7105934 on chromosome 11q13.3 (refs. [4,5]). This

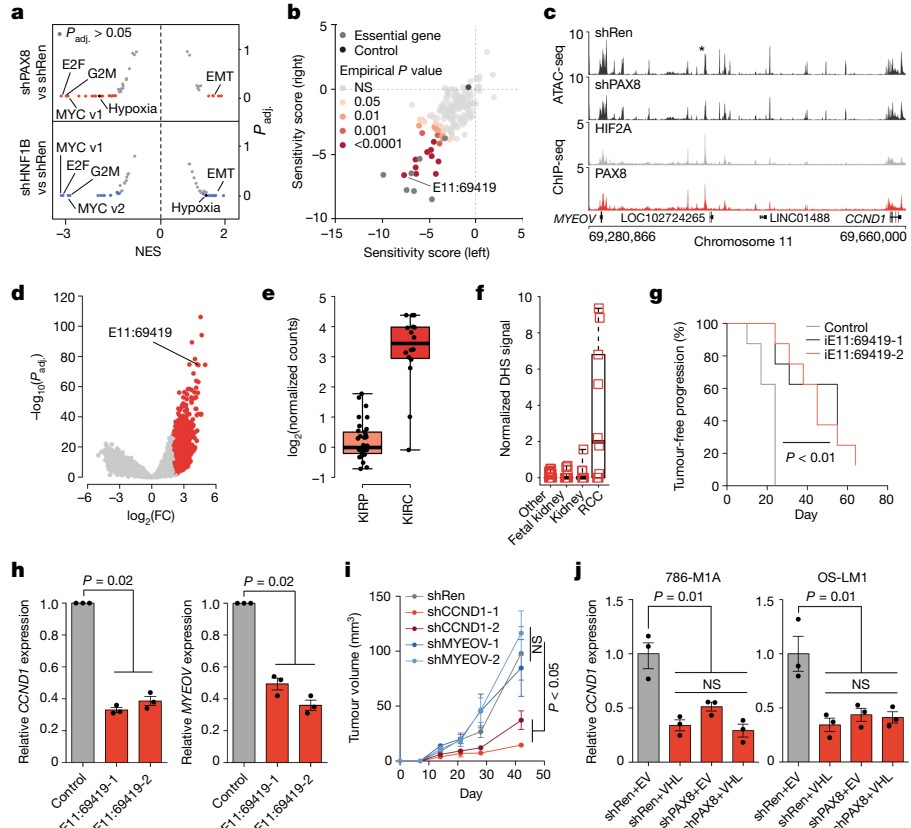

**Fig. 2 | PAX8–HIF2A interactions support oncogene activation in ccRCC.**
**a**, Gene set enrichment analysis with MSigDB Hallmark gene sets on the effects of PAX8 and HNF1B depletion compared with a control RNAi construct (shRen). Two PAX8-targeting (shPAX8) and HNF1B-targeting (shHNF1B) RNAi constructs and cell lines (786-M1A and OS-LM1) were combined for each gene, respectively. Significantly changed gene sets are in colour (blue or red). EMT, epithelial-to-mesenchymal transition; NES, normalized enrichment score. **b**, Pooled in vivo CRISPRi screening. Normalized average depletion for the two most depleted constructs per region presented for 30 tumours in two groups (left versus right mouse flank). Essential genes, positive control genes. Control, average of non-targeting constructs. Empirical one-sided $P$ values based on 10,000 permutations. **c**, Median ATAC-seq signals from shRen control ($N = 6$) and shPAX8 ($N = 6$) cells. Median HIF2A and PAX8 ChIP-seq signals from 786-M1A and OS-LM1 xenografts, three tumours each. Asterisk indicates E11:69419. **d**, Differential DNA accessibility. TCGA ATAC-seq data, 410 human

tumours, 562,709 pan-cancer peaks. ccRCCs compared to all other tumour types by DESeq2. **e**, Normalized DNA accessibility at E11:69419, TCGA ATAC-seq data. ccRCC (KIRC), $N = 16$; papillary RCC (KIRP), $N = 34$. **f**, Normalized DNAse hypersensitivity (DHS) signal for E11:69419, 733 samples from different cell types. **g**, Tumour-free survival of mice inoculated with 786-M1A cells. iE11:69419, E11:69419 targeted by CRISPRi. log-rank test. $N = 8$ tumours for each group. **h**, RT–qPCR results of E11:69419 targeted by CRISPRi in 786-M1A cells. **i**, Subcutaneous tumour growth, 786-M1A cells. shRen (control RNAi construct), shCCND1-1 and shCCND1-2 (two RNAi constructs that target CCND1), $N = 8$; shMYEOV-1 and shMYEOV-2 (two RNAi constructs that target MYEOV), $N = 10$ tumours per group. Mean and s.e.m. Two-sided Kruskal–Wallis test. **j**, RT–qPCR results. EV, empty vector. For **e** and **f**, box plots show the median and interquartile range, and whiskers show the data range. For **h** and **j**, data points indicate independent RNA preparations ($N = 3$). Mean and s.e.m. Two-sided Kruskal–Wallis test.

risk haplotype comprises E11:69419, which covers the linked single nucleotide polymorphisms (SNPs) rs7948643 and rs7939721 (ref. [30]). Motif analysis identified two putative binding sites for both HIF2A and PAX8, but not HNF1B, in the E11:69419 sequence (Fig. 3a). As OS-LM1 and 786-M1A cells carry a luciferase transgene, we used 786-O and 2801-LM1 (a metastatic derivative of 786-O) cells in luciferase-based reporter assays, which showed robust enhancer activity for the E11:69419 sequence (Fig. 3b). Mutating one HIF2A and one PAX8 site reduced E11:69419 activity, whereas the other mutations did not have an effect (Fig. 3c). In line with the CCND1 expression data (Fig. 2j), combining mutations that inactivated the functional PAX8 and HIF2A sites did not further reduce reporter activity (Fig. 3c). Pharmacological HIF2A inhibition also reduced E11:69419 activity (Extended Data Fig. 7a). rs7948643 is located exactly at the functionally important PAX8 binding site within E11:69419, in which the more common risk allele T is the nucleotide with the highest information content in the motif derived from our PAX8-depleted ATAC-seq peaks (Fig. 3d). By contrast, the rarer protective allele C, with a ccRCC odds ratio of 0.7 (ref. [5]), was predicted to reduce PAX8 affinity for the motif (Fig. 3d). In reporter assays, changing

the allele T at rs7948643 for the minor allele C resulted in a reduction in enhancer activity that was comparable to that observed with larger mutations in the predicted PAX8-binding site (Fig. 3c, e).

Gel-shift assays demonstrated that PAX8 bound the predicted PAX8 motif within E11:69419, with the allele T at rs7948643 showing higher affinity than allele C (Fig. 3f, g and Extended Data Fig. 7b). ATAC-seq analysis of a heterozygous VHL mutant ccRCC cell line, RCC-JF, suggested that there was equal chromatin accessibility of both the risk and protective E11:69419 alleles, a result supported by an analysis of heterozygous human ccRCC specimens (Extended Data Fig. 7c). In line with this, expression of a VHL-insensitive constitutively stable form of HIF2A in renal epithelial HK2 cells that endogenously express PAX8 led to E11:69419 activation in reporter assays but no accessibility at the endogenous E11:69419 locus or increase in CCND1 expression even after about 40 population doublings over 6 weeks (Extended Data Fig. 7d–g). Long-read whole-genome sequencing of RCC-JF cells resolved the haplotypes of the CCND1 locus, linking the risk allele T at rs7948643 to the allele A at rs7177 in the CCND1 3′ untranslated region (Fig. 3h and Extended Data Fig. 7h). Allele-specific ChIP–qPCR of RCC-JF cells

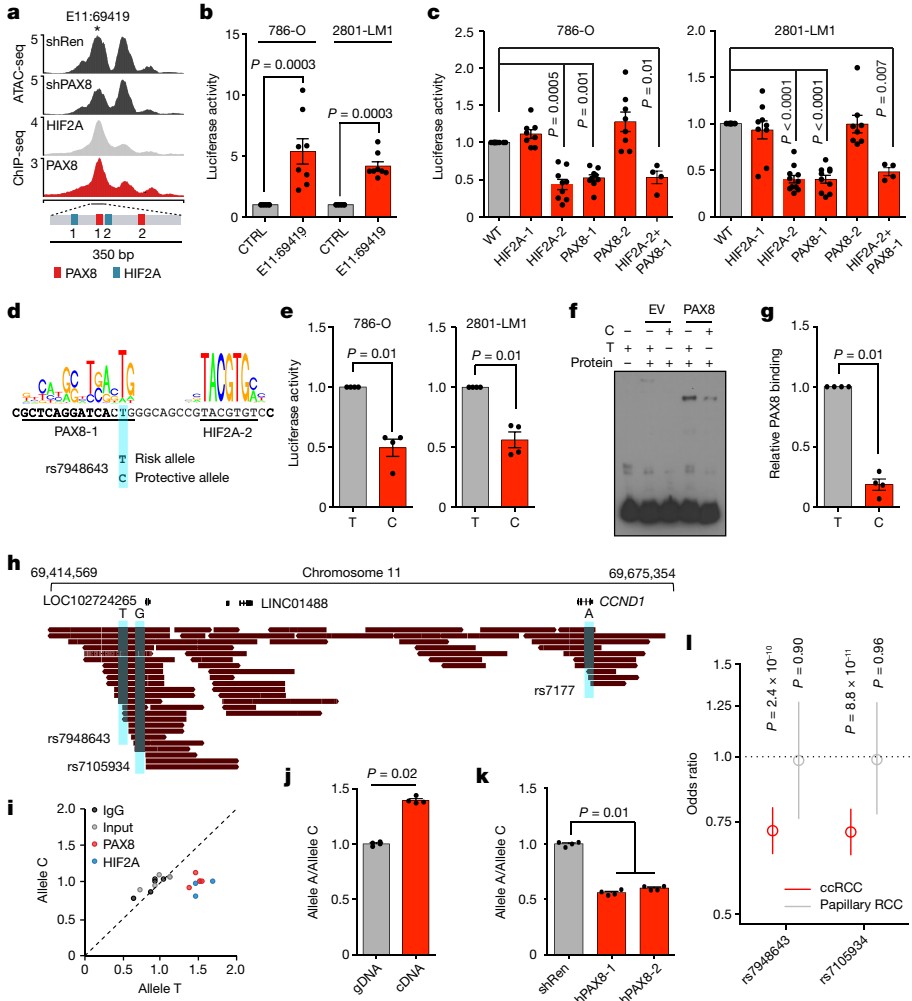

**Fig. 3 | The ccRCC risk allele at rs7948643 increases PAX8-dependent activation of an oncogenic enhancer. a**, Chromosome 11:69,417,866-69,422,866, median shRen ($N = 6$) and shPAX8 ($N = 6$) ATAC-seq signals, and median 786-M1A and OS-LM1 xenograft ($N = 3$ tumours each) HIF2A and PAX8 ChIP-seq signals. Asterisk indicates E11:69419, with the relative orientation of HIF2A and PAX8 DNA-binding motifs highlighted. **b**, Reporter assay showing E11:69419 enhancer activity, fold change over control, arbitrary units. $N = 8$. **c**, Reporter assay showing the effect of mutated HIF2A and PAX8 sites on E11:69419 activity. 786-O: wild type (WT), $N = 10$; HIF2A-1, $N = 8$; HIF2A-2, $N = 8$; PAX8-1, $N = 9$; PAX8-2, $N = 8$; HIF2A-2+PAX8-1, $N = 4$. 2801-LM1: WT, $N = 12$; HIF2A-1, $N = 9$; HIF2A-2, $N = 11$; PAX8-1, $N = 10$; PAX8-2, $N = 8$; HIF2A-2+PAX8-1, $N = 4$. **d**, PAX8- and HIF2A-binding motifs at E11:69419, with the ccRCC risk-associated SNP rs7948643 highlighted. **e**, Reporter assay showing the effect of the T>C change at rs7948643 on E11:69419 activity. $N = 4$. **f,g**, Electrophoretic mobility shift assay. Representative image (**f**) and quantification (**g**) of independent experiments ($N = 4$). Protein from

MDA-MB-231 cells expressing EV or PAX8, oligonucleotides with the T or C allele at rs7948643. **h**, Long DNA reads used for phasing of the 11q13.3 RCC risk allele in RCC-JF cells. **i**, Allele-specific HIF2A, PAX8 or IgG ChIP qPCR in RCC-JF cells at rs7948643, normalized to allele ratio of input control. Data points indicate independent immunoprecipitation reactions. HIF2A, $N = 3$; other conditions $N = 5$. **j**, Allele-specific RT–qPCR results of rs7177 in RCC-JF cells, normalized to the allele ratio in genomic DNA (gDNA). **k**, Allele-specific RT–qPCR results of rs7177 in RCC-JF cells after PAX8 depletion, normalized to shRen control. **l**, Subtype-specific RCC risk associated with rs7948643 and rs7105934. Minor allele frequency of 0.07 for both variants. ccRCC, 5,648 cases and 15,010 controls; papillary RCC, 563 cases and 14,840 controls. Odds ratio shown, with whiskers representing 95% confidence intervals. For **b**, **c** and **e**, data points indicate the average of three technical replicates, independent transfections. For **j** and **k**, data points indicate independent RNA preparations ($N = 4$). For **b**, **c**, **e**, **g**, **j** and **k**, mean and s.e.m. shown. Two-sided Kruskal–Wallis test.

confirmed there was higher binding of PAX8 and HIF2A to the allele T at rs7948643 when compared to allele C in the chromatin context (Fig. 3i). In line with this, PAX8 depletion reduced HIF2A binding at E11:69419 but not at a PAX8-independent HIF2A bound enhancer (E14:34035), and HIF2A depletion did not affect PAX8 binding (Extended Data Fig. 7i, j). Allele-specific RT–qPCR demonstrated higher baseline expression and a strong bias towards reduced expression of allele A at rs7177 following PAX8 depletion when compared to allele C (Fig. 3j, k and Extended Data Fig. 7k). We did not detect PAX8 binding at E11:69419 in papillary RCC cell lines (Extended Data Fig. 7l). A RCC-subtype-specific meta-analysis of human GWAS data[5] revealed that rs7948643 was associated with ccRCC ($P = 2.4 \times 10^{-10}$) but not papillary RCC ($P = 0.90$) (Fig. 3l, Extended

Data Fig. 8, Extended Data Table 1 and Supplementary Tables 7 and 8). Accessible E11:69419 therefore integrates the PAX8 and HIF2A signals, both of which are needed for full E11:69419 activity. Moreover, the ccRCC-protective allele C at rs7948643 inhibits PAX8 binding, which consequently reduces the activity of E11:69419 upstream of the oncogenic driver *CCND1* and possible other pro-tumorigenic mediators.

## A physiological *MYC* programme in cancer

In contrast to HIF2A inactivation (Extended Data Fig. 5b), PAX8 inhibition compromised ccRCC proliferation in vitro (Extended Data Fig. 1b), which indicated the presence of HIF2A-independent oncogenic PAX8

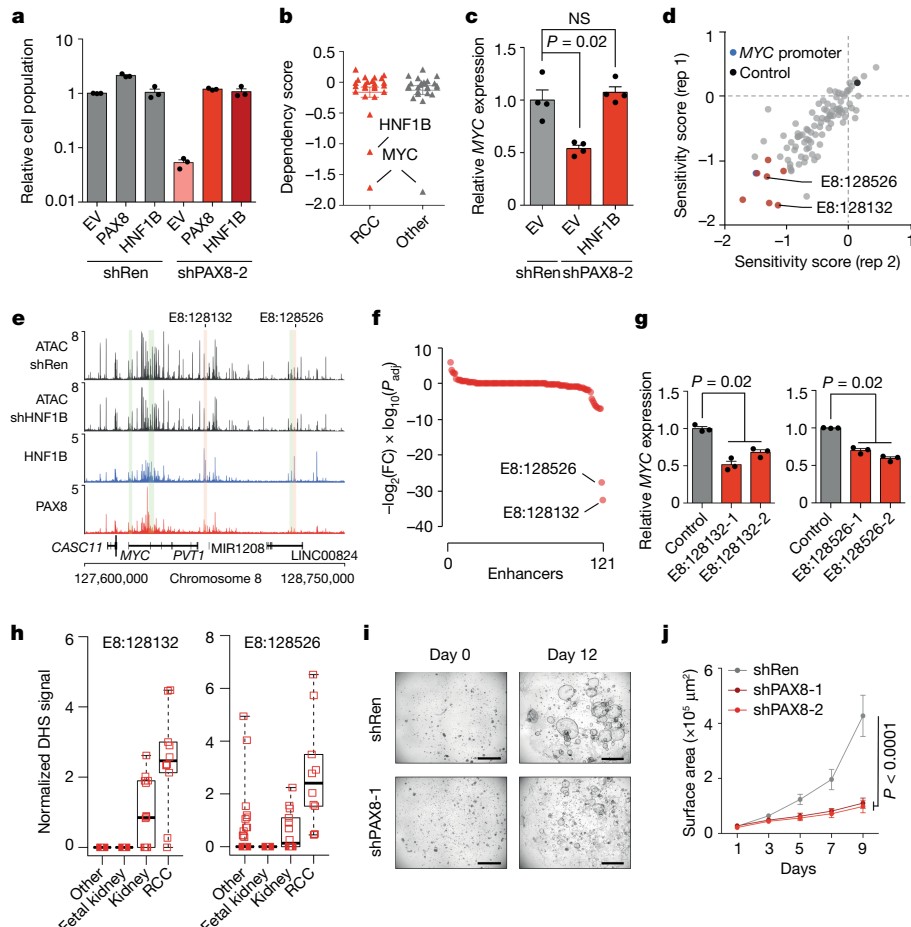

**Fig. 4 | Co-option of a normal lineage factor programme for oncogenesis in ccRCC-associated 8q21.3-q24.3 amplifications. a**, Competitive proliferation assay against EV-shRen control cells. Relative proportion of the indicated cells on day 12 compared with day 0. 786-M1A cells. Data points indicate technical replicates ($N = 3$). Mean and s.e.m. **b**, Average dependency score (CERES score) in the DepMap dataset for 25 shared genes downregulated by PAX8 and HNF1B inhibition. **c**, RT–qPCR results of 786-M1A cells. Data points indicate independent RNA preparations ($N = 4$). Mean and s.e.m. Two-sided Kruskal–Wallis test. **d**, Pooled CRISPRi-based proliferative screen for putative *MYC* enhancers in 786-M1A cells. Normalized average depletion for the two most depleted constructs per region presented for the two technical replicates (rep 1 and 2). Control, average of non-targeting control constructs. **e**, Median ATAC-seq signals in 786-M1A shRen ($N = 5$) and shHNF1B ($N = 6$) cells, and PAX8 and HNF1B ChIP-seq signals in 786-M1A and OS-LM1 xenografts (3 tumours

each) for the *MYC* locus. Enhancers that support ccRCC proliferation are highlighted (enhancers containing a HNF1B motif in red, others in green). **f**, Effect of HNF1B depletion on chromatin accessibility at the enhancers in the *MYC* locus. Fold changes and adjusted two-sided *P* values derived by DESeq2. **g**, RT–qPCR results following CRISPRi-mediated targeting of E8:128132 and E8:128526 in 786-M1A cells. Data points indicate independent RNA preparations ($N = 3$). Mean and s.e.m. Two-sided Kruskal–Wallis test. **h**, Normalized DHS signal for E8:128132 and E8:128526, 733 samples from different cell types. Box plots show the median and interquartile range, and whiskers the data range. **i,j**, Growth of normal human renal epithelial organoids with and without PAX8 depletion. Representative images (**i**) and quantification (**j**). Scale bar, 1 mm. $N = 21$ random growing organoids per condition and time point. Mean and s.e.m. Two-sided Kruskal–Wallis test.

functions. PAX8 positively regulated *HNF1B* expression, whereas *PAX8* expression was not altered in HNF1B-depleted cells (Extended Data Figs. 4h and 9a–e and Supplementary Tables 3 and 4), and *HNF1B* expression followed *PAX8* expression in the developing kidney (Extended Data Fig. 4j). Reintroduction of exogenous PAX8 or HNF1B rescued the in vitro proliferation defect caused by PAX8 depletion (Fig. 4a and Extended Data Fig. 9f, g). The effect of HNF1B depletion was also rescued by exogenous HNF1B expression (Extended Data Fig. 9h). We identified genes that were downregulated in both PAX8-depleted and HNF1B-depleted cells and that were important for ccRCC proliferation based on genome-wide CRISPR–Cas9 screening data[17,18]. Only two genes fit these criteria: *HNF1B* and *MYC* (Fig. 4b). Of note, *CCND1* was not on this list as its expression does not depend on HNF1B. The corresponding analysis of genes that were upregulated following PAX8 and HNF1B depletion did not reveal any genes that inhibited ccRCC proliferation (Extended Data Fig. 10a). We confirmed that MYC was downregulated in PAX8-depleted and HNF1B-depleted cells at the mRNA and protein

level (Extended Data Fig. 10b–e). Furthermore, knockdown of *MYC* expression to the level observed in HNF1B-depleted cells closely phenocopied the effect of HNF1B inhibition on ccRCC proliferation in vitro (Extended Data Fig. 10f, g), and HNF1B restoration in PAX8-depleted cells restored *MYC* expression (Fig. 4c). Even though HIF2A has been linked to enhanced *MYC* activity[35–38], the negative effect on *MYC* expression could explain the antiproliferative phenotype that follows PAX8 and HNF1B inhibition in vitro, and this effect may be independent of the VHL–HIF2A pathway.

An increased copy number of *MYC* and its regulatory regions is associated with ccRCC metastasis[8], and fluorescence in situ hybridization (FISH) analysis showed that cells of the metastatic 786-M1A cell line carry six copies of *MYC* (Extended Data Fig. 11a). Using a functional CRISPRi screen, we identified eight distal *MYC* enhancers that were important for ccRCC proliferation (Fig. 4d,e and Supplementary Table 9). Two of them, E8:128132 (chromosome 8:128,132,902-128,133,724) and E8:128526 (chromosome 8:128,526,339-128,526,710), showed HNF1B

binding and contained the HNF1B motif (Fig. 4e). Unlike the majority of accessible chromatin regions in the *MYC* locus, both these regions showed significantly reduced accessibility in cells in which HNF1B and PAX8 were knocked down (Fig. 4f and Extended Data Fig. 11b–d) and targeting them with CRISPRi reduced *MYC* expression (Fig. 4g). In addition to renal cancer, E8:128132 and E8:128526 showed DNA accessibility in normal cells derived from the kidney (Fig. 4h), and PAX8 depletion inhibited the proliferation of HK2 cells, a renal epithelial cell line, and normal human renal organoids (Fig. 4i,j and Extended Data Fig. 11e). PAX8 depletion also inhibited *HNF1B* and *MYC* expression in HK2 cells and renal organoids (Extended Data Fig. 11f–h), but did not reduce *CCND1* expression (Extended Data Fig. 11i). Furthermore, CRISPRi-based targeting of E8:128132 and E8:128526 reduced *MYC* expression in HK2 cells (Extended Data Fig. 11j). The cancer-specific 8q21.3-q24.3 amplifications in ccRCC cells therefore co-opt a lineage-factor-dependent physiological programme that supports *MYC* expression and proliferation that is already present in normal renal epithelial cells.

## Discussion

Tissue-specific factors are major determinants of carcinogenesis, but how they contribute to oncogenic processes remains largely unknown[9]. We identified the renal lineage factor PAX8 as a requirement for oncogenic signalling by three major genetic drivers of ccRCC, thereby providing support to the hypothesis that transcriptional lineage factors contribute to the tissue-specific manifestation of oncogenic phenotypes downstream of cancer driver mutations (Extended Data Fig. 11k). Chromatin accessibility data from human specimens together with our functional data indicate that in addition to PAX8, other factors are needed for the establishment of accessibility at crucial oncogenic PAX8-dependent enhancers. Moreover, even though *VHL* mutations are in general only associated with ccRCC in the context of common sporadic cancers, the tumour spectrum of *VHL* germline mutations is broader[7]. As-yet to be identified lineage factor programmes in other tissues may also collaborate with *VHL* loss-induced signals in tumorigenesis. Overall these observations suggest that the interaction between lineage factors and cancer-associated genetic alterations in oncogenesis depends on several layers of epigenetic conditioning.

Multiple known risk loci predispose to renal cancer[5,30,36,39]. We showed that rs7948643, a common genetic variant linked to the most significant renal cancer risk locus rs7105934 on chromosome 11q13.3, falls under a PAX8-binding site within E11:69419, and that the ccRCC risk allele T favours PAX8 binding. The requirement of both PAX8 and HIF2A for E11:69419 activity and the strong association of rs7948643 with ccRCC, but not papillary RCC, support a model in which the difference in PAX8 binding at rs7948643 is the cause of increased ccRCC risk associated with this locus. In line with the tissue and context-specific expression patterns of PAX8 and HIF2A, respectively, and the restricted accessibility of E11:69419, the rs7948643 genotype does not correlate with *CCND1* expression in most normal tissues[40].

Our results provide functional insight into the mechanisms that govern the interaction between inherited and somatic genetic alterations with developmental lineage factors in determining cancer risk, specifically in ccRCC. The distal enhancer E11:69419 integrates signals from the most commonly mutated ccRCC pathway and the most significant common ccRCC risk locus in a PAX8-dependent manner. The molecular mechanism uncovered here is therefore likely to have a significant effect on the population-level cancer burden in the kidney. Moreover, PAX8 supports the expression of two canonical oncogenes, *CCND1* and *MYC*, and genetic inactivation of *Pax8* is tolerated in the mouse kidney[41]. This suggests that PAX8 could be a viable therapeutic target in ccRCC. Strategies to inhibit lineage factors beyond nuclear hormone receptors should be of interest across different cancer types.

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

## Reporting summary

Further information on research design is available in the Nature Research Reporting Summary linked to this paper.

## Data availability

All ATAC-seq, RNA-seq and ChIP-seq data generated within this project have been uploaded into the Gene Expression Omnibus under the access code GSE163001 with the subseries GSE162948, GSE163000, GSE163485 and GSE163487. The mass spectrometry proteomics data have been deposited to the ProteomeXchange Consortium through the PRIDE[42] partner repository with the dataset identifier PXD029522. Human RNA-seq data for different tissue types were downloaded from TCGA data portal (https://tcga-data.nci.nih.gov/) and from the GTex portal (https://gtexportal.org/). TCGA ATAC-seq normalized count data were downloaded from https://gdc.cancer.gov/about-data/publications/ATACseq-AWG. Normalized DNA accessibility signal data were downloaded from https://zenodo.org/record/3838751#.YJgMJC2ZM0o. Molecular signature data were downloaded from the Molecular Signature Database (MSigDB v.7.1.1) (http://www.gsea-msigdb.org/gsea/msigdb/). Protein interaction data were obtained from the STRING database v.11.0 (https://string-db.org). CRISPR–Cas9 screen CERES scores were downloaded from https://portals.broadinstitute.org/achilles. RCC GWAS meta-analysis summary data were provided by M.P.P. (purduem@mail.nih.gov) and S.J.C. (chanocks@mail.nih.gov) and they are available in Supplementary Tables 7 and 8. Data from the original GWAS studies that comprise the meta-analysis data set are available either from the dbGaP (NCI-1, accession number phs000351.v1.p1; NCI-2, phs001736.v1.p1; and IARC-2, phs001271.v1.p1) or from the investigators upon reasonable request (IARC-1, P. Brennan (brennanp@iarc.fr); MDA, J. Gu (jiangu@mdanderson.org)). Other data that support the findings of this study are available from the corresponding author upon reasonable request. Source data are provided with this paper.

## Code availability

Custom computer code used in this study is available at https://doi.org/10.5281/zenodo.6335339.

42. Perez-Riverol, Y. et al. The PRIDE database and related tools and resources in 2019: improving support for quantification data. *Nucleic Acids Res.* **47**, D442–D450 (2019).

**Acknowledgements** We thank R. Schulte, C. Cossetti and G. Grondys-Kotarba from the Cambridge Institute for Medical Research Flow Cytometry Core Facility for assistance with cell sorting; P. Coupland and staff at the CRUK CI Genomics and Bioinformatics Core for sequencing and analysis; A. Obenauf, C. David and C. Frezza for critical reading of the manuscript; and M. Linehan for the UOK101 cells. FISH was performed at the Cambridge Genomics Laboratory, Cambridge University Hospitals NHS Foundation Trust, Cambridge, UK. G.D.S. is supported by the Mark Foundation for Cancer Research. G.D.S. and A.Y.W. are supported by the Cancer Research UK Cambridge Centre (C9685/A25177). J.S.C. acknowledges support from the University of Cambridge, Cancer Research UK core funding (grants A20411, A31344, A29580 and DRCPGM\100088) and Hutchison Whampoa. G.D.S., A.Y.W., J.S.C. and the Human Research Tissue Bank were supported by the NIHR Cambridge Biomedical Research Centre (BRC-1215-20014). The views expressed are those of the author(s) and not necessarily those of the NIHR or the Department of Health and Social Care. The GWAS analysis performed by M.J.M., M.P.P. and S.J.C. was supported by the Intramural Research Program of the National Cancer Institute, part of the National Institutes of Health. The Renal Cancer Research Fund supported the tissue microarray studies (D.J.H.). S.H. received a PhD studentship from the Rosetrees Trust. This project has received funding from the European Union's Horizon 2020 research and innovation programme under the Marie Skłodowska–Curie grant agreement No 955951. This work was supported by the Medical Research Council (MC_UU_12022/7 and MC_UU_12022/10) and Kidney Research UK (RP_033_20170303).

**Author contributions** S.A.P. designed and performed experiments, analysed data and wrote the manuscript. S.H. performed computational analyses. P.R., E.V. and E.K.R. designed and performed experiments and analysed data. S.E.S., A.S., L.W., J.G., A.D. and V.C. performed experiments. E.K.P. helped with the RIME assay and performed proteomics and data analyses. D. Bihary assisted with the computational analyses. D. Baker performed long-read sequencing. L.C. performed the haplotype analyses. D.C. and S.P. performed the FISH analysis. D.J.H., G.D.S. and C.Y. provided the tissue microarray and/or human samples. M.J.M., M.P.P. and S.J.C. provided and analysed the GWAS data. A.Y.W. supervised the immunohistochemistry experiments and analysed the data. S.A.S. supervised computational analyses. J.S.C. supervised the RIME experiments and data analysis. S.V. supervised the project, designed and performed experiments, analysed data and wrote the manuscript.

**Competing interests** The authors declare no competing interests.

**Additional information**
**Correspondence and requests for materials** should be addressed to Sakari Vanharanta.

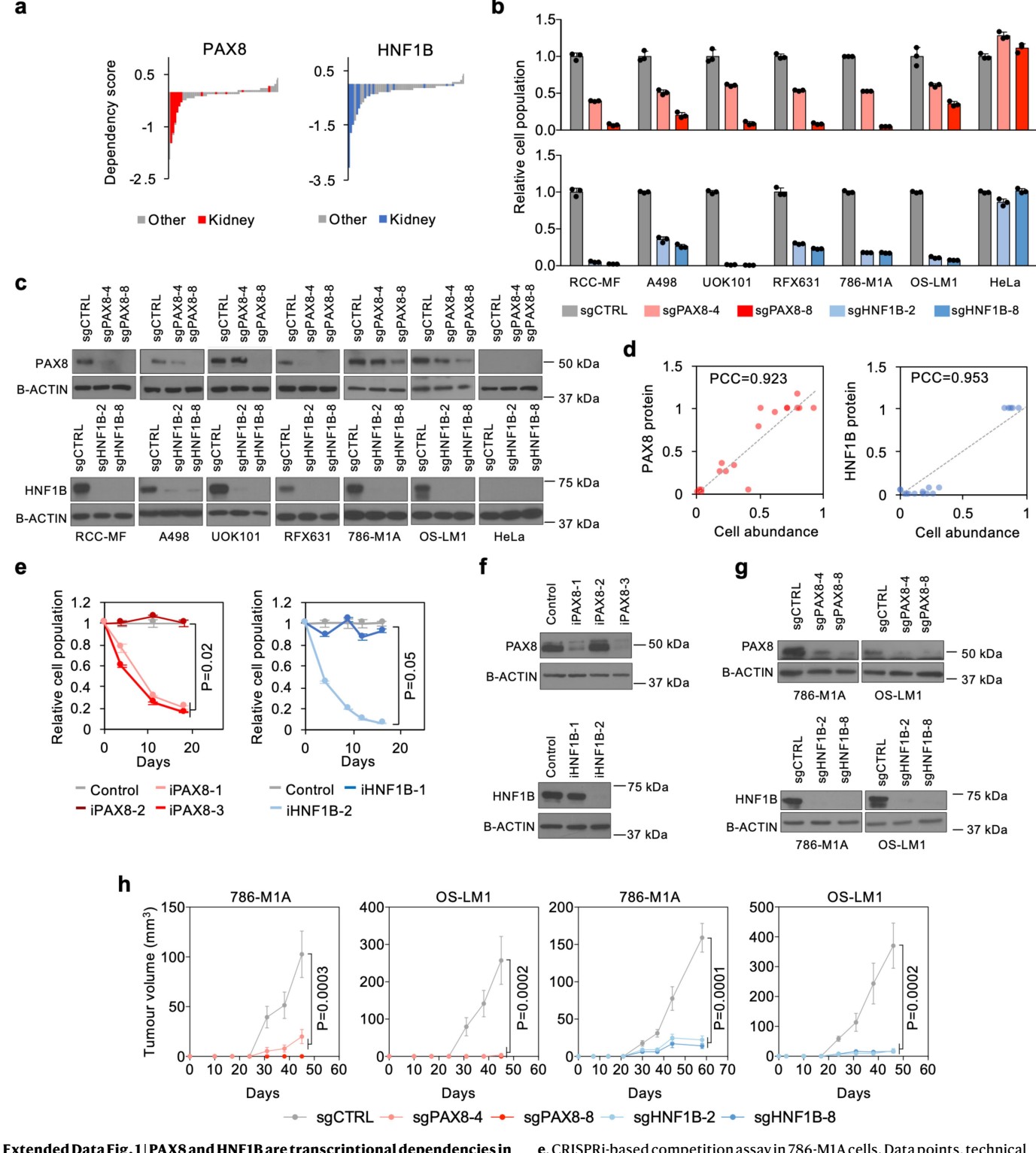

**Extended Data Fig. 1 | PAX8 and HNF1B are transcriptional dependencies in ccRCC. a**. PAX8 and HNF1B dependency (CERES score) across 788 cell lines in the DepMap data set. **b**. CRISPR-Cas9-based competitive proliferation assay against non-targeting sgRNA control cells in different cell lines. Data points, technical replicates (N = 3). Mean and standard deviation. **c**. Western blot showing PAX8 and HNF1B expression in the cells used for competition assays in panel (b). **d**. Correlation between protein expression in panel (c) and relative cell abundance in panel (b). PCC, Pearson's correlation coefficient.

**e**. CRISPRi-based competition assay in 786-M1A cells. Data points, technical replicates (N = 3). Mean and standard deviation. Two-sided Kruskal-Wallis. **f**. Western blot showing PAX8 and HNF1B expression in the cells used for competition assays in panel (e). **g**. Western blot showing PAX8 and HNF1B expression in the cells used for in vivo tumour assays in panel (h). **h**. Subcutaneous tumour growth in athymic mice. PAX8 KO: sgCTRL, N = 8; sgPAX8-4 and sgPAX8-8, N = 6; HNF1B KO: N = 10 tumours for each group. Mean and SEM. Two-sided Kruskal-Wallis test.

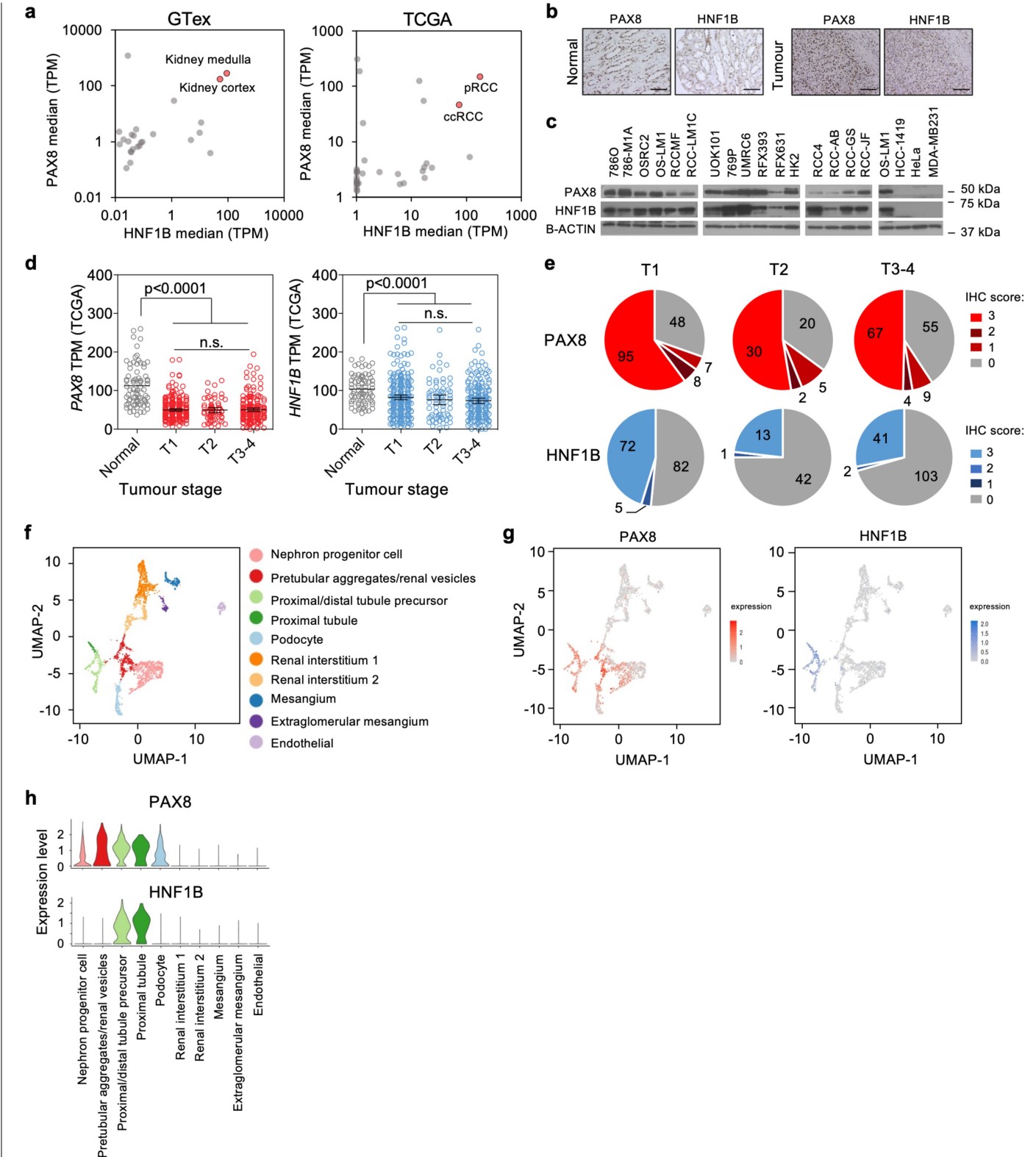

**Extended Data Fig. 2 | PAX8 and HNF1B expression patterns in normal kidney and ccRCC. a.** *PAX8* and *HNF1B* mRNA expression as determined by RNA-seq in the GTex data set of normal tissues and the TCGA cancer data set. Median expression shown for each cancer and tissue type. **b.** PAX8 and HNF1B immunohistochemistry in ccRCC and normal kidney (representative images from TMA in panel e). Scale bar, 100 μm. **c.** Western blot showing PAX8 and HNF1B expression in cell lines (N = 16 biological replicates of cells of renal origin, N = 3 biological replicates of cells of other origins; representative images, N = 2 technical replicates). **d.** PAX8 and HNF1B expression in ccRCCs when compared to normal kidney in the TCGA data set (Normal N = 72, T1 N = 252, T2 N = 66, T3-4 N = 189). Mean and SEM. Two-sided Kruskal-Wallis with Dunn's multiple comparison test. **e.** Immunohistochemistry (IHC) for PAX8 and HNF1B expression in a tissue microarray of ccRCCs. N = 350 (PAX8), N = 361 (HNF1B). **f.** Unsupervised UMAP analysis of scRNA-seq data from fetal human kidney. Different cell types labelled in different colours. **g.** PAX8 and HNF1B expression across different cell types in the fetal human kidney. **h.** PAX8 and HNF1B expression in the cell types identified in scRNA-seq data from fetal human kidney.

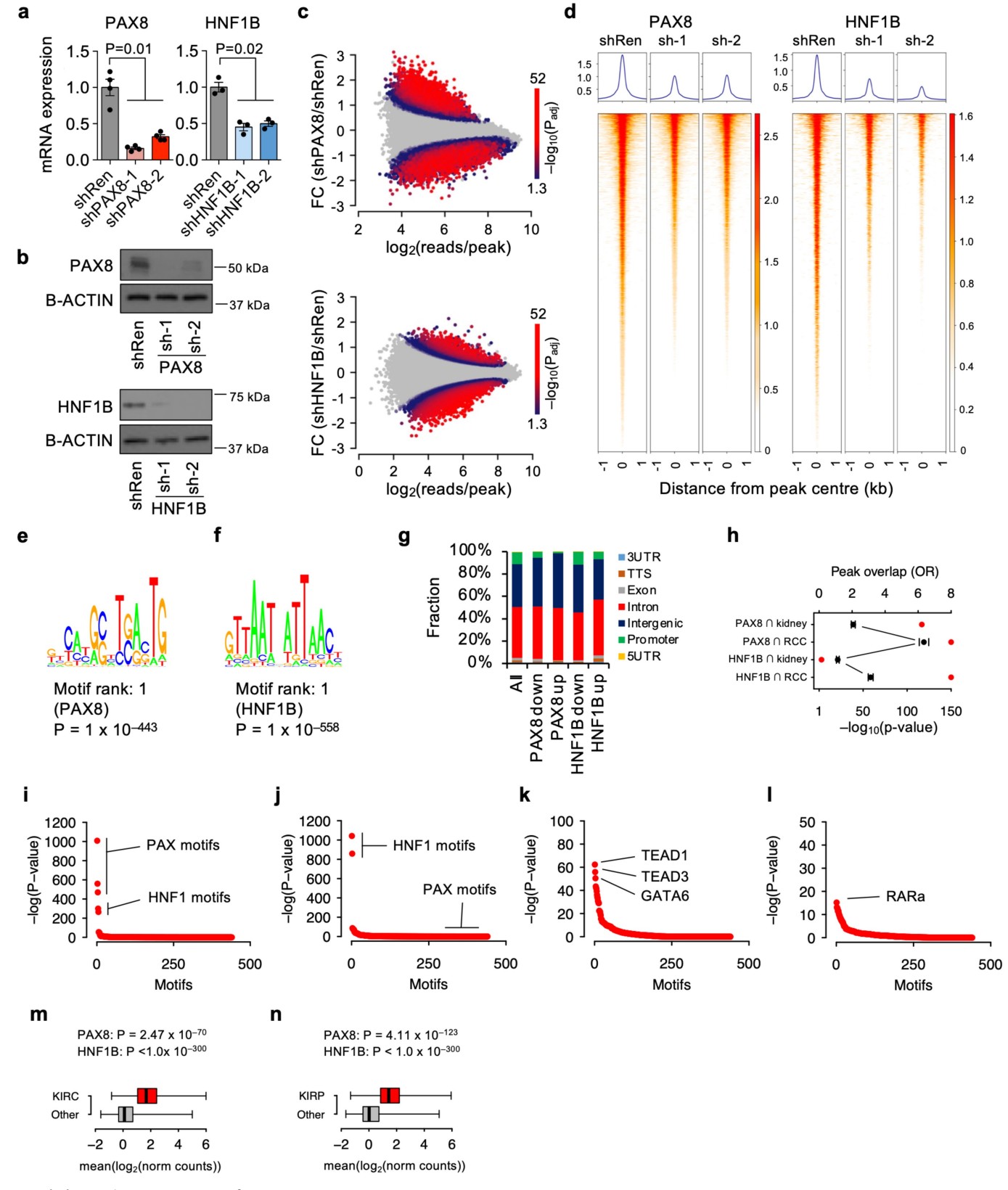

**Extended Data Fig. 3 | PAX8 and HNF1B support chromatin accessibility at distal enhancers. a**. Relative PAX8 and HNF1B mRNA expression as determined by qRT-PCR in 786-M1A cells. Data points, independent RNA preps (shPAX8 N = 4, shHNF1B N = 3). Mean and SEM. Two-sided Kruskal-Wallis test. **b**. Western blot showing PAX8 and HNF1B expression in 786-M1A cells (representative image, N = 2). **c**. ATAC-seq data analysis identifying genomic regions with altered accessibility upon shRNA-mediated PAX8 and HNF1B depletion in 786-M1A cells (two shRNAs with three replicates each). Adjusted two-sided p-values and fold changes derived by DESeq2. **d**. Heatmap showing ATAC-seq signal in the indicated cell lines for regions with reduced accessibility upon PAX8 and HNF1B depletion, respectively (three replicates for each shRNA). **e-f**. The most significant de novo DNA motif enriched in the peak set with reduced accessibility upon PAX8 (e) and HNF1B (f) depletion. **g**. Distribution of ATAC-seq peaks that change upon PAX8 or HNF1B depletion in relation to known transcripts. TTS, transcription termination site. **h**. Overlap between the peak sets with reduced accessibility upon PAX8 and HNF1B depletion in ccRCC cells and peaks sets characteristic of normal kidney and renal cancer identified by DNAse I hypersensitivity mapping in ref. [29]. Top axis, odds ratio of overlap (black), 95% confidence interval. Bottom axis, p-value, one-sided Fisher's exact test (red). **i**. Enrichment of known DNA motifs in the peak set that shows reduced accessibility upon PAX8 depletion. **j**. Enrichment of known DNA motifs in the peak set that shows reduced accessibility upon HNF1B depletion. **k**. Enrichment of known DNA motifs in the peak set that shows increased accessibility upon PAX8 depletion. **l**. Enrichment of known DNA motifs in the peak set that shows increased accessibility upon HNF1B depletion. **m-n**. Mean normalised counts within ccRCC (KIRC)-specific (m) and KIRP-specific (n) ATAC-seq regions in comparison to all other cancer types in the TCGA data set. P-values indicate the significance of enrichment of the de novo PAX8 and HNF1B motifs in the ccRCC and KIRP-specific peak sets. Boxplot, median and interquartile range. Whiskers, data range. N = 26,633 regions for KIRC; N = 68,966 regions for KIRP.

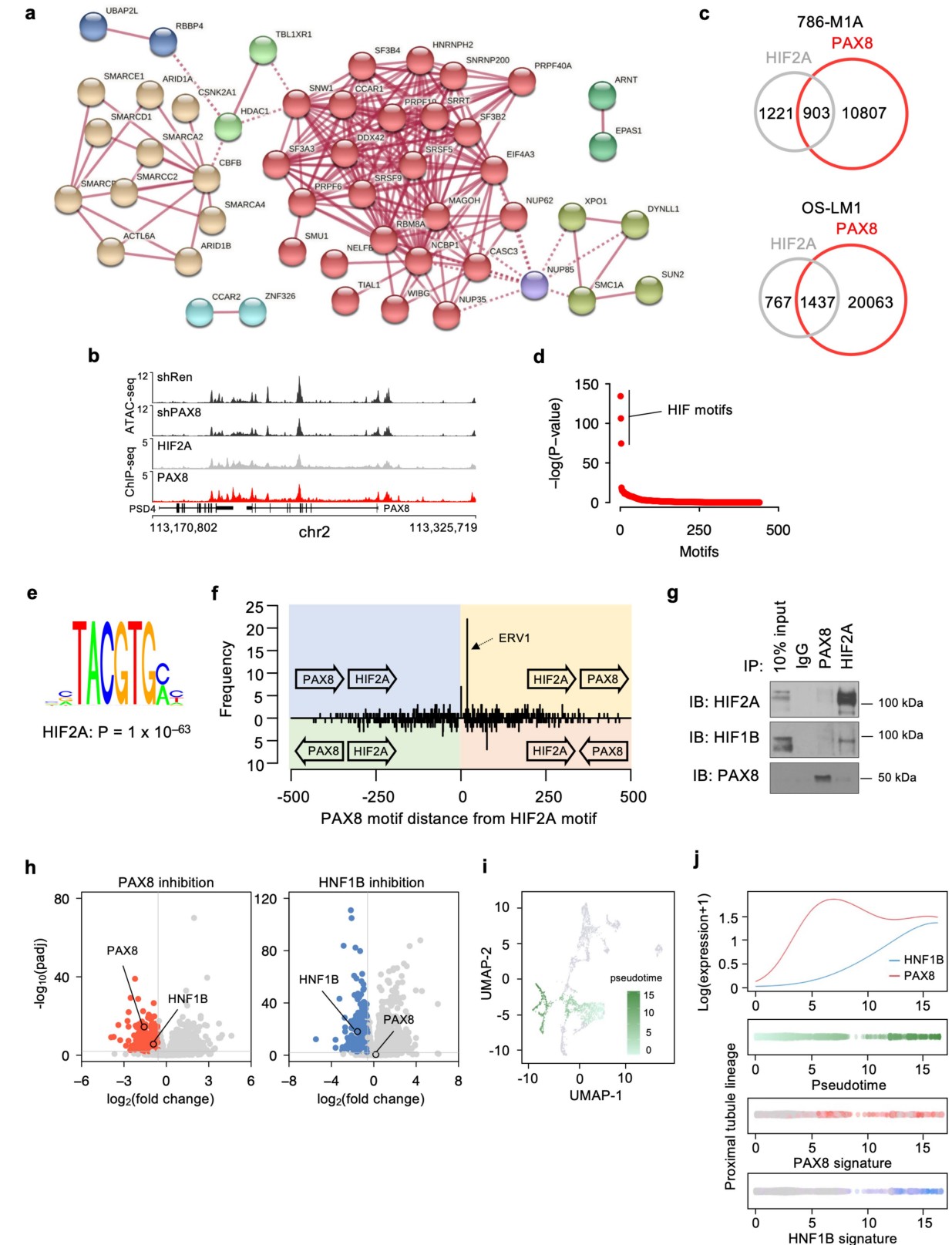

**Extended Data Fig. 4** | See next page for caption.

**Extended Data Fig. 4 | Interaction between PAX8 and HIF2A in ccRCC.**
**a**. Network presentation of the highest confidence experimental and database-derived physical connections between the 89 shared nuclear proteins from HIF2A and PAX8 interactomes as determined by String 11.0. Shading reflects MCL clustering. Isolated nodes are removed. **b**. PAX8 locus. Median ATAC-seq signal from shRen control (N = 6) and shPAX8 (N = 6) cells. Median HIF2A and PAX8 ChIP-seq signal from 786-M1A and OS-LM1 xenografts, 3 tumours each. **c**. Overlap of HIF2A and PAX8 ChIP-seq peaks in 786-M1A (top) and OS-LM1 (bottom) cells. **d**. Enrichment of known DNA motifs in the 1,948 peaks with most significantly increased DNA accessibility in ccRCCs when compared to KIRP tumours in the TCGA data set (Fold change 2, two-sided padj < 0.001 as determined by DESeq2). **e**. The most significant de novo DNA motif enriched in the ccRCC-specific peaks as described in panel (d). **f**. Distribution of distances between the centres of PAX8 and HIF2A DNA motifs in ccRCC-specific ATAC-seq regions. Cartoons in each quadrant demonstrate the motif orientation. The 18bp distance seen in the common ERV1 endogenous retrovirus highlighted. **g**. Co-immunoprecipitation with antibodies targeting HIF2A and PAX8 in C-M1A$^{HIF2A-/-}$ cells with HIF2A reintroduction (PAX8-HIF2A interaction, N = 3 independent IP reactions; HIF2A-HIF1B interaction N = 1). **h**. Global gene expression changes by RNA-seq in 786-M1A and OS-LM1 ccRCC cells upon PAX8 and HNF1B depletion when compared to non-targeting controls. Pooled analysis of both cell lines and targeting constructs. Adjusted two-sided p-value derived by DESeq2. **i**. Pseudotime analysis of the different stages of the proximal renal epithelium lineage in fetal human kidney for the cell types shown in Extended Data Fig. 2f. **j**. Expression of PAX8, HNF1B and the respective gene signatures from ccRCC cell lines as a function of the proximal renal epithelium lineage pseudotime in the fetal human kidney.

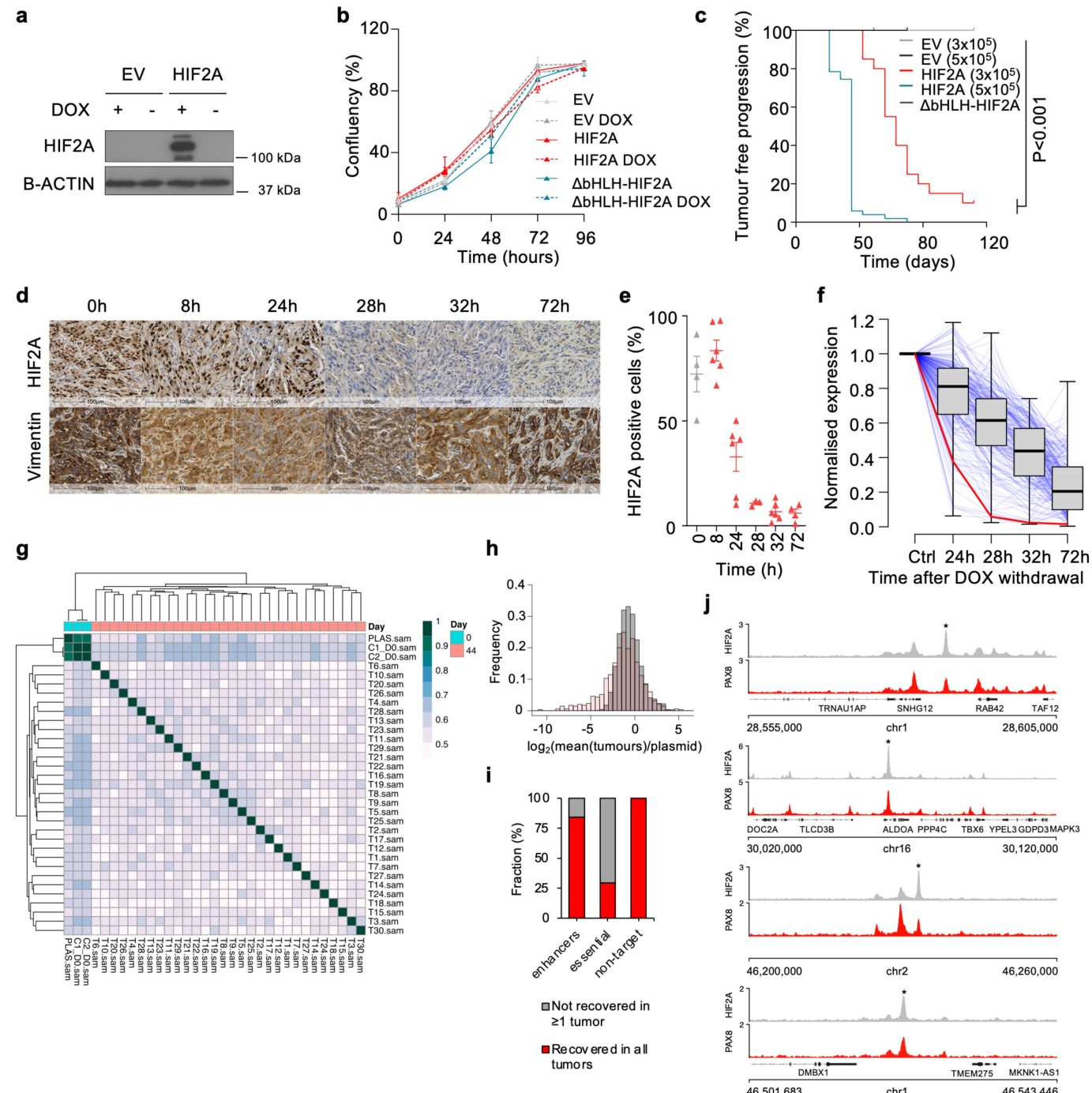

**Extended Data Fig. 5 | In vivo CRISPRi screen for oncogenic HIF2A enhancers. a.** Western blot showing doxycycline-dependent HIF2A expression in C-M1A^HIF2A-/- cells (representative image, N = 3). **b.** Proliferation of C-M1A^HIF2A-/- cells with or without HIF2A. ΔbHLH-HIF2A, DNA-binding domain HIF2A mutant. Data points, mean of three replicates per condition, SEM. **c.** Tumour-free survival of athymic mice upon inoculation of C-M1A^HIF2A-/- cells with or without doxycycline-dependent HIF2A reintroduction. ΔbHLH-HIF2A, DNA-binding domain HIF2A mutant. Logrank test. EV (3 x 10^5 cells), N = 10; EV (5 x 10^5 cells), N = 9; HIF2A (3 x 10^5 cells), N = 20; HIF2A (5 x 10^5 cells), N = 51; ΔbHLH-HIF2A (5 x 10^5 cells), N = 10. **d–e.** HIF2A protein expression at different time points after doxycycline withdrawal as determined by immunohistochemistry in xenograft tumours formed by C-M1A^HIF2A-/- cells with doxycycline-inducible HIF2A reintroduction. Mean and SEM. 0h, 72h, N = 4; 28h N = 3; 24 h, 48 h, 32 h, N = 6 tumour regions. **f.** mRNA expression of 205 genes identified as downregulated at 32h post doxycycline withdrawal with sustained low expression at 72h post doxycycline withdrawal in vivo in C-M1A^HIF2A-/- cells. Red line shows HIF2A expression. Boxplot, median and interquartile range. Whiskers, data range. **g.** Hierarchical clustering based on Pearson's correlation coefficient of the in vivo CRISPRi screening data. C1 and C2, control samples collected on day 0 before inoculation into mice. PLAS, plasmid DNA. T1-T30, individual tumours. **h.** Average distribution of sgRNA construct abundance calculated based on all tumours in relation to initial abundance in the plasmid library (red). Expected distribution (grey) based on 10,000 permutations. **i.** Percentage of constructs recovered in tumours from in vivo CRISPRi screen. **j.** HIF2A and PAX8 ChIP-seq tracks for the loci of the top enhancer dependencies from Fig. 2b. HIF2A and PAX8 ChIP-seq tracks overlapped from 786-M1A and OS-LM1 xenografts, 3 tumours each.

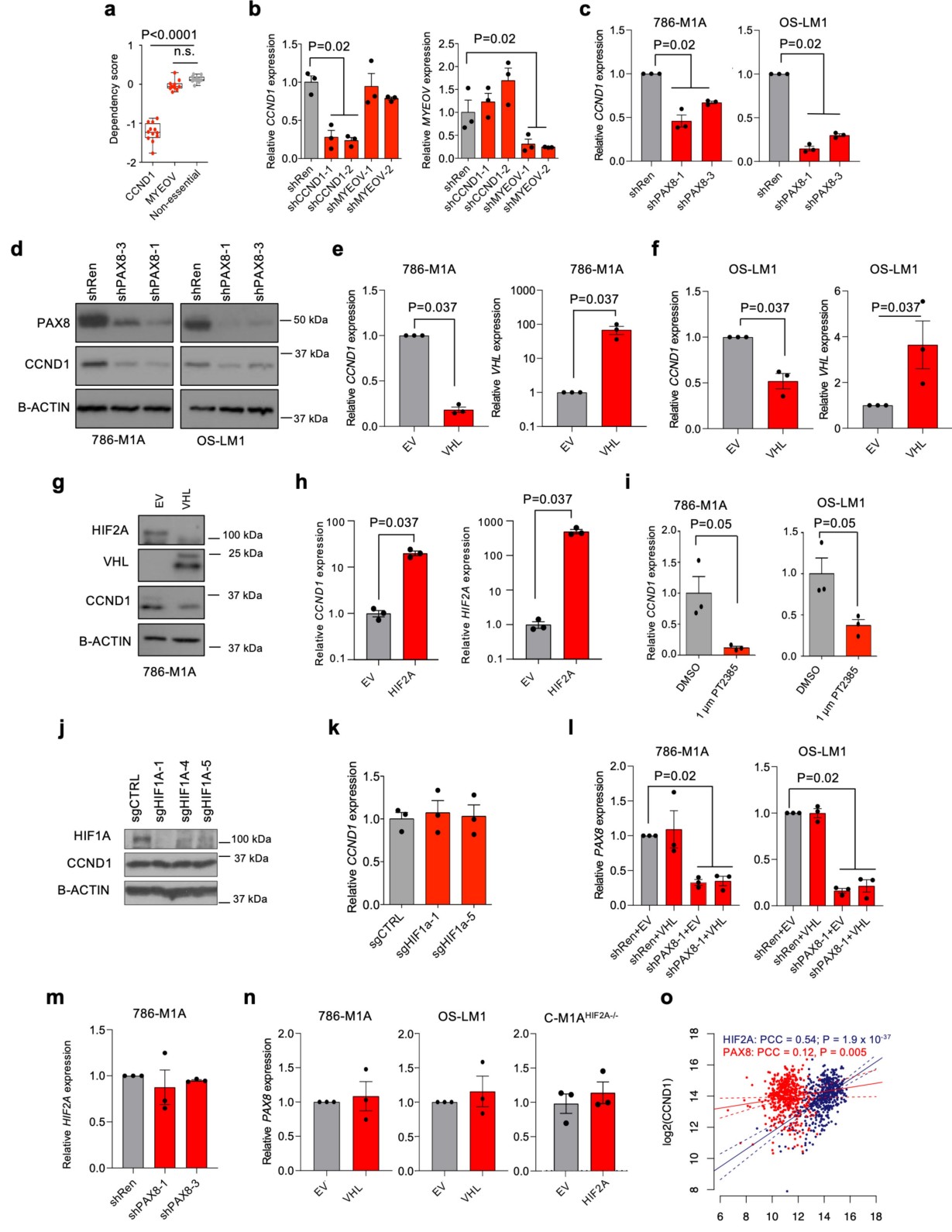

**Extended Data Fig. 6** | See next page for caption.

**Extended Data Fig. 6 | CCND1 expression depends on PAX8 and HIF2A in ccRCC cells. a**. *CCND1*, *MYEOV* and non-essential gene (*ADAM18*) dependency (CERES score) across 12 ccRCC cell lines in the DepMap data set. Two-sided Kruskal-Wallis test with Dunn's multiple comparison test. Boxplot, median and interquartile range. Whiskers, min-max. **b**. Relative *CCND1* and *MYEOV* expression as determined by qRT-PCR in 786-M1A cells. **c**. Relative *CCND1* expression as determined by qRT-PCR following PAX8 knockdown in 786-M1A and OS-LM1 cells. **d**. PAX8 and CCND1 protein expression following PAX8 knock-down in 786-M1A and OS-LM1 cells as determined by Western blotting (N = 2 biological replicates in different cell lines, N = 1 technical replicate).

**e**–**f**. Relative mRNA expression as determined by qRT-PCR. **g**. VHL, CCND1 and B-actin protein expression as determined by Western blotting in 786-M1A cells (N = 1). **h**–**i**. Relative mRNA expression as determined by qRT-PCR. EV, empty vector. **j**. HIF1A, CCND1 and B-actin protein expression as determined by Western blotting in OS-LM1 cells (representative image, N = 2). **k**–**n**. Relative mRNA expression as determined by qRT-PCR. EV, empty vector. **o**. Correlation between *HIF2A* and *CCND1*, and *PAX8* and *CCND1* mRNA levels, respectively, in the TCGA ccRCC data set. PCC, Pearson's correlation coefficient. **b,c,e,f,h,i,k,l,m,n**. Data points, independent RNA preps (N = 3). Mean and SEM. Two-sided Kruskal-Wallis test.

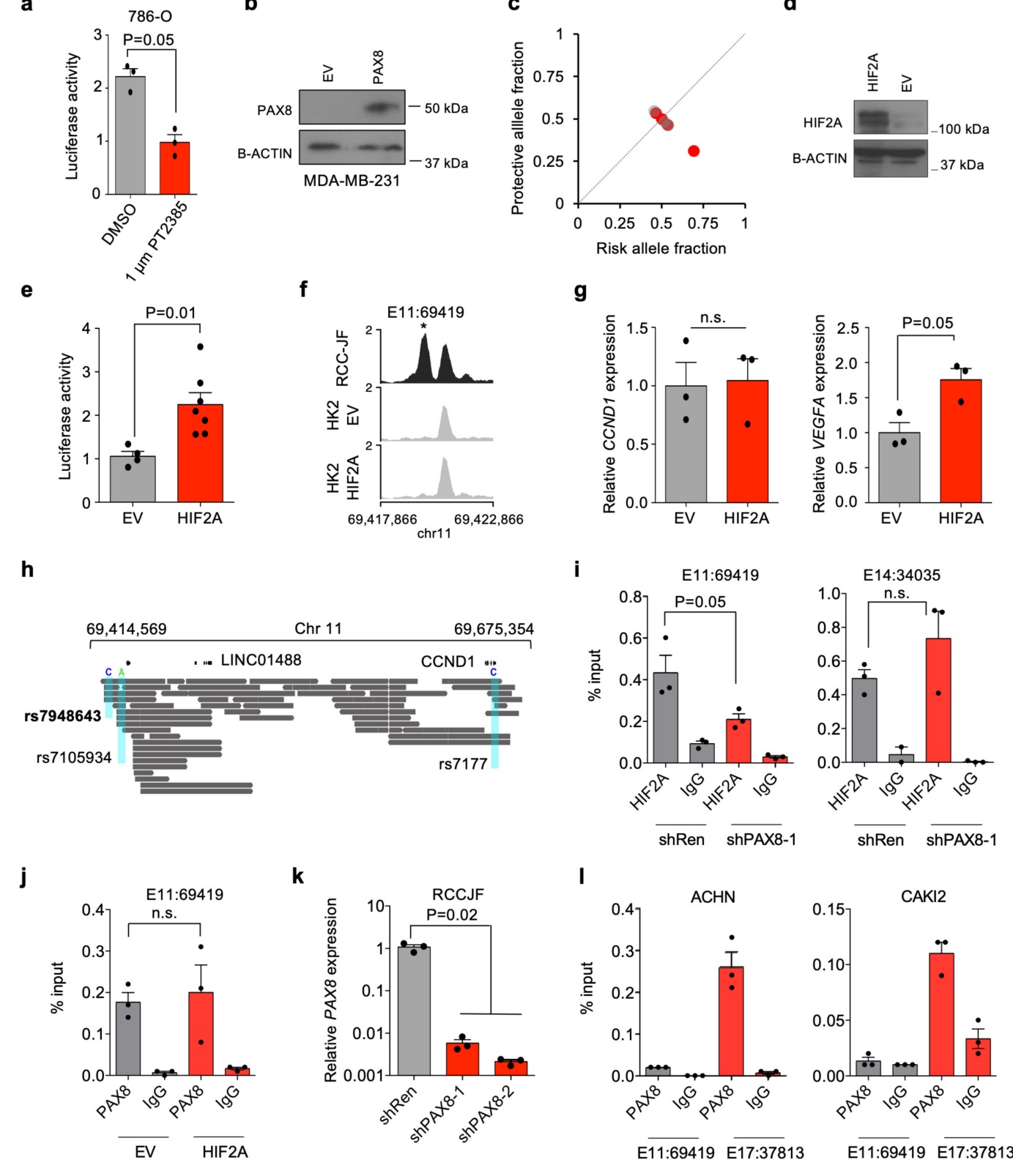

**Extended Data Fig. 7** | See next page for caption.

**Extended Data Fig. 7 | Transcriptional activation of E11:69419 by PAX8 and HIF2A. a**. Luciferase reporter assay showing E11:69419 activity in 786-O cells with and without HIF2A inhibition. Data points, average of three technical replicates, independent transfections (N = 3). Mean and SEM. Two-sided Kruskal-Wallis test. **b**. PAX8 protein expression in MDA-MB-231 cells used in EMSA in Fig. 3f, g as determined by Western blotting (N = 1). **c**. Relative ccRCC risk and protective allele fractions at E11:69419 in ATAC-seq data from three replicates of RCC-JF cells (grey) and four heterozygous human ccRCC samples (red). **d**. VHL-resistant HIF2A protein expression in HK2 cells as determined by Western blotting (representative image, N = 2). **e**. Luciferase reporter assay showing E11:69419 activity in HK2 cells from panel (d). Data points, average of three technical replicates, independent transfections (EV N = 4, HIF2A N = 7). Mean and SEM. Two-sided Kruskal-Wallis test. **f**. Median ATAC-seq signal from RCC-JF cells and HK2 cells from panel (d) six weeks after transduction with HIF2A. Three replicates in each condition. **g**. Relative mRNA expression as determined by qRT-PCR in HK2 cells. Data points, independent RNA preps (N = 3). Mean and SEM. Two-sided Kruskal-Wallis test. **h**. Graphical representation of the long DNA reads used for phasing of the RCC protective allele in the rs7948643-heterozygous ccRCC cell line RCC-JF. Variants of interest highlighted. **i-j**. ChIP qPCR at E11:69419 and PAX8-independent region E14:34035. 786-M1A cells with and without PAX8 depletion immunoprecipitated with HIF2A antibodies or IgG (i) and C-M1A$^{HIF2A-/-}$ cells with and without HIF2A immunoprecipitated with PAX8 antibodies or IgG (j). Data points, independent IP reactions (N = 3). Mean and SEM. Two-sided Kruskal-Wallis test. **k**. Relative *PAX8* mRNA expression as determined by qRT-PCR in RCC-JF cells. Data points, independent RNA preps (N = 3). Mean and SEM. Two-sided Kruskal-Wallis test. **l**. ChIP qPCR at E11:69419 and at a PAX8-bound HNF1B enhancer (E17:37813) in papillary RCC cells immunoprecipitated with PAX8 antibodies or IgG. Data points, independent IP reactions (N = 3). Mean and SEM.

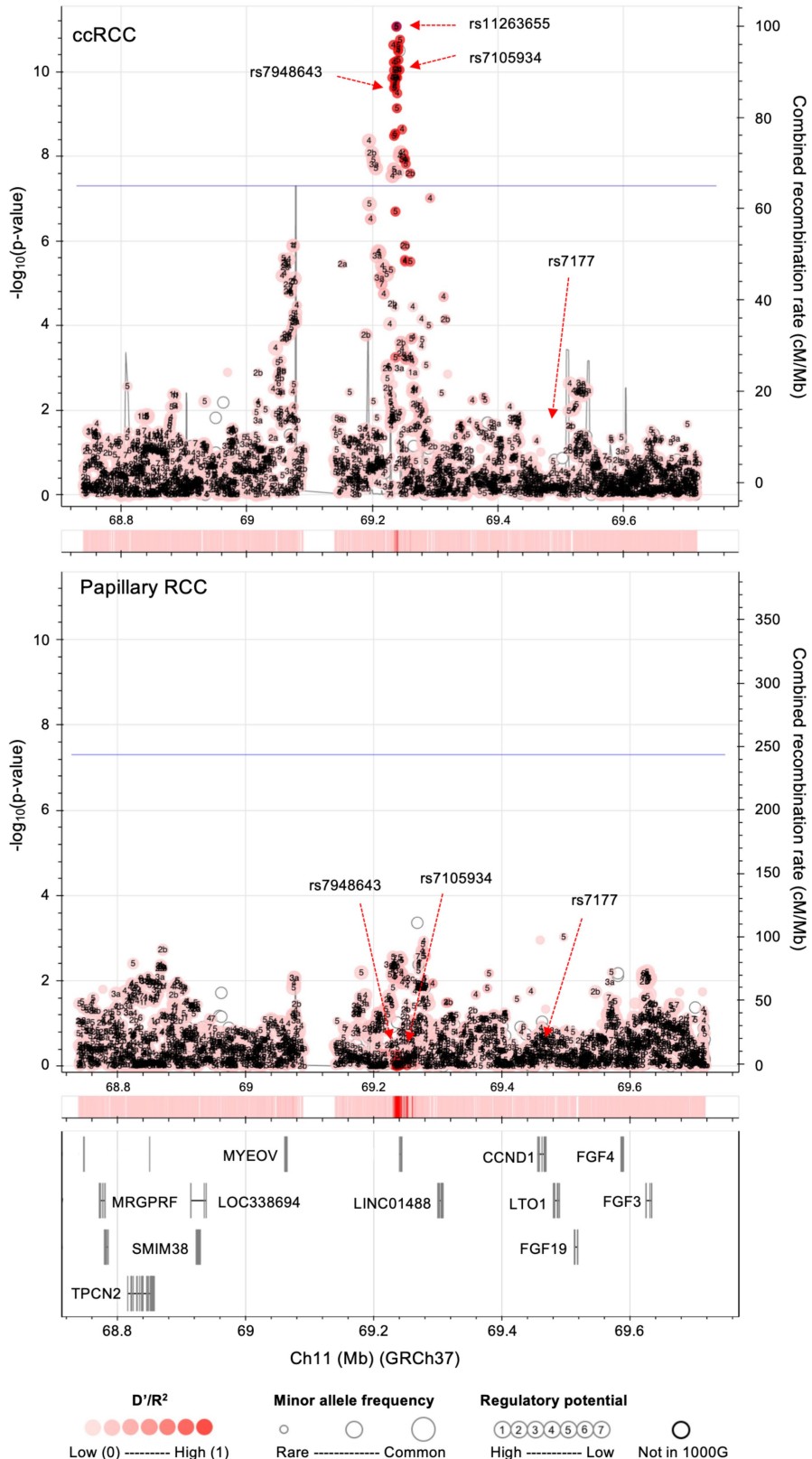

**Extended Data Fig. 8 | Genetic variants at 11q13.3 are association with increased ccRCC risk.** Regional association plots for ccRCC (top) and papillary RCC (bottom). D′/R2 estimated and plot generated by LDassoc. Regulatory potential estimated by RegulomeDB. Recombination rate overlaid on the plot. Blue line, genome-wide significance (P < 5 x 10⁸). Rug plots show variant density, nearby genes and transcripts plotted at the bottom.

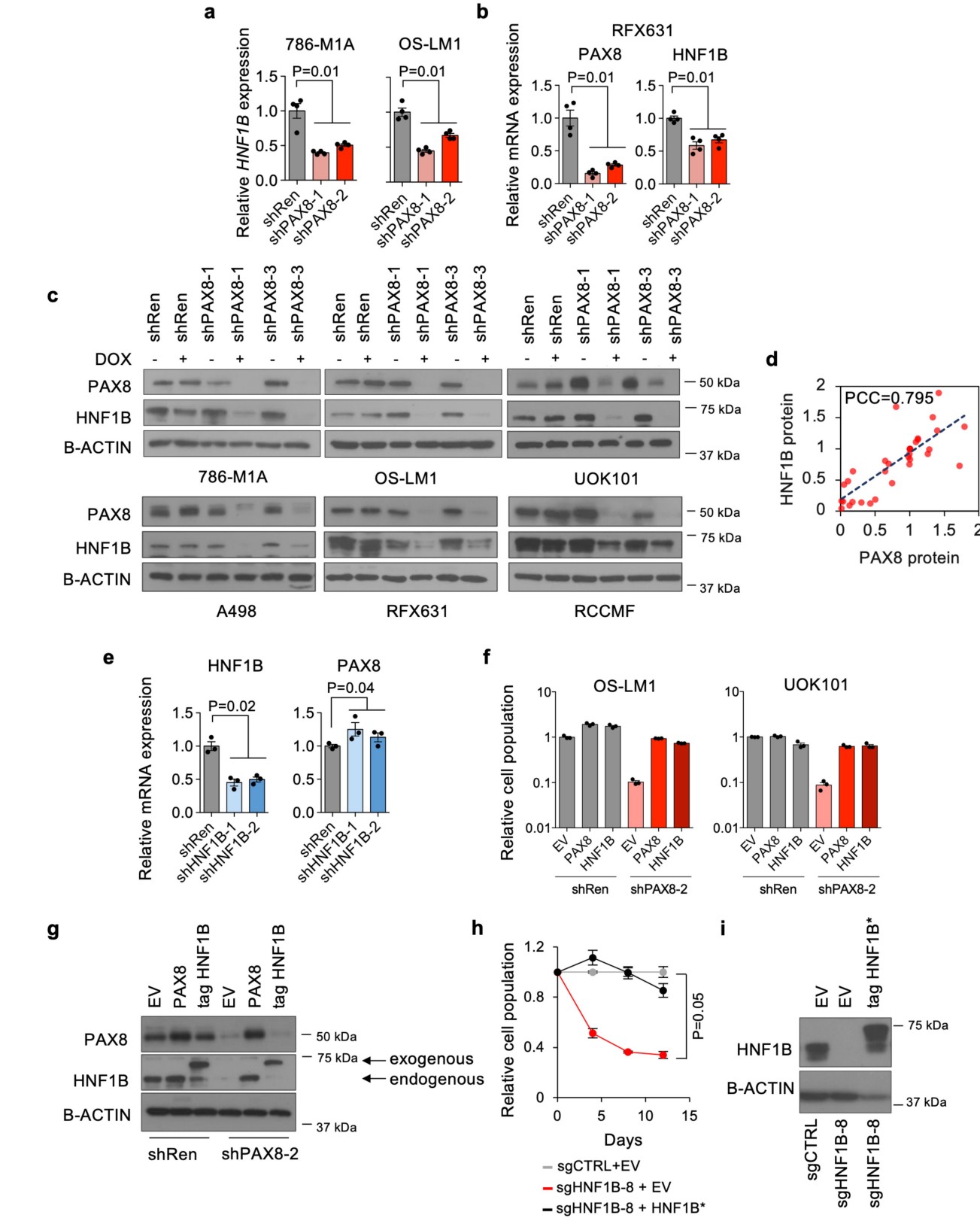

**Extended Data Fig. 9** | See next page for caption.

**Extended Data Fig. 9 | PAX8 controls HNF1B expression in ccRCC.**
**a**–**b**. Relative *PAX8* and *HNF1B* mRNA expression as determined by qRT-PCR.
**c**. PAX8 and HNF1B protein expression as determined by Western blotting.
Doxycycline-inducible shRNA constructs targeting PAX8 (N = 6 biological
replicates across different cell lines, N = 1 technical replicates for each
condition). **d**. Correlation between PAX8 and HNF1B expression in the samples
from panel (c). PCC, Pearson's correlation coefficient. **e**. Relative PAX8 and
HNF1B mRNA expression as determined by qRT-PCR in 786-M1A cells.
**f**. Competitive proliferation assay against EV-shRen control cells in the UOK101
and OS-LM1 ccRCC models. Relative proportion of the indicated cells on day 12
when compared to day 0. Data points, technical replicates (N = 3). Mean and

SEM. **g**. Western blot showing PAX8 and HNF1B expression in the OS-LM1 cells
used in the competition assay in panel (N = 3 biological replicates across
different cell lines, N = 1 technical replicate) (f). **h**. Competitive proliferation
assay against EV-sgCTRL control condition in the 786-M1A cells. Cells
expressing sgHNF1B-8 were transduced with either an EV control construct
or a construct expressing HNF1B* where the sgHNF1B-8 target site is mutated.
Data points, technical replicates (N = 3), SEM. Two-sided Kruskal-Wallis test.
**i**. HNF1B expression by Western blotting in the cells used for competition
assays in panel (N = 1 technical replicate) (h). **a,b,e**. Data points, independent
RNA preps (shPAX8 N = 4, shHNF1B N = 3). Mean and SEM. Two-sided Kruskal-
Wallis test.

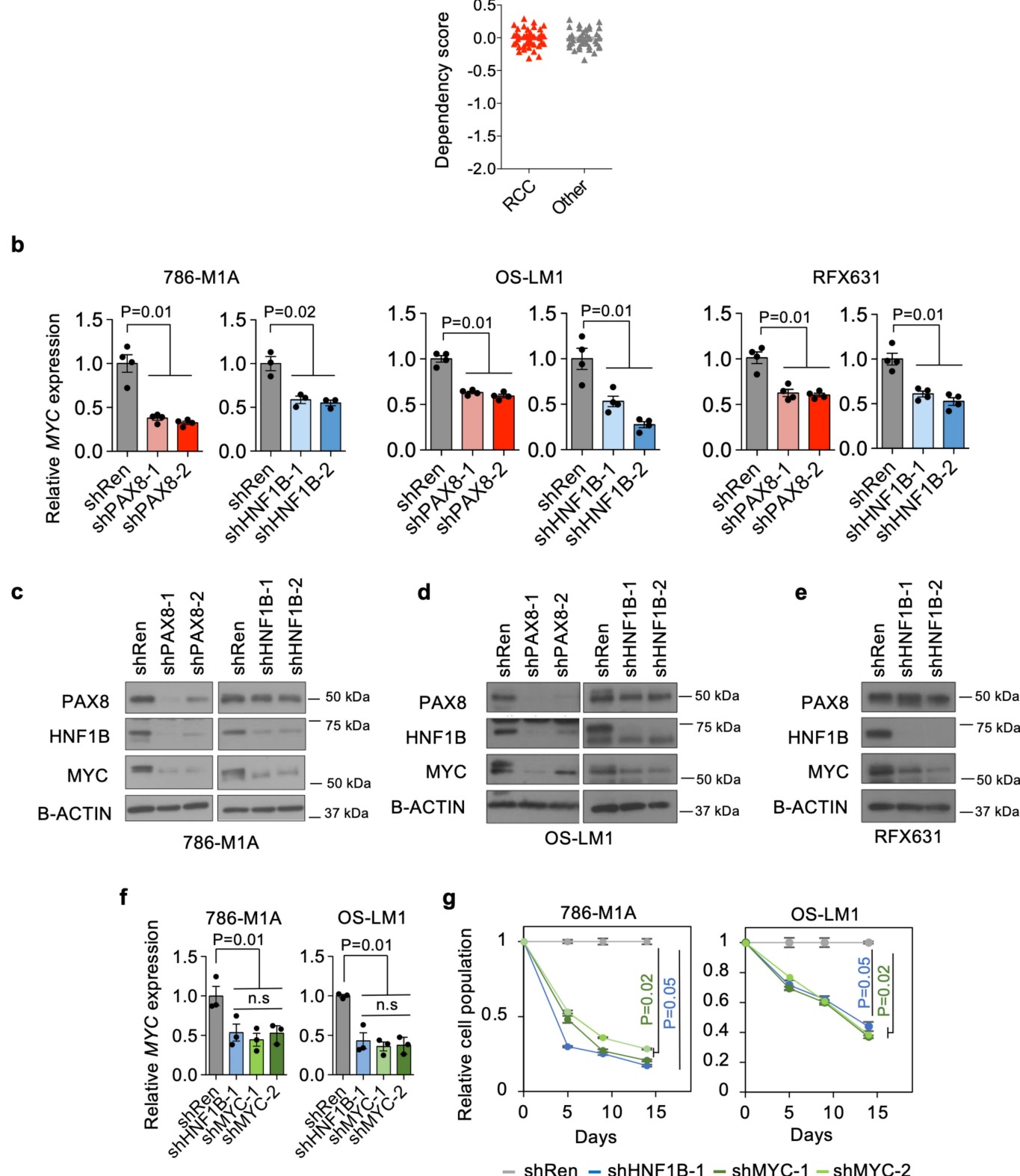

**Extended Data Fig. 10 | The PAX8-HNF1B module regulates MYC expression in ccRCC cells. a**. Dependency score (CERES score) in the DepMap CRISPR-Cas9 screen data set for 41 shared genes upregulated by PAX8 or HNF1B inhibition. **b**. Relative *MYC* mRNA expression as determined by qRT-PCR. Data points, independent RNA preps (shPAX8 N = 4, shHNF1B N = 3). Mean and SEM. Two-sided Kruskal-Wallis test. **c-e**. Western blot in 786-M1A, OS-LM1 and RFX-631 cells (N = 3 and N = 2 biological replicates across different cell lines for shPAX8 and shHNF1B, respectively; N = 1 technical replicate for each biological replicate). **f**. Relative *MYC* mRNA expression as determined by qRT-PCR in the 786-M1A and OS-LM1 cells used in the competition assay in panel (g). Data points, independent RNA preps (N = 3). Mean and SEM. Two-sided Kruskal-Wallis test. **g**. Competitive proliferation assay against shRen control cells. Data points, technical replicates (N = 3), standard deviation. Two-sided Kruskal-Wallis test.

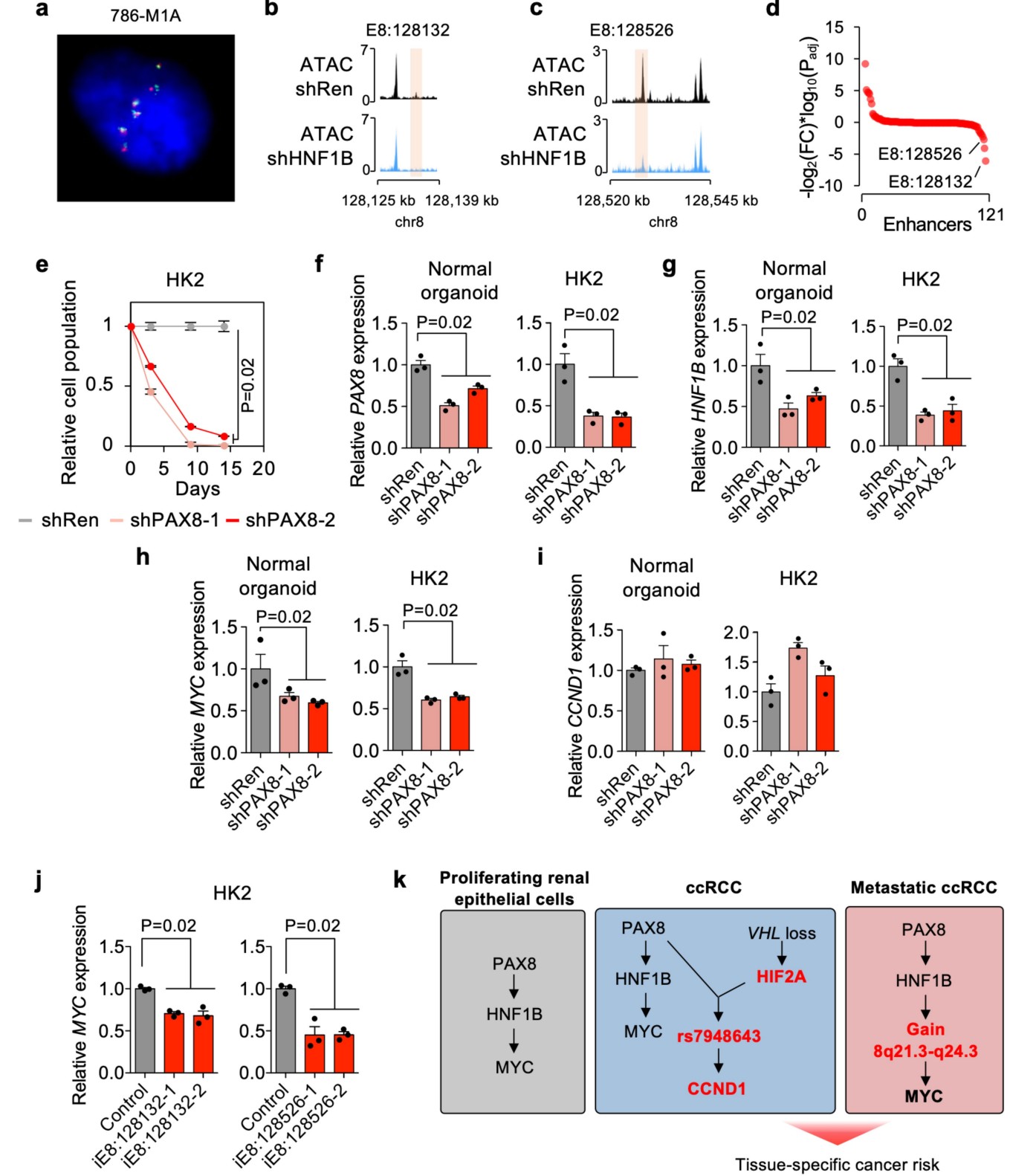

**Extended Data Fig. 11** | See next page for caption.

**Extended Data Fig. 11 | Lineage factor dependent *MYC* expression in normal renal epithelial cells. a**. Fluorescence in situ hybridisation in one nucleus of 786-M1A cells (representative image, N = 4). Blue, DAPI staining; green, chromosome 8q telomere; orange *MYC* 5'. **b**–**c**. ATAC-seq tracks overlapped for 786-M1A cells upon HNF1B depletion. shRen, N = 5; shHNF1B, N = 6. **d**. Inhibitory effect of PAX8 depletion on chromatin accessibility at the enhancers in the *MYC* locus. Fold changes and adjusted two-sided p-values derived by DESeq2. **e**. Competitive proliferation assay against shRen control cells in HK2 cells. Data points, technical replicates (N = 3), standard deviation. Two-sided Kruskal-Wallis test. **f**–**j**. Relative mRNA expression as determined by qRT-PCR. Data points, independent RNA preps (N = 3). Mean and SEM. Two-sided Kruskal-Wallis test. **k**. Summary. PAX8 is required for tissue-specific oncogenic programmes by integrating signals from inherited and acquired genetic alterations: inactivating mutations in *VHL* and the common ccRCC predisposition SNP rs7948643 upstream of *CCND1*, as well as metastasis-associated 8q21.3-q24.3 amplifications upstream of *MYC*, which co-opt the physiological PAX8-HNF1B program that supports MYC expression in proliferating normal renal epithelial cells.

**Extended Data Table 1 | GWAS meta-analysis**

| RCC subtype | SNP | Location | Group | Imputation score | N controls | N cases | Reference allele | Effect allele | Effect allele freq controls | Effect allele freq cases | OR | 95% CI | P | Phet | I² |
|---|---|---|---|---|---|---|---|---|---|---|---|---|---|---|---|
| ccRCC | rs7105934 | chr11:69,239,741 | NCI-1 | 1 | 3,424 | 446 | G | A | 0.0834 | 0.0695 | 0.97 | (0.71-1.33) | 0.84272 | | |
| ccRCC | rs7105934 | chr11:69,239,741 | NCI-2 | 0.99449 | 4,390 | 1,603 | G | A | 0.0789 | 0.0573 | 0.73 | (0.62-0.85) | 0.0001047 | | |
| ccRCC | rs7105934 | chr11:69,239,741 | MDA | 1 | 556 | 583 | G | A | 0.0791 | 0.0635 | 0.77 | (0.55-1.07) | 0.11529 | | |
| ccRCC | rs7105934 | chr11:69,239,741 | IARC | U | 6,640 | 3,016 | G | A | 0.0668 | 0.0447 | 0.65 | (0.56-0.76) | 5.46E-08 | | |
| ccRCC | rs7105934 | chr11:69,239,741 | meta-analysis | | 15,010 | 5,648 | | | 0.0746 | 0.0522 | 0.72 | (0.65-0.79) | 8.79E-11 | 0.17 | 40.60 |
| ccRCC | rs7948643 | chr11:69,234,620 | NCI-1 | 0.99916 | 3,424 | 446 | T | C | 0.0833 | 0.0695 | 0.98 | (0.71-1.34) | 0.87598 | | |
| ccRCC | rs7948643 | chr11:69,234,620 | NCI-2 | 0.98318 | 4,391 | 1,603 | T | C | 0.078 | 0.0565 | 0.72 | (0.61-0.85) | 9.34E-05 | | |
| ccRCC | rs7948643 | chr11:69,234,620 | MDA | 0.99853 | 556 | 583 | T | C | 0.0782 | 0.064 | 0.78 | (0.56-1.09) | 0.15264 | | |
| ccRCC | rs7948643 | chr11:69,234,620 | IARC | U | 6,640 | 3,016 | T | C | 0.0655 | 0.0444 | 0.66 | (0.57-0.77) | 1.43E-07 | | |
| ccRCC | rs7948643 | chr11:69,234,620 | meta-analysis | | 15,011 | 5,648 | | | 0.0737 | 0.0519 | 0.72 | (0.65-0.80) | 2.44E-10 | 0.18 | 39.02 |
| pRCC | rs7105934 | chr11:69,239,741 | NCI-1 | 1 | 3,424 | 81 | G | A | 0.0834 | 0.0679 | 0.87 | (0.47-1.64) | 0.67323 | | |
| pRCC | rs7105934 | chr11:69,239,741 | NCI-2 | 0.99405 | 4,220 | 111 | G | A | 0.0783 | 0.1069 | 1.40 | (0.89-2.22) | 0.1448 | | |
| pRCC | rs7105934 | chr11:69,239,741 | MDA | 1 | 556 | 76 | G | A | 0.0791 | 0.0723 | 0.89 | (0.46-1.70) | 0.7198 | | |
| pRCC | rs7105934 | chr11:69,239,741 | IARC | U | 6,640 | 295 | G | A | 0.0667 | 0.0523 | 0.83 | (0.56-1.22) | 0.3399 | | |
| pRCC | rs7105934 | chr11:69,239,741 | meta-analysis | | 14,840 | 563 | G | A | 0.0744 | 0.0680 | 0.99 | (0.78-1.27) | 0.9557 | 0.32 | 14.81 |
| pRCC | rs7948643 | chr11:69,234,620 | NCI-1 | 0.99909 | 3,424 | 81 | T | C | 0.0833 | 0.0679 | 0.88 | (0.47-1.65) | 0.6854 | | |
| pRCC | rs7948643 | chr11:69,234,620 | NCI-2 | 0.98317 | 4,220 | 111 | T | C | 0.0773 | 0.1069 | 1.53 | (0.92-2.55) | 0.104 | | |
| pRCC | rs7948643 | chr11:69,234,620 | MDA | 1 | 556 | 76 | T | C | 0.0782 | 0.0724 | 0.90 | (0.47-1.74) | 0.7615 | | |
| pRCC | rs7948643 | chr11:69,234,620 | IARC | U | 6,640 | 295 | T | C | 0.0655 | 0.0512 | 0.82 | (0.56-1.21) | 0.32 | | |
| pRCC | rs7948643 | chr11:69,234,620 | meta-analysis | | 14,840 | 563 | T | C | 0.0735 | 0.0675 | 0.98 | (0.76-1.27) | 0.9027 | 0.28 | 22.54 |

Summary data for the GWAS meta-analysis for ccRCC and papillary RCC. U, unknown; OR, odds ratio; Phet, P value for heterogeneity test across data sets.

# Reporting Summary

Nature Research wishes to improve the reproducibility of the work that we publish. This form provides structure for consistency and transparency in reporting. For further information on Nature Research policies, see our Editorial Policies and the Editorial Policy Checklist.

## Statistics

For all statistical analyses, confirm that the following items are present in the figure legend, table legend, main text, or Methods section.

| n/a | Confirmed | |
|---|---|---|
| ☐ | ☒ | The exact sample size ($n$) for each experimental group/condition, given as a discrete number and unit of measurement |
| ☐ | ☒ | A statement on whether measurements were taken from distinct samples or whether the same sample was measured repeatedly |
| ☐ | ☒ | The statistical test(s) used AND whether they are one- or two-sided *Only common tests should be described solely by name; describe more complex techniques in the Methods section.* |
| ☒ | ☐ | A description of all covariates tested |
| ☐ | ☒ | A description of any assumptions or corrections, such as tests of normality and adjustment for multiple comparisons |
| ☐ | ☒ | A full description of the statistical parameters including central tendency (e.g. means) or other basic estimates (e.g. regression coefficient) AND variation (e.g. standard deviation) or associated estimates of uncertainty (e.g. confidence intervals) |
| ☐ | ☒ | For null hypothesis testing, the test statistic (e.g. $F$, $t$, $r$) with confidence intervals, effect sizes, degrees of freedom and $P$ value noted *Give P values as exact values whenever suitable.* |
| ☒ | ☐ | For Bayesian analysis, information on the choice of priors and Markov chain Monte Carlo settings |
| ☒ | ☐ | For hierarchical and complex designs, identification of the appropriate level for tests and full reporting of outcomes |
| ☐ | ☒ | Estimates of effect sizes (e.g. Cohen's $d$, Pearson's $r$), indicating how they were calculated |

*Our web collection on statistics for biologists contains articles on many of the points above.*

## Software and code

Policy information about availability of computer code

| Data collection | No code was used for data collection |
|---|---|
| Data analysis | The following programs were used for data analysis:<br>R (3.2.3, 3.6.2 and 4.0.3)<br>Bowtie2 (2.3.4.3)<br>FastQC (0.11.9)<br>BWA (0.7.17)<br>Samtools (1.2 and 1.9)<br>deepTools (3.5.0)<br>MACS2 (2.2.7.1)<br>deepTools (3.5.0)<br>clusterprofiler (3.16.1)<br>BD FACSDiva software (8.0.1)<br>longshot (0.4.1)<br>pysam (0.16)<br>htslib (1.14)<br>SCTransform (0.3.2)<br>Seurat (3.2.2)<br>Tidyverse (1.3.0)<br>Rcolorbrewer (1.1-2)<br>ComplexHeatmap (2.4.3)<br>Slingshot (1.6.1)<br>tradeSeq (1.2.01) |

ggbeeswarm (0.6.0)
cutadapt (version 2.10)
RSamtools (2.0.3)
DESeq2 (1.24.0)
BEDOPS (2.4.39)
dplyr (2.1.1)
bedtools (2.27.1)
Minimac (version 3)
IMPUTE2 (version 2.2.2)
SNPTEST (version 2.2)
LDassoc

Custom computer code used in this study is available at 10.5281/zenodo.6335339.

For manuscripts utilizing custom algorithms or software that are central to the research but not yet described in published literature, software must be made available to editors and reviewers. We strongly encourage code deposition in a community repository (e.g. GitHub). See the Nature Research guidelines for submitting code & software for further information.

## Data

Policy information about availability of data

All manuscripts must include a data availability statement. This statement should provide the following information, where applicable:
- Accession codes, unique identifiers, or web links for publicly available datasets
- A list of figures that have associated raw data
- A description of any restrictions on data availability

All ATAC-seq, RNA-seq and ChIP-seq data generated within this project have been uploaded into the Gene Expression Omnibus under the access code GSE163001 with the subseries GSE162948, GSE163000, GSE163485 and GSE163487. The mass spectrometry proteomics data have been deposited to the ProteomeXchange Consortium via the PRIDE90 partner repository with the dataset identifier PXD029522. Human RNA-Seq data for different tissue types were downloaded from the TCGA data portal (https://tcga-data.nci.nih.gov/) and from the GTex portal (https://gtexportal.org/). TCGA ATAC-seq normalised count data were downloaded from https://gdc.cancer.gov/about-data/publications/ATACseq-AWG. Normalised DNA accessibility signal data were downloaded from https://zenodo.org/record/3838751#.YJgMJC2ZM0o. Molecular signature data were downloaded from the Molecular Signature Database (MSigDB version 7.1.1) (http://www.gsea-msigdb.org/gsea/msigdb/). Protein interaction data were obtained from the STRING database version 11.0 (https://string-db.org). CRISPR-Cas9 screen CERES scores were downloaded from https://portals.broadinstitute.org/achilles. RCC GWAS meta-analysis summary data were provided by M.P.P. (purduem@mail.nih.gov) and S.J.C. (chanocks@mail.nih.gov) and they are available in the Supplementary Tables 7-8. Data from the original GWAS studies that comprise the meta-analysis data set are available either from dbGaP (NCI-1, accession number phs000351.v1.p1; NCI-2, phs001736.v1.p1; IARC-2, phs001271.v1.p1) or from the investigators upon reasonable request (IARC-1, Paul Brennan, brennanp@iarc.fr; MDA, Jian Gu, jiangu@mdanderson.org). Other data that support the findings of this study are available from the corresponding author upon reasonable request.

# Field-specific reporting

Please select the one below that is the best fit for your research. If you are not sure, read the appropriate sections before making your selection.

☒ Life sciences  ☐ Behavioural & social sciences  ☐ Ecological, evolutionary & environmental sciences

For a reference copy of the document with all sections, see nature.com/documents/nr-reporting-summary-flat.pdf

# Life sciences study design

All studies must disclose on these points even when the disclosure is negative.

| | |
|---|---|
| Sample size | No statistical method was used to predetermine sample size. For key in vitro experiments, a minimum of three independent experiments were performed, with further validation in additional model cell lines. For in vivo experiments a minimum of 6 tumours/group was used, with further validation obtained from additional genetic constructs and/or cell lines. The sample sizes were determined based on previous experience from similar experiments. |
| Data exclusions | No data were excluded from the analyses. |
| Replication | The reproducibility of the experimental findings were verified by performing additional independent experiments or by having several technical replicates (as described in the figure legends). Furthermore, independent experiments were also conducted in several cell lines to ensure the findings were reproducible as well as cross-checking the findings with clinical data. All attempts at replication were confirmed to be successful. |
| Randomization | Samples were not randomized. For mouse experiments mice were age and sex matched to reduce variability between groups and all experimental groups were analyzed in parallel. For in vitro experiments, the conditions were controlled as well as possible to reduce unintended variability and the need for randomization. |
| Blinding | The experimental groups were not blinded. For most in vitro experiments blinding is not feasible, but whenever possible, data collection was performed automatically and/or with internal controls, reducing the need for blinding. Similarly, for mouse experiments tumour growth was tracked by caliper measurements with additional validation by bioluminescence imaging, reducing the need for blinding. |

# Reporting for specific materials, systems and methods

We require information from authors about some types of materials, experimental systems and methods used in many studies. Here, indicate whether each material, system or method listed is relevant to your study. If you are not sure if a list item applies to your research, read the appropriate section before selecting a response.

## Materials & experimental systems

| n/a | Involved in the study |
|---|---|
| ☐ | ☒ Antibodies |
| ☐ | ☒ Eukaryotic cell lines |
| ☒ | ☐ Palaeontology and archaeology |
| ☐ | ☒ Animals and other organisms |
| ☐ | ☒ Human research participants |
| ☒ | ☐ Clinical data |
| ☒ | ☐ Dual use research of concern |

## Methods

| n/a | Involved in the study |
|---|---|
| ☐ | ☒ ChIP-seq |
| ☐ | ☒ Flow cytometry |
| ☒ | ☐ MRI-based neuroimaging |

## Antibodies

**Antibodies used**

For western blotting: PAX8 (Santa Cruz Biotech, sc-81353 1:250), HNF1B (Human Protein Atlas, HPA002083 1:5000), MYC (Abcam, ab32072, 1:1000), HIF2A (Novus Biologicals, NB100-122, 1:1000), VHL (BD Pharmingen, 565183, 1:1000), CCND1 (Abcam, ab134175, 1:1000), HIF1A (R&D systems, MAB1536, 1:500), and B-actin (Sigma-Aldrich, A1978, 1:20000) antibodies. Secondary antibodies were polyclonal goat anti-mouse IgG/HRP (Dako, P0447, 1:10000) and polyclonal goat anti-rabbit IgG/HRP conjugated (Dako, P0448, 1:5000).

For IHC: Human Vimentin (Cell Signaling Technology; cat. 5741, 1:100), HIF2A (Santa Cruz sc-46691,1:200), PAX8 primary antibody clone MRQ-50 (363M-16, Cell Marque) and the HNF1B primary antibody (Human Protein Atlas, HPA002083)

For co-IP: PAX8 (ProteinTech 10336-1-AP), HIF2A (Abcam, ab199), rabbit polyclonal IgG (Abcam, ab27478) and HIF1B (Santa Cruz Biotech, H-10, sc-55526, 1:200). Secondary antibodies used were anti-mouse IgG/HRP for IP (Abcam, ab131368, 1:5000) and VeriBlot for IP detection (HRP) (Abcam, ab131366, 1:5000)

**Validation**

1. Polyclonal goat anti-mouse IgG/HRP (Dako, P0447, 1:10000) - antibody technical datasheet [https://www.agilent.com/cs/ library/ packageinsert/public/104706002.PDF]
2. Polyclonal goat anti-rabbit IgG/HRP conjugated (Dako, P0448, 1:5000) - antibody technical datasheet [https:// www.agilent.com/ cs/library/packageinsert/public/104707002.PDF]
3. B-actin (Sigma-Aldrich, A1978, 1:20000) - antibody technical datasheet [https://www.sigmaaldrich.com/content/dam/sigma-aldrich/docs/Sigma/Datasheet/6/a1978dat.pdf]
4. HIF2A (Novus Biologicals, NB100-122, 1:1000) - antibody technical datasheet [https://www.novusbio.com/PDFs/ NB100-122.pdf].
5. VHL (BD Biosciences, 564183, 1:1000) - antibody technical datasheet [http://www.bdbiosciences.com/ds/pm/tds/564183.pdf]
6. PAX8 (Santa Cruz Biotech, sc-81353 1:250)-antibody technical datasheet [https://datasheets.scbt.com/sc-81353.pdf]
7. HNF1B (Human Protein Atlas, HPA002083 1:5000)-antibody technical datasheet [https://www.atlasantibodies.com/api/ print_datasheet/HPA002083.pdf]
8. MYC (Abcam, ab32072, 1:1000)-antibody technical datasheet [https://www.abcam.com/c-myc-antibody-y69-bsa-and-azide-free-ab168727.pdf]
9. HIF1B (Santa Cruz Biotech, H-10, 1:200)-antibody technical datasheet [https://datasheets.scbt.com/sc-55526.pdf]
10. HIF1A (R&D systems, MAB1536, 1:500), -antibody technical datasheet [https://resources.rndsystems.com/pdfs/datasheets/ mab1536.pdf?v=20211102&_ga=2.202325457.1710034994.1635864392-808840216.1615813694]
11. CCND1 (Abcam, ab134175, 1:1000) – antibody technical datasheet [https://www.abcam.com/cyclin-d1-antibody-epr2241-c-terminal-ab134175.pdf]
12. Human Vimentin (Cell Signaling Technology; cat. 5741, 1:100) - antibody technical datasheet [https://www.cellsignal.com/ datasheet.jsp?productId=5741&images=1]
13. HIF2A (Santa Cruz sc-46691,1:200) – antibody technical datasheet [https://datasheets.scbt.com/sc-46691.pdf]
14. PAX8 primary antibody clone MRQ-50 (363M-16, Cell Marque) - antibody technical datasheet [https://www.cellmarque.com/ antibodies/CM/2127/PAX-8_MRQ-50]
15. HIF2A (Abcam, ab199) – antibody technical datasheet [https://www.abcam.com/hif-2-alpha-antibody-ab199.pdf]
16. rabbit polyclonal IgG (Abcam, ab27478) - antibody technical datasheet [https://www.abcam.com/rabbit-igg-polyclonal-isotype-control-ab27478.pdf]
17. anti-mouse IgG/HRP for IP (Abcam, ab131368, 1:5000) - antibody technical datasheet [https://www.abcam.com/veriblot-for-ip-detection-reagent-hrp-ab131366.pdf]
18. VeriBlot for IP detection (HRP) (Abcam, ab131366, 1:5000) - antibody technical datasheet [https://www.abcam.com/mouse-igg-for-ip-hrp-ab131368.html]

# Eukaryotic cell lines

Policy information about cell lines

Cell line source(s)

The UOK101 cell line was obtained from M. Linehan (the UOB Tumor Cell Line Repository, National Cancer Institute, Bethesda, MD). The HK2 cell line was obtained from C. Frezza (MRC Cancer Unit, Cambridge, UK). ACHN and CAKI-2 were obtained from E. Maher (Department of Medical Genetics, University of Cambridge, UK). All other human cancer cell lines and the HEK293T cells were obtained from J. Massagué (MSKCC, New York, USA). 786-M1A and OS-LM1 are the respective metastatic derivatives of 786-O and OS-RC2 cells and have been previously described (Vanharanta et al., Nat Med. (2013) PMID: 23223005). 2806-LM1A is a metastatic derivative of 786-O and it does not carry a luciferase reporter gene. C-M1A HIF2A-/- is a single cell- derived HIF2A-/- clone from 786-M1A cells generated by CRISPR-Cas9 mediated knockout of HIF2A.

Authentication

Cell lines were authenticated by short tandem repeat profiling.

Mycoplasma contamination

Cell lines were confirmed to be mycoplasma negative using the MycoAlertTM Mycoplasma Detection Kit (Lonza, LT07-318) or by qRT-PCR (PhoenixDx® Mycoplasma Mix).

Commonly misidentified lines
(See ICLAC register)

At the time of the study, none of the cell lines used in this study were listed in the database of commonly misidentified cell lines maintained by ICLAC.

# Animals and other organisms

Policy information about studies involving animals; ARRIVE guidelines recommended for reporting animal research

Laboratory animals

Athymic nude mice, female, 5-8 weeks old (Charles River Laboratories 490 (Homozygous)). NSG mice, male, 5-7 weeks old (Charles River Laboratories, strain: NOD.Cg-Prkdcscid Il2rgtm1WjI/SzJ). The housing conditions were as follows: 12/12h dark/light cycle, humidity 45-65%, temperature 20-24°C.

Wild animals

This study did not involve wild animals.

Field-collected samples

This study did not involve field-collected samples.

Ethics oversight

Home Office (UK) and the University of Cambridge Animal Welfare and Ethical Review Body.

Note that full information on the approval of the study protocol must also be provided in the manuscript.

# Human research participants

Policy information about studies involving human research participants

Population characteristics

A tissue microarray with 427 ccRCC patients (170 females, 257 males, age range 28-90 with a median of 65) tumours represented on it from the full range of stages of disease was used. It was created as previously published (Laird A, O'Mahony FC, Nanda J, Riddick AC, O'Donnell M, Harrison DJ, et al. Differential expression of prognostic proteomic markers in primary tumour, venous tumour thrombus and metastatic renal cell cancer tissue and correlation with patient outcome. PLoS One. 2013;8(4):e60483. doi:10.1371/journal.pone.0060483.) Normal human kidney tissue used for organoid derivation was sampled from a 75-year-old male.

Recruitment

Patients who had primary renal cell carcinoma at the time of surgery or at a later date were identified from a prospectively compiled database. Formalin fixed paraffin embedded (FFPE) tumour samples were identified from 427 of these patients who underwent radical nephrectomy between 1983 and 2010, in the Department of Urology, Edinburgh. Where possible, written informed consent was gained for use of tissue surplus to diagnostic requirement and linked anonymised patient data. Normal human kidney tissue from a nephrectomy specimen was sampled with informed consent.

Ethics oversight

Ethical approval to use these archived tissues was granted by the Lothian Regional Ethics Committee (08/S1101/41 and 10/S1402/33). Normal human kidney tissue used for organoid derivation was collected under an ethical approval by the East of England - Cambridge Central Research Ethics Committee (19/EE/0161).

Note that full information on the approval of the study protocol must also be provided in the manuscript.

# ChIP-seq

## Data deposition

☒ Confirm that both raw and final processed data have been deposited in a public database such as GEO.

☒ Confirm that you have deposited or provided access to graph files (e.g. BED files) for the called peaks.

Data access links
*May remain private before publication.*

https://www.ncbi.nlm.nih.gov/geo/query/acc.cgi?acc=GSE163487

Files in database submission

```
SLX-14864.A002.HVL5LBBXX.s_5.r_1_trim_unmapped_merged.bl.bw
SLX-14864.A006.HVL5LBBXX.s_5.r_1_trim_unmapped_merged.bl.bw
SLX-14864.A007.HVL5LBBXX.s_5.r_1_trim_unmapped_merged.bl.bw
SLX-16309.A002.HWMMGBBXX.s_3.r_1_trim_unmapped.bl.bw
SLX-16309.A014.HWMMGBBXX.s_3.r_1_trim_unmapped.bl.bw
SLX-16309.A016.HWMMGBBXX.s_3.r_1_trim_unmapped.bl.bw
SLX-16309.A019.HWMMGBBXX.s_3.r_1_trim_unmapped.bl.bw
SLX-14864.A013.HVL5LBBXX.s_5.r_1_trim_unmapped_merged.bl.bw
SLX-14864.A014.HVL5LBBXX.s_5.r_1_trim_unmapped_merged.bl.bw
SLX-14864.A015.HVL5LBBXX.s_5.r_1_trim_unmapped_merged.bl.bw
SLX-16309.A004.HWMMGBBXX.s_3.r_1_trim_unmapped.bl.bw
SLX-16309.A007.HWMMGBBXX.s_3.r_1_trim_unmapped.bl.bw
SLX-16309.A013.HWMMGBBXX.s_3.r_1_trim_unmapped.bl.bw
SLX-16301.A002.HWCFNBBXX.s_8.r_1_trim_unmapped.bl.bw
SLX-16301.A004.HWCFNBBXX.s_8.r_1_trim_unmapped.bl.bw
SLX-16301.A005.HWCFNBBXX.s_8.r_1_trim_unmapped.bl.bw
SLX-16301.A013.HWCFNBBXX.s_8.r_1_trim_unmapped.bl.bw
SLX-16301.A014.HWCFNBBXX.s_8.r_1_trim_unmapped.bl.bw
SLX-16301.A015.HWCFNBBXX.s_8.r_1_trim_unmapped.bl.bw
SLX-16301.A006.HWCFNBBXX.s_8.r_1_trim_unmapped.bl.bw
SLX-16301.A007.HWCFNBBXX.s_8.r_1_trim_unmapped.bl.bw
SLX-16302.A007.H23WKBBXY.s_3.r_1_trim_unmapped.bl.bw
SLX-16301.A016.HWCFNBBXX.s_8.r_1_trim_unmapped.bl.bw
SLX-16301.A018.HWCFNBBXX.s_8.r_1_trim_unmapped.bl.bw
SLX-16301.A019.HWCFNBBXX.s_8.r_1_trim_unmapped.bl.bw
SLX-16302.A012.H23WKBBXY.s_3.r_1_trim_unmapped.bl.bw
SLX-16310.A002.H23WKBBXY.s_5.r_1_trim_unmapped.bl.bw
SLX-16310.A004.H23WKBBXY.s_5.r_1_trim_unmapped.bl.bw
SLX-16310.A005.H23WKBBXY.s_5.r_1_trim_unmapped.bl.bw
SLX-16310.A012.H23WKBBXY.s_5.r_1_trim_unmapped.bl.bw
SLX-16310.A013.H23WKBBXY.s_5.r_1_trim_unmapped.bl.bw
SLX-16310.A014.H23WKBBXY.s_5.r_1_trim_unmapped.bl.bw
SLX-16310.A006.H23WKBBXY.s_5.r_1_trim_unmapped.bl.bw
SLX-16310.A007.H23WKBBXY.s_5.r_1_trim_unmapped.bl.bw
SLX-16310.A019.H23WKBBXY.s_5.r_1_trim_unmapped.bl.bw
SLX-16310.A015.H23WKBBXY.s_5.r_1_trim_unmapped.bl.bw
SLX-16310.A016.H23WKBBXY.s_5.r_1_trim_unmapped.bl.bw
SLX-16310.A018.H23WKBBXY.s_5.r_1_trim_unmapped.bl.bw
SLX-14864.A002.HVL5LBBXX.s_5.r_1.fq.gz
SLX-14864.A006.HVL5LBBXX.s_5.r_1.fq.gz
SLX-14864.A007.HVL5LBBXX.s_5.r_1.fq.gz
SLX-16309.A002.HWMMGBBXX.s_3.r_1.fq.gz
SLX-16309.A014.HWMMGBBXX.s_3.r_1.fq.gz
SLX-16309.A016.HWMMGBBXX.s_3.r_1.fq.gz
SLX-16309.A019.HWMMGBBXX.s_3.r_1.fq.gz
SLX-14864.A013.HVL5LBBXX.s_5.r_1.fq.gz
SLX-14864.A014.HVL5LBBXX.s_5.r_1.fq.gz
SLX-14864.A015.HVL5LBBXX.s_5.r_1.fq.gz
SLX-16309.A004.HWMMGBBXX.s_3.r_1.fq.gz
SLX-16309.A007.HWMMGBBXX.s_3.r_1.fq.gz
SLX-16309.A013.HWMMGBBXX.s_3.r_1.fq.gz
SLX-16301.A002.HWCFNBBXX.s_8.r_1.fq.gz
SLX-16301.A004.HWCFNBBXX.s_8.r_1.fq.gz
SLX-16301.A005.HWCFNBBXX.s_8.r_1.fq.gz
SLX-16301.A013.HWCFNBBXX.s_8.r_1.fq.gz
SLX-16301.A014.HWCFNBBXX.s_8.r_1.fq.gz
SLX-16301.A015.HWCFNBBXX.s_8.r_1.fq.gz
SLX-16301.A006.HWCFNBBXX.s_8.r_1.fq.gz
SLX-16301.A007.HWCFNBBXX.s_8.r_1.fq.gz
SLX-16302.A007.H23WKBBXY.s_3.r_1.fq.gz
SLX-16301.A016.HWCFNBBXX.s_8.r_1.fq.gz
SLX-16301.A018.HWCFNBBXX.s_8.r_1.fq.gz
SLX-16301.A019.HWCFNBBXX.s_8.r_1.fq.gz
SLX-16302.A012.H23WKBBXY.s_3.r_1.fq.gz
SLX-16310.A002.H23WKBBXY.s_5.r_1.fq.gz
SLX-16310.A004.H23WKBBXY.s_5.r_1.fq.gz
SLX-16310.A005.H23WKBBXY.s_5.r_1.fq.gz
SLX-16310.A012.H23WKBBXY.s_5.r_1.fq.gz
SLX-16310.A013.H23WKBBXY.s_5.r_1.fq.gz
SLX-16310.A014.H23WKBBXY.s_5.r_1.fq.gz
SLX-16310.A006.H23WKBBXY.s_5.r_1.fq.gz
SLX-16310.A007.H23WKBBXY.s_5.r_1.fq.gz
SLX-16310.A019.H23WKBBXY.s_5.r_1.fq.gz
SLX-16310.A015.H23WKBBXY.s_5.r_1.fq.gz
SLX-16310.A016.H23WKBBXY.s_5.r_1.fq.gz
```

April 2020

SLX-16310.A018.H23WKBBXY.s_5.r_1.fq.gz
SLX-14864.A002.HFH52BBXY.s_1.r_1.fq.gz
SLX-14864.A006.HFH52BBXY.s_1.r_1.fq.gz
SLX-14864.A007.HFH52BBXY.s_1.r_1.fq.gz
SLX-14864.A013.HFH52BBXY.s_1.r_1.fq.gz
SLX-14864.A014.HFH52BBXY.s_1.r_1.fq.gz
SLX-14864.A015.HFH52BBXY.s_1.r_1.fq.gz

Genome browser session
(e.g. UCSC)

NA

## Methodology

### Replicates

HIF2A M1A ChIP: 3 biological replicates
HIF2A M1A input: 4 biological replicates
HIF2A LM1B ChIP: 3 biological replicates
HIF2A LM1B input: 3 biological replicates

PAX8 M1A ChIP: 3 biological replicates
PAX8 M1A input: 3 biological replicates
PAX8 LM1B ChIP: 3 biological replicates
PAX8 LM1B ChIP: 4 biological replicates

HNF1B M1A ChIP: 3 biological replicates
HNF1B M1A input: 3 biological replicates
HNF1B LM1B ChIP: 3 biological replicates
HNF1B LM1A input: 3 biological replicates

### Sequencing depth

All the samples are single-end with 50bp read length.

The total number of reads (after excluding genes mapping to mouse genome) and the number of reads after filtering out reads with mapping quality < 20, reads mapping to blacklisted regions and reads mapping to regions other than chr1 to 22, X and Y is reported for each sample.

SLX-14864.A002 Total: 24278880; After filtering: 22016100
SLX-14864.A006 Total: 16141095; After filtering: 14547194
SLX-14864.A007 Total: 17380560; After filtering: 15697922
SLX-16309.A002 Total: 10940226; After filtering: 9915172
SLX-16309.A014 Total: 14594845; After filtering: 13162705
SLX-16309.A016 Total: 14776625; After filtering: 13337506
SLX-16309.A019 Total: 10291446; After filtering: 9284789
SLX-14864.A013 Total: 13906559; After filtering: 12505916
SLX-14864.A014 Total: 87053321; After filtering: 79063102
SLX-14864.A015 Total: 18828982; After filtering: 17082291
SLX-16309.A004 Total: 51148025; After filtering: 46215659
SLX-16309.A007 Total: 16635642; After filtering: 14945684
SLX-16309.A013 Total: 30450304; After filtering: 27511220
SLX-16301.A002 Total: 11127778; After filtering: 10111323
SLX-16301.A004 Total: 27172651; After filtering: 24631106
SLX-16301.A005 Total: 20029312; After filtering: 18224040
SLX-16301.A013 Total: 22492314; After filtering: 20283038
SLX-16301.A014 Total: 39913582; After filtering: 35968614
SLX-16301.A015 Total: 16962961; After filtering: 15292985
SLX-16301.A006 Total: 21505330; After filtering: 19517946
SLX-16301.A007 Total: 24900285; After filtering: 22550039
SLX-16302.A007 Total: 10405114; After filtering: 9444550
SLX-16301.A016 Total: 29484377; After filtering: 26557099
SLX-16301.A018 Total: 29616225; After filtering: 26685369
SLX-16301.A019 Total: 22889998; After filtering: 20535788
SLX-16302.A012 Total: 4513129; After filtering: 4075220
SLX-16310.A002 Total: 21338846; After filtering: 19156581
SLX-16310.A004 Total: 5656074; After filtering: 5189524
SLX-16310.A005 Total: 18598964; After filtering: 17006151
SLX-16310.A012 Total: 26730720; After filtering: 24030433
SLX-16310.A013 Total: 29122467; After filtering: 26298413
SLX-16310.A014 Total: 36061004; After filtering: 32708244
SLX-16310.A006 Total: 24232763; After filtering: 21942289
SLX-16310.A007 Total: 10971401; After filtering: 9941291
SLX-16310.A019 Total: 13340589; After filtering: 12128048
SLX-16310.A015 Total: 28101167; After filtering: 25312242
SLX-16310.A016 Total: 15663471; After filtering: 14116857
SLX-16310.A018 Total: 9664508; After filtering: 8725758

### Antibodies

PAX8 (ProteinTech 10336-1-AP), HNF1B (Human Protein Atlas, HPA002083), HIF2A (Novus Biologicals NB100-122) and rabbit polyclonal IgG (Abcam, ab27478)

| Peak calling parameters | Mapping<br>bwa mem -M hg38.fa chip.fq > chip.sam<br>samtools view -S -b -h -T hg38.fa chip.sam | samtools sort -O bam -T chip.tmp -o chip.bam;<br>samtools index chip.bam<br><br>Peak calling<br>macs2 callpeak --bdg -t chip.bam -c input.bam -f BAM -g 2913022398 |
|---|---|
| Data quality | Number of peaks with fold change more than 5 and FDR 0.05<br><br>HIF2A_M1A_1: 1541<br>HIF2A_M1A_2: 276<br>HIF2A_M1A_3: 649<br>HIF2A_LM1B_1: 541<br>HIF2A_LM1B_2: 1766<br>HIF2A_LM1B_3: 748<br><br>HNF1B_M1A_1: 2522<br>HNF1B_M1A_2: 7511<br>HNF1B_M1A_3: 5360<br>HNF1B_LM1B_1: 3210<br>HNF1B_LM1B_2: 6435<br>HNF1B_LM1B_3: 5310<br><br>PAX8_M1A_1: 3116<br>PAX8_M1A_2: 1044<br>PAX8_M1A_3: 6930<br>PAX8_LM1B_1: 6665<br>PAX8_LM1B_2: 4045<br>PAX8_LM1B_3: 16225 |
| Software | FastQC (version 0.11.7)<br>Cutadapt(version 1.10.0)<br>Bowtie 2 (version 2.3.4.3)<br>MACS2 (version 2.2.7.1)<br>Samtools (version 1.2)<br>BEDOPS (version 2.4.37)<br>deepTools (version 3.5.0)<br>bedtools (version 2.27.1) |

# Flow Cytometry

## Plots

Confirm that:

☒ The axis labels state the marker and fluorochrome used (e.g. CD4-FITC).

☒ The axis scales are clearly visible. Include numbers along axes only for bottom left plot of group (a 'group' is an analysis of identical markers).

☒ All plots are contour plots with outliers or pseudocolor plots.

☒ A numerical value for number of cells or percentage (with statistics) is provided.

## Methodology

| Sample preparation | In brief, the cells were either lentivirally transduced with mCherry, GFP or BFP sgRNA/shRNA expression vectors, mixed and plated onto multi-well plates in triplicate. For flow cytometry analyses at the different time points, the mixed cells populations were trypsinized and directly analyzed on the instrument specified below.<br>Source of cells : human renal cancer cell lines |
|---|---|
| Instrument | LSR Fortessa (BD Biosciences) |
| Software | BD FACSDiva software (8.0.1) |
| Cell population abundance | The abundance of relevant cell population was determined based on the specific fluorescent marker expressed by the cells. The control cell population increased over times and outgrew the knockout cell population that resulted in reduced percentage/ abundance of the knockout cell population. |

Gating strategy

Competitive proliferation assay gating strategy
1. FSC-A / SSC-A : to select for live cell population
2. FSC-W / SSC-A : to select for single cells
3. mCherry (561nm/610nm), BFP (383nm/445nm) or GFP (488nm/510nm) : to discriminate between the cell populations

☒ Tick this box to confirm that a figure exemplifying the gating strategy is provided in the Supplementary Information.

