## [Peer Review File · Nature]

Manuscript Title: Renal lineage factor PAX8 controls oncogenic signalling in kidney cancer

Redactions – unpublished data: Parts of this Peer Review File have been redacted as indicated to maintain the confidentiality of unpublished data.

Reviewer Comments & Author Rebuttals

Reviewer Reports on the Initial Version:

Referee #1 (Remarks to the Author):

The key findings of this manuscript by Patel et al provides an intriguing linkage between the known lineage factor Pax8 transcription factor and VHL driven ccRCC. They show that Pax8 is necessary to drive proliferation, which has been demonstrated before (Bleu et al 2019) and promotes expression of HNF1b (also shown previously (Buisson et al 2015)). The intriguing part comes from the connection of Pax8/HNF1b to be necessary to drive myc expression, and that this complex further mediates CCND1 expression by co-transcription of Pax8 and HIF2 on an open chromatin module previously identified as a risk allele for ccRCC. Here a variant snp reveals a dependency on both transcription factors. The concept of Pax8 being essential to allow HIF2 to function as a tumor promoter is intriguing and could provide an explanation for the kidney specificity of VHL mutation-driven tumors.

There are some issues that raise concern over the extent of the conceptual advance this puts forward.

1. the bulk of the findings - of Pax8 dependency for growth (Bleu 2018), of Myc dependency (Tang 2009), of Pax8 driving HNF1b (Buisson 2015), ATACseq in KIRC and KIRP tumors demonstrating more open chromatin (Corces 2017, referenced), have been reported previously, such that the truly new concept is a fairly small fraction of the whole story. This most compelling portion, regarding the interaction of Pax9 and HIF2 with the SNP is treated fairly superficially for such a pivotal finding.
2. The cell lines used were selected for aggressive metastatic behavior, thus there may be evolved features, particularly with regard to myc and CCND1 signaling, that are not solely linked to VHL loss and HIF2 expression. At some points the work switches back to the parent cell line 786-0. The rationale for these selections should be clear.
3. The HIF2 driving feature is not developed sufficiently to understand how or if HIF1 would engage this process. Specific studies to add HIF2 would address this.
4. Additional information regarding the prevalence of the SNP, any skew in type of kidney cancer based on SNP, or other relevant human data would be valuable.
5. The authors should also consider that HNF1b is implicated as an important factor in chromophobe RCC (KICH), which is considered to arise from the distal tubule (Sun 2017).
6. Additional discussion regarding the potential mechanisms by which VHL loss does drive tumor

formation in the hemangioblastomas, pheochromocytomas, and pancreatic NET tumors in the setting of VHL disease would be valuable.

7. Prior work by this group has endorsed co-option of an NF κ B transcriptional hub with HIF2 as a mechanism of metastasis. How does this finding integrate with that transition?

Referee #2 (Remarks to the Author):

Patel and colleagues investigate the rationale for tissue-specific tumorigenesis in the context of sporadic clear cell renal cell carcinoma (ccRCC). Through a transcription factor (TF) screen in two RCC cell lines they uncover PAX8-HNF1B-MYC axis as a ccRCC-specific mechanism for acquiring the proliferative phenotype. Through interrogation of the protein interactome and ChIP-seq peak overlaps, and functional study of DNA motifs they demonstrate that PAX8 cooperates with HIF2A to activate downstream CCND1 expression through a distal enhancer – rs794863, a ccRCC risk SNP, illustrating potential molecular basis of this common ccRCC risk variant.

Major comments

1. Regarding the cell lines in which the screen was performed. I was not able to find information on OS-LM1, while 786-M1A1 is a human line that was made metastatic by injection in a NOD-SCID mouse and derived from consequent lung metastases which likely would have led to selection of a specific phenotype. An explanation for the choice of cell lines and information on key genotypes would be appreciated given this can impact the generalizability of the study findings in a cancer defined by a variety of genetic alterations.
2. There is a possibility that PAX8 and HNF1B maintain chromatin activation states and fitness in normal renal cell lineage rather than renal cancer, and the authors should implement a normal kidney control such as HK2 or PTEC cell lines for the knock out and downstream experiments (besides HeLa).
3. The authors have focused on the proliferative phenotype with cell competition assay and tumor xenograft growth measurements, but should evaluate other features such as invasion/metastases formation and tumour histology.
4. The authors clearly demonstrate PAX8-HNF1B-MYC axis as a mechanism underlying the proliferative phenotype, however it would be interesting to know whether this is a consequence of upstream transcriptional motif binding and activation.
5. The interaction between PAX8 and HIF2A was inferred by RIME interactome and ChIP-seq, but direct interaction should be investigated by CoIP or pulldown assay/ Furthermore, KIRC-56592 is 220kb away from CCND1, suggesting additional mediators contribute to how this distal enhancer works.

6. It would be useful to explore the findings in a clinical context such as survival data in TCGA (E.G. PAX8 expression).

7. Line 179-181, what about rescue experiments on HNF1B depletion cells?

8. Line 198-203, could the authors rescue MYC expression in HNF1B-depleted cells to further validate the PAX8-HNF1B-MYC axis.

9. Both ccRCC and papillary RCC are dependent on PAX8 (Fig1), while KIRC-56592 is only essential in ccRCC (Fig3h), can the authors reconcile this?

10. Give the tissue specificity of inherited VHL disease it would be of interest for the authors to comment on their findings in the context of this cancer syndrome.

Minor comments

1. Fig1a, there is also an outlier in the upright quadrant, why was this sensitive TF omitted?

2. Line 61, ref 9, relates not to HIF2A but HIF1A.

3. Extended Fig1d, the knock out efficiency appears low, especially in 786-M1A and OS-LM1 cells with sgRNA No.4, but in spite of this while the tumor formation is efficient; suggest repeating the western blot

4. In extended Fig5d, could the authors annotate PAX8 and HNF1B in the two volcano plots and reference the figure in lines 166-167.

5. In Fig2a,b, the sgRNAs used in mRNA validation and protein validation appear inconsistent.

6. The statement that MYC is not “commonly amplified in ccRCC” ignores the frequent gain of 8q which contains MYC locus.

7. Line 246-247, could the authors add references or results to support “VHL restoration and consequent inhibition of HIF2A does not have a strong effect on the proliferation of ccRCC cells in vitro”.

8. HIF2A is intermittently referred to EPAS1 (e.g. Extended Fig10f right panel), it would be better to be consistent.

9. Line 491-495, could the author describe in more detail the calculation of dependency score.

Referee #3 (Remarks to the Author):

Patel et al present a CRISPR-screen to identify key factors in two ccRCC cell lines, and identify the two previously known lineage factors (in the cell of origin of ccRCC), PAX8 and HNF1B. It was shown previously that these factors are needed to transdifferentiate fibroblasts into renal tubular epithelial cells (Kaminski et al. NCB 2016). Knock-down of these factors causes cells to arrest in cell cycle, and to lose their lineage transcriptional program (i.e., the lineage gene regulatory network collapses). Next they show by ATAC-seq that the chromatin accessibility of lineage specific regulatory regions in the genome depend on these factors (this is expected, given the previous point). Upon PAX8/HNF1B knock-down, as we see cell cycle arrest, MYC targets are down-regulated. For the RIME (protein complexes on chromatin) and ChIP-seq experiments, the authors target HIF2A, to investigate

whether HIF2A co-binds to the PAX8/HNF1B enhancers. With the protein interaction data they show that HNF1B and PAX8 are in the HIF2A interactome. With the ChIP-seq data, they show that a subset of HIF2A and PAX8 binding sites overlap. Next, 175 HIF2A binding sites are targeted by dCas-KRAB in a pooled assay, using xenotransplantation, and identified a region upstream of CCND1 as co-bound by PAX8 and HIF2A, and repressing this enhancer leads to reduced tumour formation (due to lower CCND1 expression?). Finally, the authors describe a SNP in this enhancer that affects the PAX8 motif (risk allele has a stronger PAX8 site, compared to the rarer protective allele, that has a weaker PAX8 site). This study integrates multiple assays and circumstantial evidences showing how PAX8 and HNF1B maintain ccRCC lineage identity while promoting cancer growth (e.g., via MYC); how the lineage factor PAX8 facilitates HIF2A-dependent up-regulation of CCND1; and that rs7948643 affects ccRCC risk by changing that particular PAX8 binding site. Given that only few cis-regulatory variants are functionally/mechanistically understood, this study is timely and may fuel new studies to decipher the role of other cis-regulatory SNPs in cancer.

Comments

1. The key finding is that rs7948643 affects PAX8 binding, this in turn affects the regulation of CCND1 by HIF2A, thereby connecting the regulatory program of the cell type of origin (PAX8) with cancer-related TFs (HIF2A). That rs7948643 changes PAX8 binding is only shown by an EMSA assay, while this is a central element in this study. Is it possible to use the PAX8 ChIP-seq data to verify whether binding is biased to the risk allele? In a heterozygous context, both PAX8 binding and chromatin accessibility (ATAC-seq signal) are expected to show allelic imbalance (only the allele with the common allele, and the strong PAX8 binding site, should be bound by PAX8. If phasing is possible, then also the expression of CCND1 at that same allele should have higher CCND1 expression. If phasing is not possible, then at least the CCND1 expression is expected to be allele-specific?
2. Does HIF2A activation, upon PAX8 depletion, no longer induce CCND1 expression?
3. It is important to show the ChIP-seq (both HIF2A and PAX8) and ATAC-seq signals as bigWig tracks in the main figures for additional loci, such as PAX8 itself, and the rs7948643 locus all the way to CCND1.
4. The “hypothesis” that lineage transcription factors are essential determinants of oncogenic phenotypes (p3) is a bit over-stated. Lineage TFs are maintained in most cancer types, this is more the rule than the exception. Androgen receptor, estrogen receptor, SOX10/MITF in melanoma, many examples in leukemia, etc. Perhaps the authors can provide more nuance, discussing more examples, how this is a common feature in cancer, rather than “we are testing this novel hypothesis”. The novel question can be phrased more precisely? (e.g., how lineage TFs co-control cancer-related genes?). Also, knock-down of these lineage factors in other cancer types (e.g., depletion of SOX10 in melanoma cells) has been shown repeatedly to cause cell cycle arrest, and induction of EMT-like genes, thus presenting a highly similar process to the current study (depletion of the lineage TFs causes collapse of the lineage chromatin state, cell cycle arrest, and induction of a stress-like state). For this particular aspect, the current study is not that novel, and some nuance or context can be provided (more novelty lies in the characterisation of the SNP, and linking that with the lineage program). Perhaps the up-regulated genes can even be compared with the up-regulated genes upon SOX10 depletion in melanoma cells (or similar data sets for other cancer types with clear lineage TFs).
5. Related to the previous point - would PAX8 also control CCND1 and MYC during development or

maintenance of renal tubular epithelial cells, and does the expression in ccRCC cells constitute an up-regulation rather than a de novo activation?

6. Also related: line 121 concludes that PAX8 and HNF1B maintain “cancer-specific” chromatin accessibility patterns; I would argue that they maintain lineage specific chromatin accessibility, as shown by the data presented. Otherwise, more comparisons are needed to discriminate normal accessibility in renal cells, versus cancer-specific regions. The statement on line 139, that PAX8 and HNF1B regulate proximal tubule cells lineage (and that their induction can create such cells) should be moved to the very beginning, as this is essential information to interpret the first experiments/data.

7. PAX8 + HNF1B are argued to maintain a ccRCC chromatin state. However, it is more likely that they maintain a renal cell chromatin state - as the cancer state strongly reflects the cell type of origin (in many cancer types). How different is the chromatin state between normal renal tubular epithelial cells and ccRCC cells? Are the PAX8+HNF1B binding sites (accessible regions) the same or not? The recent Meuleman et al. DHS atlas may come to rescue here.

8. Regions with increased accessibility after PAX8/HNF1B depletion show “motifs for other factors” : which motifs and for which factors? AP-1?

9. MYC target genes are strongly inhibited in the PAX8/HNF1B depletion - this is normal, since the cells go in arrest.

10. Line 207 concludes that PAX8 and HNF1B were now identified as drivers of ccRCC formation - but this is not shown and seems like an incorrect conclusion. These are the drivers of the cell type of origin, and maintain their expression in the cancer cell.

11. 205 HIF2A dependent transcripts are found, with 175 HIF2A binding sites in 500kb. Can the authors randomly select 205 expressed genes, and calculate how many binding sites are found, and repeat that 100 times? This would give an empirical p-value. It seems quite likely to find this number of binding sites.

12. Line 272: ‘several’ HIF2A bound enhancers were depleted in the dCas-KRAB assay: how many exactly, and is this significant? Is there a relationship between depletion and HIF2A sites or PAX8 sites?

13. Does PAX8 knock-down affect CCND1 up-regulation by HIF2A, in an allele-specific manner (when the heterozygous line for rs7948643 is used)?

14. Is CCND1 an endogenous target of PAX8 during development (most parsimonious explanation), and HIF2A hijacks this interaction to cause increased CCND1 expression? Also, CCND1 might also be high in ccRCC that has no HIF2A activation. Is CCND1 overall higher in tcga samples with HIF2A activation? Or simply, are HIF2A and CCND1 correlated across tcga samples (and is that stronger than pax8-ccnd1 correlation)?

Referee #4 (Remarks to the Author):

In the manuscript “Lineage factor-dependent signalling by kidney cancer-associated genetic alterations” Patel et al. identify PAX8 and HNF1b as lineage specific transcription factors that guide oncogenic signaling in clear cell renal cancer. They use a CRISPR Cas9 library to screen for transcription factors relevant for renal tumor growth in xenograft models. The two most kidney specific factors PAX8 and HNF1b are examined further by knock-out cell models and genome-wide

screens using ATAC-seq. Through this and by RNAseq of PAX8 and HNF1b depleted RCC cells the authors show that sites positively regulated by PAX8/HNF1b are enriched in kidney cancer relevant open chromatin regions. They follow up HNF1b and MYC as downstream targets of PAX8. Having observed a hypoxia signature in PAX8 depleted cells, in a second set of experiments the authors test for regulatory elements important for HIF2A signaling. RNA-seq from RCC xenografts with and without HIF2A and HIF2A ChIPseq from xenografts define sites regulated by HIF2A. By this the authors retrieve a known enhancer of CCND1 expression, which is harboring the most significant RCC GWAS hit so far and has been linked to regulation of CCND1 expression by various groups before. From PAX8 and HIF2A RIME analysis and ChIPseq data it appears that both signaling pathways seem to overlap. The authors can also define a hypoxia signature in the PAX8 depleted RCC cells suggesting that both pathways have common targets. This is evident at the CCND1 enhancer at which PAX8 DNA interaction might be dependent on the presence of the risk allele for rs7948643 possibly mediating CCND1 regulation in RCC.

The manuscript describes important findings for renal cancer biology. Lineage specific effects must contribute to the tissue specific development of cancers, but so far evidence for that especially in the kidney is lacking. Examples from other tissues exist such as TRPS1 for breast cancer (Witwicki et al. Cell Rep. 2018 Oct 30; 25(5): 1255–1267.e5. or SOX proteins in gastric cancer (Francis R. et al. Science Advances 11 Dec 2019:Vol. 5, no. 12, eaax8898 DOI: 10.1126/sciadv.aax8898). In addition, overexpression of PAX8 and PAX8 mediated chromatin changes have been linked to the development of various malignancies including ovarian cancer (Chaves-Moreira D. et al. Cancer Res. 2020 Dec 23;canres.3173.2020.doi: 10.1158/0008-5472.CAN-20-3173).

The authors use elegant experimental approaches to define PAX8 as a lineage specific factor potentially contributing to renal cancer development. In fact, PAX8 immunohistochemistry already is used in clinical routine to define renal tubular origin of tumors in the kidney, since it is a marker for clear cell and papillary tumors. The experimental set up to generate and validate the data appears to be adequate and precise. Though for some of the experiments only technical triplicates have been used, the experiments were conducted in many cell lines confirming the results. The manuscript is well written and data presentation is well intelligible. Selection of references is adequate and comprehensive.

I have some comments on specific aspects of the manuscript:

ATAC analysis.

PAX8 depletion and ATACseq: sites which show decreased accessibility upon PAX8 depletion are enriched in KIRC and KIRP specific ATAC sites (Figure 1e). For this analysis, were the PAX8 depleted sites derived from the renal cancer cells? If so, is it possible that the overlap between the two data sets is generated from the same tissue background rather than from specific PAX8 sites. How was this analysis done and what was the comparator? If PAX8 is necessary to promote oncogenic signaling in renal tumors why is it downregulated in tumors when compared to normal tissue? An overall reduction of PAX8 would also compromise accessibility to tumor promoting sites which is in contrast to the hypothesis presented by the authors and in contrast to the proposed effects of PAX8 in other tissues(compare Chaves-Moreira D. et al. Cancer Res. 2020 Dec 23;canres.3173.2020.doi: 10.1158/0008-5472.CAN-20-3173).

MYC regulation.

The observation that the PAX8/HNF1b axis regulates MYC is of great interest, but the work lacks more mechanistic insights into this regulation. Does MYC expression (RNA or Protein) correlate with PAX8/HNF1b expression in ccRCCs? Is the regulation observed mediated via direct promoter binding? Does an RCC specific enhancer module exist which binds PAX8 or HNF1b?

CCND1 enhancer and CCND1 expression.

KIRC-56592 is an established enhancer of CCND1 expression that has been detected in HIF ChIPseq experiments and comprehensive mapping of renal cancer associated enhancers (histone marks, FAIRE-seq) by various groups (Schodel et al. Nat Gen 2012 REF28, Yao et al. Cancer Discovery 2017 REF29). It has been physically linked to the CCND1 promoter using different chromatin capture technology (Schodel et al Nat Gen 2012, Platt et al. EMBO Rep. 2016 Oct;17(10):1410-1421.doi: 10.15252/embr.201642198) and tissue-specific HIF effects on CCND1 expression were described extensively in earlier reports. The SNPs described in the manuscript and identified by Purdue et al. (Nature Gen 2011 REF5) have been linked to HIF and CCND1 by Schodel et al. (REF28). This part of the current manuscript therefore confirms existing data, which is acknowledged in the manuscript. The observation that PAX8 contributes to CCND1 regulation is novel and may yield some interesting aspects for this enhancer. However, I wonder why relevant differences in CCND1 expression did not appear in PAX8 knock-out experiments in cells as reported in the comparison with the dependency scores (compare Figure 4a and 2e). Though in the supplementary data there is some repression of CCND1 expression upon PAX8 depletion in RCC cells (suppl. Table 2), this is in contrast to the prominent changes observed by HIF2A manipulation (suppl. Table 6) and the effects observed in the reporter assays which suggest that PAX8 is relevant for HIF signaling at this site (Figure 4f). Does loss of PAX8 influence HIF DNA interactions at this site? Do they directly interact at this site? This would complement the very nice data derived from the RIME experiments.

Regarding CCND1 expression it would be of interest whether PAX8 regulates CCND1 expression in cell lines derived from proximal tubules with and without HIF (e.g.HK-2). To my knowledge HK-2 cells (and other proximal tubular derived cell lines) do not regulate CCND1 via HIF, despite the expression of HIF2A (and the expression of PAX8). If the hypothesis of the authors that PAX8 maintains chromatin accessibility at this site is correct, than PAX8 and HIF2A binding should be observed at this site already in non-cancer derived cell lines (with HIF2A expression). In addition, papillary renal cell carcinomas with FH mutations overexpress HIF2A (Pollard PJ et al. Cancer Cell. 2007 Apr;11(4):311-9.doi: 10.1016/j.ccr.2007.02.005.), but enhancer activity appears to be reduced in these tumors at this site when compared to ccRCC (Figure 3h). The KIRP ATAC data set appears to have a strong (maybe even stronger than KIRC) PAX8 signature (Figure 1e). Does PAX8 bind to this region in papillary RCCs?

The authors point out that the enhancer might regulate the expression of genes other than CCND1. In fact, the enhancer has been shown to physically interact with the promoter of MYEOV (Platt J. et al Embo rep 2016) which is upstream of CCND1 and the enhancer and the second most downregulated gene in the HIF2A depleted xenografts (suppl. Table 6). Suggesting that regulation of this gene could be involved in (or even responsible for) the generation of the observed effects as well. Does PAX8 regulate the expression of MYEOV in a genotype specific manner?

Allele specific effects (Figure 4g-i).

The authors use reporter assays and EMSA to examine binding of PAX8 to the different alleles of rs7948643. Though the data fit to the hypothesis that PAX8 preferentially binds to the risk allele, data from chromatinized DNA should be included, i.e. ChIP for PAX8 in cell lines heterozygous for the SNP. The minor allele frequency of this SNP will be low, but this experiment would strengthen the results a lot.

Author Rebuttals to Initial Comments:

Point-by-point response to referee comments

We would like to thank the referees for a thorough review of our manuscript. We really appreciate the overall positive comments made about the novel concept emerging from our work and value the constructive remarks that have allowed us to considerably improve the study. In the revised manuscript we have addressed all the comments; our point-by-point reply is below. We have added or revised 58 data panels in total, 22 of which are presented in the main figures and 36 in the supplementary figures. The most notable new elements of the study are:

- 1. Human genetic data on RCC subtype-specific risk associated with rs7948643.** New analysis of a large human GWAS data set comprising thousands of cases and normal controls demonstrates that rs7948643 is very significantly associated with ccRCC but not papillary RCC, validating important predictions arising from our work in human genetic data.
- 2. Comprehensive characterisation of the distal *CCND1* enhancer that integrates signals from the *VHL-HIF2A* and *PAX8* pathways.** Detailed mechanistic analysis of the distal rs7948643-containing cyclin D1 enhancer, including long-read DNA sequencing-based haplotyping, showing allele specific PAX8 binding and transcriptional activity, thus solidifying the functional evidence for our model of lineage-factor dependent signalling by the two most common ccRCC-causing genetic alterations in humans.
- 3. Demonstration that the *PAX8-HNF1B* effect on proliferation and *MYC* expression in cancer cells is co-opted from normal renal epithelial cells.** Through CRISPRi-based functional screening, analysis of large-scale human DNA accessibility profiles, and experimental analysis of normal renal epithelial cells we show that HNF1B controls proliferation of cells of the renal lineage via specific distal *MYC* enhancers that are co-opted for carcinogenesis by ccRCC metastasis-associated 8q21.3-q24.3 amplifications. This clarifies the role of PAX8 and HNF1B as renal lineage factors and further highlights their function as essential determinants of oncogenic programmes that arise downstream of common cancer-associated genetic alterations.
- 4. Restructured presentation of the data.** To emphasize the most important and conceptually novel findings of our work, we have comprehensively restructured the revised manuscript. We hope this has improved readability and clarified the message.
- 5. Updated enhancer nomenclature.** There is currently no universally accepted guidance on how to name enhancers. To increase the readability of the revised manuscript, which contains analyses of several enhancers, we have adopted a simple approach that uses the approximate chromosomal location of each enhancer as their name. E.g. E11:69419 (the updated nomenclature for KIRC-56592) refers to an enhancer on chromosome 11 with approximate start position at 69419kb.

Referees' comments:

Referee #1 (Remarks to the Author):

“The key findings of this manuscript by Patel et al provides an intriguing linkage between the known lineage factor Pax8 transcription factor and VHL driven ccRCC. They show that Pax8 is necessary to drive proliferation, which has been demonstrated before (Bleu et al 2019) and promotes expression of HNF1b (also shown previously (Buisson et al 2015). The intriguing part comes from the connection of Pax8/HNF1b to be necessary to drive myc expression, and that this complex further mediates CCND1 expression by co-transcription of Pax8 and HIF2 on an open chromatin module previously identified as a risk allele for ccRCC. Here a variant snp reveals a dependency on both transcription factors. The concept of Pax8 being essential to allow HIF2 to function as a tumor promoter is intriguing and could provide an explanation for the kidney specificity of VHL mutation-driven tumors.”

“There are some issues that raise concern over the extent of the conceptual advance this puts forward.”

Reply 1.1. We appreciate the referee’s positive comments about our work and hope that the extensive new data and improved presentation of the revised manuscript clarify our message and the conceptual advance our results put forward.

“1. the bulk of the findings - of Pax8 dependency for growth (Bleu 2018), of Myc dependency (Tang 2009), of Pax8 driving HNF1b (Buisson 2015), ATACseq in KIRC and KIRP tumors demonstrating more open chromatin (Corces 2017, referenced), have been reported previously, such that the truly new concept is a fairly small fraction of the whole story. This most compelling portion, regarding the interaction of Pax9 and HIF2 with the SNP is treated fairly superficially for such a pivotal finding.”

Reply 1.2. We agree that the most important discovery of our work is that oncogenic signalling by ccRCC-associated genetic alterations is dependent on the renal lineage factor PAX8, a conclusion further strengthened by significant new data in the revised manuscript. This finding is new and it provides us with a novel concept that is likely to be applicable more generally across different tumour types. We also agree that lineage factors and central oncogenic regulators such as MYC have been linked to cancer before. However, the previous studies listed here by the referee, while important, do not contain any data on the interaction between renal lineage factors and genetic alterations that drive ccRCC formation. In fact, Bleu et al. (1) suggest that the effects of PAX8 are independent of VHL mutation status, missing the lineage factor-dependent genetic mechanism we have identified. Thus, while we have replicated some previous observations within our study, an essential component of reproducible science that solidifies the basis of our work, there is no conceptual overlap between these studies and the main finding of our paper.

We appreciate that the previous version of our manuscript may not have adequately emphasized the most important findings of our work, and that additional data were needed to solidify our conclusions. We have addressed these points in the revised manuscript by restructuring the presentation to highlight the central discovery of interaction between PAX8 and the genetic alterations that drive ccRCC, and by providing substantial new data sets that provide strong additional support for our conclusions. In particular, we provide human genetic data on the strong association between rs7948643 and ccRCC, but not papillary RCC (new data in **Fig. 3l**); we demonstrate the effects of rs7948643 variants on PAX8 binding in the chromatin context (new data in **Fig. 3i**); we provide long-read DNA sequencing data that link rs7948643 to a specific CCND1 3’UTR variant and show that PAX8 preferentially supports expression from the risk allele (new data in **Fig. 3h,j,k** and **Extended Data Fig. 7h,k**); we provide evidence of the role of PAX8 as a facilitator of HIF2A binding and their collaborative function as transcriptional activators of the rs7948643-containing CCND1 enhancer, not

pioneer factors that control initial enhancer accessibility (new data in **Extended Data Fig. 7c-g,i,j**); and we characterise the detailed role of the PAX8-HNF1B-axis as a regulator of *MYC* expression in cells of the renal lineage, highlighting this pathway as a cancer co-opted physiological programme needed for the oncogenic activity of ccRCC metastasis-associated 8q21.3-q24.3 amplicons (new data in **Fig. 4d-j** and **Extended Data Fig. 10a-j**). These extensive new data and conclusions provide strong support for the central novel finding of our paper, which is that three common genetic driver alterations in ccRCC depend on the renal lineage factor PAX8 for oncogenic activity.

*“2. The cell lines used were selected for aggressive metastatic behaviour, thus there may be evolved features, particularly with regard to *myc* and *CCND1* signaling, that are not solely linked to *VHL* loss and *HIF2* expression. At some points the work switches back to the parent cell line 786-O. The rationale for these selections should be clear.”*

Reply 1.3. We agree that the choice of experimental models is important and has to be well-justified. The metastatic ccRCC clones were selected for our initial screen for several reasons. First, metastases cause the majority of cancer-related deaths and understanding their biology is therefore of significant clinical relevance. Second, the 786-M1A and OS-LM1 models have been extensively characterised at the phenotypic, genomic, transcriptomic and gene regulatory levels with parallel clinical validation in multiple previous studies (2–5). Importantly, genetic evolution of ccRCC metastases has been dissected in detail in human data sets, with little evidence emerging for metastasis-specific genetic alterations (6,7). On the other hand, previous results by us and others have clearly demonstrated the relevance of the tumour-initiating *VHL*-*HIF2A* pathway in advanced ccRCC (2,3,8), a finding supported by recent clinical data with a *HIF2A* inhibitor (9). This suggests that the tissue-specific molecular mechanisms that lead to the selection of *VHL* mutations during ccRCC formation are likely to persist in advanced ccRCC clones, a possibility now supported by our present results.

As suggested by the referee, phenotypic evolution associated with metastasis can lead to rewiring of oncogenic pathway activity. Indeed, our previous studies have described precise mechanisms of how the activity of the *VHL*-*HIF2A* pathway evolves in support of metastasis (2,3). It is clear from our current work, however, that the mechanisms described in the present paper are relevant for non-metastatic ccRCC clones as well. First, the role of the PAX8-HNF1B-*MYC* pathway is supported by results from several different *VHL*-mutant ccRCC lines, most of which have not been selected for metastatic potential (**Extended Data Fig. 1a-b** and **Extended Data Fig. 8f**), as well as by our new data on normal renal epithelial cells (new data in **Fig. 4h-j** and **Extended Data Fig. 10e-j**). Also, the mechanism of *CCND1* activation is shown in our metastatic ccRCC clones but also by our new data on RCC-JF cells, which are heterozygous for the ccRCC risk SNP rs7948643 (new data in **Fig. 3i-k** and **Extended Data Fig. 7c,f,k**). Of note, the reporter assays were performed on both the parental 786-O cells as well as the metastatic derivative 2801-LM1, as these cells do not carry the luciferase-GFP reporter that was used for the original isolation of the 786-M1A and OS-LM1 cells (3). The reporters behave similarly in the parental and metastatic clones (**Fig. 3b,c,e**). Moreover, the new analysis of the human GWAS data give strong support for our mechanistic analysis from the context of ccRCC incidence at the population level (new data in **Fig. 3l**).

In sum, several lines of evidence suggest that the mechanisms described in this paper are not specific for metastatic ccRCC, but rather represent conserved programmes that operate at several stages of ccRCC development. We have revised the text to highlight this by explaining our choice of cell lines and their characteristics in more detail, as well as by including a substantial amount of new data that support the conclusions of our work.

*“3. The *HIF2* driving feature is not developed sufficiently to understand how or if *HIF1* would engage this process. Specific studies to add *HIF2* would address this.”*

Reply 1.4. We agree that distinguishing between *HIF1A* and *HIF2A* is important. We have now addressed this point by showing that *CCND1* expression is dependent on *HIF2A*, but not

HIF1A (new data in **Extended Data Fig. 6h-k**) and by performing the E11:69419 enhancer reporter assay under HIF2A inhibition (new data in **Extended Data Fig. 7a**). These data, together with the extensive previous data showing the pro-tumorigenic role for HIF2A (8,10), but not for HIF1A (11), in ccRCC give strong support to our model in which the interaction between HIF2A and PAX8 is critical for ccRCC formation.

“4. Additional information regarding the prevalence of the SNP, any skew in type of kidney cancer based on SNP, or other relevant human data would be valuable.”

Reply 1.5. This is an excellent suggestion, thank you. We have now performed this analysis in a combined human RCC GWAS data set (12). The data set comprises 6,211 RCC cases for which the subtype is known as well as 15,000 healthy controls. As predicted, we observe a very strong association between rs7948643 and ccRCC ($P = 2.4 \times 10^{-10}$), but no evidence of association for papillary RCC ($P = 0.9$). These data provide strong human genetic support for our model (new data in **Fig. 3I**).

“5. The authors should also consider that HNF1b is implicated as an important factor in chromophobe RCC (KICH), which is considered to arise from the distal tubule (Sun 2017).”

Reply 1.6. As opposed to ccRCCs, which arise from the renal proximal tubules, chromophobe RCCs are thought to arise from distal tubules. They have a distinctive histological appearance and their genetic driver alterations are different from those in ccRCC. Most notably, chromophobe RCCs are not associated with *VHL* mutations, demonstrating the fundamental difference between chromophobe RCCs and ccRCCs. As noted by the referee, work by Sun et al. has demonstrated that chromophobe RCCs typically lose the expression of HNF1B (13), whereas our data clearly shows a strong pro-tumorigenic function for HNF1B in ccRCC. These results further highlight the fact that chromophobe RCCs and ccRCCs are different disease entities. Why the role of HNF1B varies in these two different tumour types is an interesting question that can be addressed in future studies, but it is not central to the arguments of the present manuscript.

“6. Additional discussion regarding the potential mechanisms by which VHL loss does drive tumor formation in the hemangioblastomas, pheochromocytomas, and pancreatic NET tumors in the setting of VHL disease would be valuable.”

Reply 1.7. We agree that the association between germline *VHL* mutations and tumour types other than ccRCC should be highlighted. In our study we describe a PAX8-dependent mechanism for HIF2A in ccRCC formation, a finding supported by extensive functional and genetic data. This mechanism, in combination with our results on the requirement of the PAX8-HNF1B pathway in supporting *MYC* expression, provides general evidence for a novel model by which oncogenic signalling downstream of cancer-associated genetic alterations is dependent on tissue-specific lineage factors. The uneven distribution of cancer driver mutations across tumour types in both somatic and inherited contexts indicates that this general mechanism may be applicable to other tissues and tumour types as well.

Genotype-phenotype correlation analysis in patients with germline *VHL* mutations has revealed that the development of pheochromocytomas is associated with specific mutations that do not always lead to dysregulated HIF signalling (14), indicating that the tumorigenic mechanism is likely to be different from the one described here for ccRCC. On the other hand, *VHL* loss-associated pancreatic neuroendocrine tumours (PaNETs) show signs of increased hypoxia signalling and *CCND1* expression (15), but do not express PAX8 (16). Similarly, hemangioblastomas express *CCND1* (17) but not PAX8 (18). The specific molecular mechanisms that underlie the interaction between PAX8 and HIF2A in renal carcinogenesis are therefore unlikely to play a role in the other *VHL* syndrome-associated tumour types. Moreover, our new data suggest that PAX8 and HIF2A control enhancer activation, but they alone are not capable of establishing accessibility at new genomic loci (new data in **Extended**

Data Fig. 7d-g). This highlights the fact that the mechanisms that lead to tissue-specific oncogenic programmes is controlled at several different levels of epigenetic regulation, one of which is the PAX8-HIF2A interaction described in the present study. However, the general mechanism identified here is likely to also be applicable to other tumour contexts, in which *VHL* mutations may be dependent on lineage factors for oncogenic signalling, a possibility that should be investigated in further studies. We have added discussion about the *VHL* syndrome and other tumour types, as well as the layered epigenetic regulation of oncogenic programmes, in the revised manuscript.

“7. Prior work by this group has endorsed co-option of an NFκB transcriptional hub with HIF2 as a mechanism of metastasis. How does this finding integrate with that transition?”

Reply 1.8. In our previous work we have described mechanisms that facilitate the expansion of the *VHL*-HIF2A pathway phenotypic output in support of metastasis (2,3). In particular, we have demonstrated that enhanced NF-κB signalling leads to the activation of a distal enhancer, which in collaboration with another HIF2A-bound enhancer activates *CXCR4* expression and consequently enhances metastatic fitness (2). These results provide an explanation for the fact that while *VHL* mutations are ubiquitous in ccRCC, only in some ccRCCs the *VHL*-HIF2A pathway activates metastasis genes. This mechanism, while perfectly compatible with our current results, is not related to the earlier stages of ccRCC formation and addresses a different question from our present work. It is notable, however, that the HIF2A-bound enhancer upstream of *CXCR4* which we have previously shown to promote ccRCC metastasis (2), also binds PAX8. Thus, the interaction between lineage factors and oncogenic drivers is likely to be relevant even for the activation of advanced cancer phenotypes at the transition towards metastasis.

Referee #2 (Remarks to the Author):

“Patel and colleagues investigate the rationale for tissue-specific tumorigenesis in the context of sporadic clear cell renal cell carcinoma (ccRCC). Through a transcription factor (TF) screen in two RCC cell lines they uncover PAX8-HNF1B-MYC axis as a ccRCC-specific mechanism for acquiring the proliferative phenotype. Through interrogation of the protein interactome and ChIP-seq peak overlaps, and functional study of DNA motifs they demonstrate that PAX8 cooperates with HIF2A to activate downstream CCND1 expression through a distal enhancer – rs794863, a ccRCC risk SNP, illustrating potential molecular basis of this common ccRCC risk variant.”

“Major comments

1. Regarding the cell lines in which the screen was performed. I was not able to find information on OS-LM1, while 786-M1A1 is a human line that was made metastatic by injection in a NOD-SCID mouse and derived from consequent lung metastases which likely would have led to selection of a specific phenotype. An explanation for the choice of cell lines and information on key genotypes would be appreciated given this can impact the generalizability of the study findings in a cancer defined by a variety of genetic alterations.”

Reply 2.1. We agree that the choice of experimental models is important and has to be well-justified. The metastatic ccRCC clones were selected for our initial screen for several reasons. First, metastases cause the majority of cancer-related deaths and understanding their biology is therefore of significant clinical relevance. Second, the 786-M1A and OS-LM1 models, both of which were similarly derived from experimental lung metastases (3), have been extensively characterised at the phenotypic, genomic, transcriptomic and gene regulatory levels with parallel clinical validation in multiple previous studies (2–5). They both carry truncating mutations in *VHL*. 786-M1A cells also carry a *PTEN* p.Q149* mutation. Importantly, tumours

formed by these cell lines in mice recapitulate salient histological features of human ccRCC (3), indicating that they contain the genetic and gene regulatory information required for ccRCC formation.

Several lines of evidence suggest that the mechanisms described in our work are not limited to the cell lines used for our initial genetic screen. The lineage factor-dependent growth phenotype has been validated in several *VHL* mutant ccRCC cell lines (**Extended Data Fig. 1a-b** and **Extended Data Fig. 8f**). Also, the mechanisms of *CCND1* activation is shown in our metastatic ccRCC clones but also by our new data on RCC-JF cells, which are heterozygous for the ccRCC risk SNP rs7948643 (new data in **Fig. 3i-k** and **Extended Data Fig. 7c,f,k**). Of note, the reporter assays were performed on both the parental 786-O cells as well as the metastatic 2801-LM1 cells, as these cells do not carry the luciferase-GFP reporter that was used for the original isolation of the 786-M1A and OS-LM1 cells (3). The reporters behave similarly in the parental and metastatic clones (**Fig. 3b,c,e**). Moreover, the new analysis of the human GWAS data give strong support to our mechanistic analysis in the context of RCC incidence (new data in **Fig. 3l**).

In sum, several lines of evidence suggest that the mechanisms described in this paper are not specific for the cell lines used in our screen, but rather represent conserved programmes that operate at several stages of ccRCC development. We have revised the text to highlight this by explaining our choice of cell lines and their characteristics in more detail, as well as by including a substantial amount of new data that support the conclusions of our work.

“2. There is a possibility that PAX8 and HNF1B maintain chromatin activation states and fitness in normal renal cell lineage rather than renal cancer, and the authors should implement a normal kidney control such as HK2 or PTEC cell lines for the knock out and downstream experiments (besides HeLa).”

Reply 2.2. We agree, several lines of evidence suggest that PAX8 and HNF1B are renal lineage factors and not factors specific to renal cancer. Indeed, the central hypothesis underlying this project was that tissue-specific lineage factors, such as PAX8 and HNF1B, are critical determinants of the oncogenic phenotypes that arise downstream of tissue and/or cancer type-specific oncogenic mutations. We demonstrate this to be the case, with extensive new data provided in the revised manuscript, by showing that the ccRCC risk SNP rs7948643 modulates the potential of PAX8 to regulate the activity of a critical ccRCC enhancer that also depends on the activity of the VHL-HIF2A pathway upstream of *CCND1*, and by characterizing in detail the mechanism through which the PAX8-HNF1B axis regulates *MYC* through distal gene regulatory elements (new data in **Fig. 4d-h** and **Extended Data Fig. 10a-d,j**). PAX8 function at E11:69419 is likely to be cancer cell specific as it is also dependent on the mutationally activated VHL-HIF2A pathway. In agreement, there is little evidence of E11:69419 accessibility in tissues other than RCC (**Fig. 2f**), and PAX8 depletion does not inhibit *CCND1* expression in non-cancerous renal epithelial cells (**Extended Data Fig. 10i**). However, our new data suggest that PAX8-HNF1B-dependent *MYC* expression in cancer cells is co-opted from a physiological programme that also supports proliferation in normal renal epithelial cells. In the revised manuscript we show this using human renal epithelial organoids and the renal epithelial cell line HK-2, as suggested by the referee. As in ccRCC cell lines, PAX8 is required for proliferation and it regulates the expression of *HNF1B* and *MYC* in HK-2 cells and human renal epithelial organoids (new data in **Fig. 4h-j** and **Extended Data Fig. 10e-h**). The two enhancers that support HNF1B-dependent *MYC* expression and proliferation in ccRCC cells also promote *MYC* expression in HK-2 cells (new data in **Fig. 4g** and **Extended Data Fig. 10j**). These data give strong support to the novel concept arising from our results, i.e. that the oncogenic potential of cancer-associated genetic alterations is critically dependent on tissue-specific lineage factors. We have edited the text throughout the manuscript to clarify this point.

“3. The authors have focused on the proliferative phenotype with cell competition assay and tumor xenograft growth measurements, but should evaluate other features such as invasion/metastases formation and tumour histology.”

Reply 2.3. We agree that depending on the specific research question studying advanced cancer phenotypes such as metastasis can be important. However, the genetic evidence on *VHL* loss and the ccRCC risk SNP rs7948643 suggest that their associated pro-tumorigenic mechanisms operate already early in ccRCC formation (12,19,20). Also, as discussed in detail in our Reply 2.1 above, the *CCND1* upstream regulatory mechanisms described in our manuscript are shared by metastatic and non-metastatic cancer clones. Similarly, the physiological lineage factor dependent proliferative programme that regulates *MYC* expression through specific distal enhancers in cancer cells is already active in normal cells of the renal lineage. Thus, while we agree that the mechanisms we have discovered are likely to play a role also in the growth of ccRCC metastases, we are confident that the experimental assays used, ranging from detailed mechanistic analysis of enhancer function to experimental tumour assays and human GWAS data, are sufficient for the questions we have focused on in this work. Based on the referee comment, it is also not obvious to us what specific questions related to E11:69419, rs7948643 or the PAX8-HNF1B-MYC axis would need additional experimental metastasis assays. We are, however, happy to perform such experiments if needed for specific reasons.

We appreciate that analysis of tumour histology is critical. As demonstrated before, the main ccRCC models used in this work, the 786-M1A and OS-LM1 cells, form xenograft tumours that are indistinguishable from human ccRCC (3). Specifically, tumours by 786-M1A cells display a dedifferentiated phenotype with sarcomatoid features, whereas OS-LM1 cells display a more classical clear cell histology. **[Redacted]**. We agree that it would be interesting to study the effects of PAX8/HNF1B inhibition on tumour histology. However, based on our data, the cells that grow out from PAX8 and HNF1B knock-out tumours represent escapers in which PAX8 or HNF1B expression has most likely never been lost. This makes it difficult to assess the effects of PAX8/HNF1B inactivation on tumour histology with the currently available models.

[Figure redacted]

“4. The authors clearly demonstrate PAX8-HNF1B-MYC axis as a mechanism underlying the proliferative phenotype, however it would be interesting to know whether this is a consequence of upstream transcriptional motif binding and activation.”

Reply 2.4. We agree that this is an interesting point. The PAX8-HNF1B-MYC axis supports ccRCC proliferation, with the prediction that HNF1B directly or indirectly promotes *MYC* expression. In order to test this model, we have now conducted a proliferation-based CRISPRi screen to identify relevant *MYC* enhancers (new data in **Fig. 4d-e**). Combining these data with chromatin accessibility and ChIP-seq data we have identified two enhancer elements that (i) are needed for ccRCC proliferation (new data in **Fig. 4d**), (ii) show reduced accessibility upon HNF1B and PAX8 depletion by RNAi (new analysis in **Fig. 4f** and **Extended Data Fig. 10d**), (iii) show evidence of HNF1B binding by ChIP-seq (new analysis in **Fig. 4e**), and (iv) contain the predicted HNF1B binding motif. Independent CRISPRi-based targeting of these two enhancers led to reduced *MYC* expression (new data in **Fig. 4g**). Importantly, as discussed above in our Reply 2.2 above, we find that these two enhancers are also important for *MYC*

expression in the non-cancerous HK-2 renal epithelial cells (new data in **Extended Data Fig. 10j**). Moreover, comprehensive whole genome sequencing analysis has recently identified large 8q21.3-q24.3 amplifications in association with ccRCC metastasis (21). This region covers the *MYC* coding region but also the regulatory regions now identified by us as HNF1B-dependent *MYC* enhancers. Interestingly, our new FISH data shows that the metastatic 786-M1A cells carry six copies of the *MYC* locus (new data in **Extended Data Fig. 10a**), but they are still dependent on the PAX8-HNF1B axis for *MYC* expression (**Fig. 4c**). These data thus demonstrate that the ccRCC metastasis-associated genetic amplifications are dependent on the renal lineage factor module for *MYC* expression and proliferation, further highlighting the general concept arising from our work, i.e. that tissue-specific lineage factors are required for oncogenic signalling by cancer-associated genetic alterations. We have revised the text extensively to include these new data.

“5. The interaction between PAX8 and HIF2A was inferred by RIME interactome and ChIP-seq, but direct interaction should be investigated by CoIP or pulldown assay/ Furthermore, KIRC-56592 is 220kb away from CCND1, suggesting additional mediators contribute to how this distal enhancer works.”

Reply 2.5. We agree that direct protein-protein interaction is a possible mechanism through which two transcription factors can jointly mediate transcriptional activation. However, functionally relevant transcription factor interaction can take place at multiple different levels. First, transcription factors may be dependent on heterodimerization, as is the case with HIF2A and HIF1B/ARNT, both of which are required for the formation of a functional DNA binding domain (22). In such cases the interaction is typically strong and mediated by significant interactive surfaces on the two proteins (22). Second, two transcription factors can bind DNA co-operatively through a DNA-facilitated manner in which case only some amino acids in the collaborating factors interact directly (23). Third, transcription factors may bind DNA in a completely DNA-mediated yet co-operative manner, in which subtle transcription factor binding-induced alterations in the nearby DNA structure are thought to enhance the binding of the second factor (24,25). Systematic high throughput analysis suggests that the vast majority of co-operative DNA binding by transcription factors falls in this category and is not dependent on direct protein-protein interaction, even when the DNA motifs are overlapping (23). Finally, transcription factors may co-operate at a functional level within an enhancer by independently binding DNA but collaboratively regulating the activation of the enhancer. In contrast to co-operative binding, which in the vast majority of cases depends on the DNA motifs being <5bp from each other (23), this kind of functional enhancer level interaction between transcription factors can occur over any distance within the enhancer DNA sequence.

The RIME experiment, which is based on two steps of cross-linking, can detect interaction that is mediated through all these different mechanisms. Similarly, ChIP-seq experiments can detect any enhancer-level interaction as co-localization within chromatin. This is the benefit of RIME-based approaches and transcription factor co-mapping validation, since it reveals functional connections that are genuine, but might be missed by less sensitive traditional approaches. On the other hand, co-IP experiments without cross-linking, as suggested by the referee, are likely to only detect strong protein-protein interactions, such as that between HIF2A and HIF1B/ARNT (22). In fact, we know that oestrogen receptor (ER) activity in breast cancer (work from one of the co-authors, Prof. Jason Carroll) involves the pioneer factor FOXA1 in mediating ER-enhancer interactions. This is now a well validated functional interaction and the paradigm of nuclear receptor activity (26–29). Despite the fact that FOXA1 was discovered as an ER-associated protein 15 years ago, via ChIP-seq co-binding and (more recently via) RIME-based discovery (30), the Carroll lab is yet to successfully show an interaction by co-IP, attesting to the limitations of traditional biochemical approaches.

Given that most known co-operative DNA-binding interactions between transcription factors are dependent on close proximity and defined distances between their individual motifs on DNA (23), the distribution of the distances between PAX8 and HIF2A motifs within the

PAX8/HIF2A co-bound enhancers can be used to infer the type of interaction they are likely to be engaged in. In other words, if PAX8/HIF2A co-bound ccRCC enhancers depend on a specific orientation determined by PAX8 and HIF2A structures, we would expect there to be an enrichment of PAX8 and HIF2A motifs in the specific orientation and distance that would allow this interaction to happen. On the other hand, if we see no enrichment for a particular configuration, the interaction is likely to be mediated by DNA and shared protein complexes, but not through direct protein-protein interaction. New analysis in **Extended Data Fig. 4f** shows that the distance between PAX8 and HIF2A motifs ranges between 0-500 nucleotides with a continuum over the whole range and different orientations, the only exception being the HIF2A-PAX8 motif sequence with an 18bp distance, which is repeatedly observed in the LTR sequence of the common endogenous retrovirus ERV1. Interestingly, ERV1 expression has been recently linked to ccRCC progression (31,32). The sequence level data thus suggest that PAX8 and HIF2A are unlikely to interact directly at the protein level, and that their cooperative effects on transcription (**Fig. 2a,j** and **3c**) is most likely mediated by enhancer level or enhancer cluster level (2) interaction through DNA and shared protein complexes. In line with this, and in contrast to PAX8 interaction with transcriptional co-factors that do not directly bind DNA, such as PRDM3 that is recruited to chromatin by PAX8 in ovarian cancer (33), we did not detect direct interaction between PAX8 and HIF2A in ccRCC cells by co-IP, while HIF2A showed interaction with HIF1B/ARNT, as expected (new data in **Extended Data Fig. 4g**). However, we do see evidence of PAX8 facilitating HIF2A binding at E11:69419, but not vice versa (new data in **Extended Data Fig. 7i-j**). We have clarified these important points on the nature of PAX8-HIF2A interaction in the revised manuscript.

As stated by the referee, gene activation by distal enhancers is a complex process that depends on several different co-factors in addition to the effects of DNA-binding transcription factors. These factors are recruited to specific genomic loci by transcription factors and include proteins with various functions ranging from nucleosome remodelling, histone acetylation, 3D chromatin structure and general transcription etc. (34). This is reflected in our proteomic data, in which the PAX8 and HIF2A-associated nuclear complexes, in addition to several transcription factors with DNA binding specificities, contain various transcriptional co-factors (**Fig. 1e**, **Extended Data Fig. 4a** and **Supplementary Tables 1-2**). For example, the histone acetyltransferases p300 and CBP (PAX8 RIME) and the several members of the SWI/SNF chromatin remodelling complex, such as SMARCA2, SMARCA4 and SMARCC2, are known to regulate enhancer function (35,36). Moreover, the structural proteins SMC1A (PAX8 and HIF2A RIME) and STAG2 (PAX8 RIME) mediate enhancer-promoter looping (37,38). In line with this, the E11:69419 region has been shown to interact with the promoters of the nearby genes *CCND1* and *MYEOV* (39). The specificity of enhancer activation is, however, dependent on the DNA sequence, which can only be read by transcription factors with DNA binding specificities. Our data show that E11:69419 is required for *CCND1* expression and tumorigenic potential in ccRCC cells, and we provide detailed insight into the combinatorial transcription factor requirements of E11:69419 activation, suggesting a molecular basis for the increased genetic ccRCC risk associated with rs7948643.

“6. It would be useful to explore the findings in a clinical context such as survival data in TCGA (E.G. PAX8 expression).”

Reply 2.6. We agree that human data is important. Indeed, and as discussed above, the revised manuscript contains critical new results from a large human GWAS data set that supports our experimentally characterised model of rs7948643 function as a risk locus for ccRCC, but not papillary RCC (new data in **Fig. 3I**). Another prediction of our model, as pointed out by Referee 3, is that *CCND1* expression should correlate with *HIF2A* expression more strongly than with *PAX8* expression in ccRCCs. We find this to be the case: the expression of *HIF2A* and *CCND1* correlate strongly in the TCGA data set (Pearson’s correlation coefficient (PCC) = 0.54; $P = 1.9 \times 10^{-37}$), whereas the correlation between *PAX8* and *CCND1* expression is significantly weaker (PCC = 0.12, $P = 0.005$). These results give additional clinical support

to our model and they have been added into the revised manuscript (new data in **Extended Data Fig. 6o**).

We agree that correlative analysis of clinical data sets in terms of patient survival can be informative in some contexts, although the results can be difficult to interpret, as recently systematically demonstrated (40). In terms of patient survival and *PAX8* expression in the TCGA data set, it is not clear what the predicted association should be based on our data, as the mechanisms we describe in the present study are not specifically linked to advanced cancer phenotypes, such as metastatic competence. For analogy, *VHL* mutations in ccRCC are truncal and mRNA expression of the best characterised downstream effector HIF2A is associated with good patient survival (41). Yet, the *VHL*-HIF2A pathway can co-opt new target genes, such as *CXCR4*, the expression of which is very strongly associated with metastasis and poor patient outcome (2,3). Similarly, *PAX8* expression, which marks the renal epithelium from which ccRCCs originate, is present already in the trunk of ccRCC, and as such its expression is unlikely to directly lead to the most aggressive cancer phenotypes that are associated with poor outcome. This is in line with our analysis of the TCGA data set (referee **Fig. R2**). However, the transcriptomic output of the *PAX8* pathway may also change through epigenetic alterations that could support metastasis, and interesting avenue for future work.

Figure R2. Association between *PAX8* mRNA expression and progression free survival in the TCGA ccRCC cohort. Patients divided to bottom, mid and top thirds of *PAX8* expression. *P*-value from a Cox proportional hazards model with *PAX8* expression treated as a continuous variable.

“7. Line 179-181, what about rescue experiments on *HNF1B* depletion cells?”

Reply 2.7. We agree, this is an important control experiment. We have confirmed that *HNF1B* re-expression can rescue the proliferative defect caused by CRISPR/Cas9-induced *HNF1B* knockout. These new data are shown in the **Extended Data Fig. 8h** of the revised manuscript.

“8. Line 198-203, could the authors rescue *MYC* expression in *HNF1B*-depleted cells to further validate the *PAX8*-*HNF1B*-*MYC* axis.”

Reply 2.8. We agree, this is an important experiment. We have confirmed that *HNF1B* expression in *PAX8* depleted cells rescues *MYC* expression. These new data are shown in **Fig. 4c** of the revised manuscript.

“9. Both ccRCC and papillary RCC are dependent on *PAX8* (Fig1), while *KIRC-56592* is only essential in ccRCC (Fig3h), can the authors reconcile this?”

Reply 2.9. We agree that the difference between ccRCC and papillary RCC in terms of E11:69419 (the updated name for *KIRC-56592*) function is of interest. Our data show that the activity of E11:69419 is dependent on *PAX8* and the genetically activated *VHL*-HIF2A pathway (**Fig. 3c**). Even though both tumour types express *PAX8*, *VHL* mutations are commonly detected only in ccRCC. Thus, based on these data it is expected that E11:69419 should only be active in ccRCC, but not in papillary RCC. In addition to the chromatin accessibility data (**Fig. 2e**), this model is strongly supported by our new genetic data from human GWAS analysis, which shows very significant association between rs7948643 and ccRCC but not papillary RCC (new data in **Fig. 3i**). Moreover, our new data using the renal epithelial HK-2 cells show that *PAX8* alone, which is endogenously expressed in these cells, is not sufficient

to activate E11:69419 in enhancer reporter assays, whereas the expression of a constitutively stable HIF2A in these cells leads to E11:69419 activation (new data in **Extended Data Fig. 7d-e**). However, the endogenous E11:69419 is not accessible in HK-2 cells, even after long-term HIF2A expression (new data in **Extended Data Fig. 7f**). These data, combined with the new observations that solidify the role of PAX8 as an allele-specific mediator of rs7948643 and E11:69419 effects on transcription (**Fig. 3h-k** and **Extended Data Fig. 7h,k**), suggest a model whereby the transcriptional activation of E11:69419 is dependent on the combined action of PAX8 and HIF2A, but the initial accessibility at this locus is controlled by factors that are not present in HK-2 cells. The activation of tissue-specific oncogenic enhancers is thus controlled at several epigenetic layers, a point now clarified in the discussion of the revised manuscript.

As opposed to the well-defined role of *VHL* mutations as tumour-initiating events in ccRCC, it is less clear what the critical drivers of early papillary RCC are (42). Although both RCC subtypes are thought to originate from the renal proximal epithelium, ccRCCs and papillary RCC may develop from different cell types, leading to different oncogenic trajectories. However, another possibility is that the early mutations define the oncogenic trajectory of renal proximal epithelial cells. In this scenario early *VHL* mutations would lead to ccRCC, whereas other yet unidentified molecular events would take the tumour-initiating cell towards a papillary RCC phenotype, which would by definition not be dependent on *VHL* mutations and downstream oncogenic driver mechanisms. The oncogene *MET*, mutated in ~10% of papillary RCCs but not in ccRCC could serve this function. Current data is compatible with both scenarios and further work on papillary RCC will be required before we can reach a definitive conclusion. It is worth noting, however, that apart from mutations in *VHL* and *MET*, the pattern of genetic driver alterations between these RCC subtypes is very similar (42), indicating that after tumour initiation the pressures that lead to the selection of cancer driver mutations may be shared between these tumour types.

“10. Give the tissue specificity of inherited VHL disease it would be of interest for the authors to comment on their findings in the context of this cancer syndrome.”

Reply 2.10. We agree that the association between germline *VHL* mutations and tumour types other than ccRCC should be highlighted. In our study we describe a PAX8-dependent mechanism for HIF2A in ccRCC formation, a finding supported by extensive functional and genetic data. This mechanism, in combination with our results on the requirement of the PAX8-HNF1B pathway in supporting *MYC* expression, provides general evidence for a novel model by which oncogenic signalling downstream of cancer-associated genetic alterations is dependent on tissue-specific lineage factors. The uneven distribution of cancer driver mutations across tumour types in both somatic and inherited contexts indicates that this general mechanism may be applicable to other tissues and tumour types as well.

Genotype-phenotype correlation analysis in patients with germline *VHL* mutations has revealed that the development of pheochromocytomas is associated with specific mutations that do not always lead to dysregulated HIF signalling (14), indicating that the tumorigenic mechanism is likely to be different from the one described here for ccRCC. On the other hand, *VHL* loss-associated pancreatic neuroendocrine tumours (PaNETs) show signs of increased hypoxia signalling and *CCND1* expression (15), but do not express PAX8 (16). Similarly, hemangioblastomas express *CCND1* (17) but not PAX8 (18). The specific molecular mechanisms that underlie the interaction between PAX8 and HIF2A in renal carcinogenesis are therefore unlikely to play a role in the other *VHL* syndrome-associated tumour types. Moreover, our new data suggest that PAX8 and HIF2A control enhancer activation, but they alone are not capable of establishing accessibility at new genomic loci (new data in **Extended Data Fig. 7d-g**). This highlights the fact that the mechanisms that lead to tissue-specific oncogenic programmes is controlled at several different levels of epigenetic regulation, one of which is the PAX8-HIF2A interaction described in the present study. However, the general mechanism identified here is likely to also be applicable to other tumour contexts, in which *VHL* mutations in other tumour types may be dependent on lineage factors for oncogenic

signalling, a possibility that should be investigated in further studies. We have added discussion about the VHL syndrome and other tumour types, as well as the layered epigenetic regulation of oncogenic programmes, in the revised manuscript.

“Minor comments

1. Fig1a, there is also an outlier in the upright quadrant, why was this sensitive TF omitted?”

Reply 2.11. This data point has been labelled for clarification in the revised figure panel (**Fig. 1a**). The gene is *TP53*, the most commonly mutated tumour suppressor in human cancers, but it is not of particular interest in the context of this study.

“2. Line 61, ref 9, relates not to HIF2A but HIF1A.”

Reply 2.12. Thank you for pointing this out, we have corrected the references in this section.

“3. Extended Fig1d, the knock out efficiency appears low, especially in 786-M1A and OS-LM1 cells with sgRNA No.4, but inspite of this while the tumor formation is efficient; suggest repeating the western blot”

Reply 2.13. Thank you for pointing this out. We have added in **Extended Data Fig. 1g** another Western blot which originally confirmed the level of PAX8 depletion in the batch of cells that were used in the *in vivo* experiment. For the *in vitro* experiment, the level of PAX8 depletion correlated well with the proliferative phenotype (**Extended Data Fig. 1d**).

“4. In extended Fig5d, could the authors annotate PAX8 and HNF1B in the two volcano plots and reference the figure in lines 166-167.”

Reply 2.15. Thank you for the suggestion, we have made the changes in the revised manuscript (new **Extended Data Fig. 4h**).

“5. In Fig2a,b, the sgRNAs used in mRNA validation and protein validation appear inconsistent.”

Reply 2.16. Thank you for pointing this out. We used three different PAX8 shRNAmiR constructs, two for qPCR analysis and two for Western blots. We have double-checked the labels to make sure they are correct. The shRNAmiR sequences are also listed in **Supplementary Table 8**.

“6. The statement that MYC is not “commonly amplified in ccRCC” ignores the frequent gain of 8q which contains MYC locus.”

Reply 2.17. We agree that this is a very important point that we have now thoroughly addressed. While early stage ccRCCs, such as the majority of the TCGA cohort, typically do not carry amplifications of the *MYC* locus, the *MYC* coding region and surrounding regulatory regions are frequently amplified in metastatic ccRCC (21). By integrating data from a functional CRISPRi screen and ATAC-seq analysis, we have now identified regulatory regions that promote *MYC* expression downstream of the PAX8-HNF1B pathway (new data in **Fig. 4d-g** and **Extended Data Fig. 10b-d**). Importantly, these regions support *MYC* expression in the immortalised renal epithelial HK-2 cells as well as metastatic ccRCC clones in which the *MYC* copy number is increased (new data in **Fig. 4g** and **Extended Data Fig. 10a,j**). Thus, the ccRCC metastasis-associated 8q21.3-q24.3 amplicons are also dependent on the PAX8-HNF1B axis for *MYC* expression and ccRCC proliferation, further supporting our model by which renal lineage factors are critical for oncogenic signalling downstream of cancer-associated genetic alterations.

“7. Line 246-247, could the authors add references or results to support “VHL restoration and consequent inhibition of HIF2A does not have a strong effect on the proliferation of ccRCC cells *in vitro*”.”

Reply 2.18. Thank you for the suggestion, we have added a reference for this sentence. The lack of a HIF2A-dependent proliferative phenotype *in vitro* is also evident in the data shown in **Extended Data Fig. 5a-b**.

“8. HIF2A is intermittently referred to EPAS1 (e.g. Extended Fig10f right panel), it would be better to be consistent.”

Reply 2.19. Thank you for pointing this out, we have changed EPAS1 for HIF2A in the figure legend (new **Extended Data Fig. 6h**) and elsewhere in the manuscript for consistency.

“9. Line 491-495, could the author describe in more detail the calculation of dependency score.”

Reply 2.20. Thank you for pointing this out, we have clarified this in the methods section.

Referee #3 (Remarks to the Author):

“Patel *et al* present a CRISPR-screen to identify key factors in two ccRCC cell lines, and identify the two previously known lineage factors (in the cell of origin of ccRCC), PAX8 and HNF1B. It was shown previously that these factors are needed to transdifferentiate fibroblasts into renal tubular epithelial cells (Kaminski *et al*. NCB 2016). Knock-down of these factors causes cells to arrest in cell cycle, and to lose their lineage transcriptional program (i.e., the lineage gene regulatory network collapses). Next they show by ATAC-seq that the chromatin accessibility of lineage specific regulatory regions in the genome depend on these factors (this is expected, given the previous point).”

Reply 3.1. We agree that our data demonstrates reduced accessibility at chromatin regions marking the renal lineage upon PAX8 or HNF1B depletion. However, we do not observe a “collapse” in the lineage gene regulatory network. In other words, PAX8 and HNF1B depletion leads to a significant reduction at a number of gene regulatory elements, but our data show little evidence of complete loss of accessibility at gene regulatory elements with high baseline accessibility (**Fig 1g,2c, 3a** and **Extended Data Fig. 3c-d**). This is important, as it suggests that the phenotypic changes following lineage factor inhibition in this context are not caused by a large-scale destruction of the gene regulatory chromatin landscape, but rather by more subtle changes, indicating that specific gene responses, not a wholesale collapse of cell identity, is important. Indeed, as discussed under the following point below, our data suggest that the *in vitro* phenotype of PAX8-HNF1B reflects effects of HNF1B on *MYC* expression through specific enhancers. Similarly, PAX8 loss leads directly to reduced *CCND1* expression through a specific distal enhancer element.

“Upon PAX8/HNF1B knock-down, as we see cell cycle arrest, *MYC* targets are down-regulated.”

Reply 3.2. We agree that PAX8/HNF1B knockdown leads to cell cycle arrest and inhibition of *MYC* targets. However, it is important to clarify that based on our new data the PAX8-HNF1B axis regulates *MYC* expression through specific *MYC* enhancers, not through a collapse of the overall chromatin activation landscape. Even though PAX8 and HNF1B contribute to chromatin accessibility at partially overlapping genomic regions, reprogramming of cells into renal epithelial cells requires, in addition to *EMX2* and *HNF4A*, both PAX8 and HNF1B (43).

The renal lineage identity, and the corresponding chromatin accessibility landscape, is thus determined by the combinatorial activity of all four factors. In contrast, HNF1B alone is sufficient to rescue the antiproliferative effects (**Fig. 4a** and **Extended Data Fig. 8f**) and *MYC* expression (new data in **Fig. 4c**) upon PAX8 inhibition. This suggests that maintaining the renal lineage may not be the critical reason for the PAX8-HNF1B axis being required for ccRCC cell proliferation, but rather the pathway promotes the expression of *MYC* more directly.

The revised manuscript contains new data that gives significant new insight into the role of the PAX8-HNF1B axis in *MYC* regulation. We have conducted a proliferation-based CRISPRi screen to first identify relevant *MYC* enhancers (new data in **Fig. 4d**). Combining these data with chromatin accessibility and ChIP-seq data we have identified two enhancer elements that (i) are needed for ccRCC proliferation (new data in **Fig. 4d**), (ii) show reduced accessibility upon HNF1B and PAX8 depletion by RNAi (new analysis in **Fig. 4f** and **Extended Data Fig. 10d**), (iii) show evidence of HNF1B binding by ChIP-seq (new analysis in **Fig. 4e**), and (iv) contain the predicted HNF1B binding motif. Moreover, CRISPRi-based targeting of these two enhancers led to reduced *MYC* expression (new data in **Fig. 4g**). These data support a model whereby the PAX8-HNF1B pathway supports *MYC* expression through HNF1B-mediated effects at selected distal *MYC* enhancer elements.

As discussed further in our Reply 3.10 below, we find that these two *MYC* enhancers are also important for *MYC* expression in the non-cancerous HK-2 renal epithelial cells. This means that the PAX8-HNF1B-*MYC*-dependent proliferative phenotype is conserved between proliferating normal and cancerous cells of the proximal renal tubular origin. It thus reflects a physiological programme co-opted by cancer cells, and further enhanced in advanced ccRCC clones through the ccRCC metastasis-associated 8q21.3-q24.3 amplifications.

“For the RIME (protein complexes on chromatin) and ChIP-seq experiments, the authors target HIF2A, to investigate whether HIF2A co-binds to the PAX8/HNF1B enhancers. With the protein interaction data they show that HNF1B and PAX8 are in the HIF2A interactome. With the ChIP-seq data, they show that a subset of HIF2A and PAX8 binding sites overlap. Next, 175 HIF2A binding sites are targeted by dCas-KRAB in a pooled assay, using xenotransplantation, and identified a region upstream of CCND1 as co-bound by PAX8 and HIF2A, and repressing this enhancer leads to reduced tumour formation (due to lower CCND1 expression?). Finally, the authors describe a SNP in this enhancer that affects the PAX8 motif (risk allele has a stronger PAX8 site, compared to the rarer protective allele, that has a weaker PAX8 site). This study integrates multiple assays and circumstantial evidences showing how PAX8 and HNF1B maintain ccRCC lineage identity while promoting cancer growth (e.g., via MYC); how the lineage factor PAX8 facilitates HIF2A-dependent up-regulation of CCND1; and that rs7948643 affects ccRCC risk by changing that particular PAX8 binding site. Given that only few cis-regulatory variants are functionally/mechanistically understood, this study is timely and may fuel new studies to decipher the role of other cis-regulatory SNPs in cancer.”

Reply 3.3. We appreciate the positive assessment of our study. As described in our responses to the specific comments below, the revised manuscript includes significant new data sets that further clarify and support the model emerging from our work.

“Comments

1. Their key finding is that rs7948643 affects PAX8 binding, this in turn affects the regulation of CCND1 by HIF2A, thereby connecting the regulatory program of the cell type of origin (PAX8) with cancer-related TFs (HIF2A). That rs7948643 changes PAX8 binding is only shown by an EMSA assay, while this is a central element in this study. Is it possible to use the PAX8 ChIP-seq data to verify whether binding is biased to the risk allele? In a heterozygous context, both PAX8 binding and chromatin accessibility (ATAC-seq signal) are expected to show allelic imbalance (only the allele with the common allele, and the strong PAX8 binding site, should be bound by PAX8.”

Reply 3.4. We agree that based on our model PAX8 binding should favour the risk allele at rs7948643. In line with the EMSA data, which we have expanded and quantified (new data in **Fig. 3g**), ChIP-qPCR analysis of a *VHL* mutant ccRCC cell line RCC-JF, which is heterozygous for rs7948643, demonstrated that PAX8 binding is stronger for the T allele also in the chromatin context (new data in **Fig. 3i**). We also find that PAX8 depletion reduces HIF2A binding at E11:69419 but not vice versa (**Extended Data Fig. 7i-j**). In line with this, HIF2A binding is also stronger at the risk allele of E11:69419 (**Fig. 3i**).

We also agree that one possibility arising from these results is that chromatin accessibility could also be higher at the risk allele. However, even though PAX8 leads to reduced chromatin accessibility at PAX8-bound enhancers, the magnitude of the effect is often subtle and we do not commonly observe complete loss of accessibility, especially at loci with high accessibility, as is the case with the rs7948643-containing *CCND1* enhancer E11:69419 (**Fig 1g, 3a and Extended Data Fig. 3c-d**). This suggests an alternative possibility of layered epigenetic regulation, by which the initial accessibility at E11:69419 and other highly accessible PAX8-bound ccRCC enhancers may be controlled independently of PAX8, the role of PAX8 being critical for transcriptional activation. To distinguish between these possibilities, we have used ATAC-seq to analyse RCC-JF cells. As expected, E11:69419 is highly accessible, but we do not observe biased accessibility (new data in **Extended Data Fig. 7c,f**). Importantly, this result was confirmed in human ccRCCs, in which three out of four heterozygous cases showed balanced accessibility (new analysis in **Extended Data Fig. 7c**). To further explore the activation of E11:69419, we used the renal epithelial cell line HK-2, which expresses endogenous PAX8, with and without an exogenous transgene that encodes a *VHL*-insensitive HIF2A protein. With HIF2A, but not without, these cells activated E11:69419 in reporter assays (new data in **Extended Data Fig. 7d-e**), indicating that with HIF2A these cells have what is needed for transcriptional activation of the E11:69419 sequence from accessible DNA. However, even after 6 weeks and 40 population doublings, the HIF2A-expressing cells did not show increased chromatin accessibility at the endogenous E11:69419 locus (new data in **Extended Data Fig. 7f**). In line with these data, HIF2A expression does induce *CCND1* expression in HK-2 cells (new data in **Extended Data Fig. 7g**). Collectively these results suggest that PAX8 and HIF2A, while necessary for E11:69419 transcriptional activation, are not sufficient to make E11:69419 initially accessible. This is an important clarification and it highlights the layered nature of epigenetic regulation that underlies tissue-specificity in carcinogenesis.

“If phasing is possible, then also the expression of CCND1 at that same allele should have higher CCND1 expression. If phasing is not possible, then at least the CCND1 expression is expected to be allele-specific?”

Reply 3.5. We agree, this is an important point. We have now used long-read whole genome sequencing to phase the *CCND1* locus haplotypes in the heterozygous RCC-JF cells. We find that the risk allele at rs7948643 is linked to the allele A at the heterozygous SNP rs7177 in the *CCND1* 3'UTR (new data in **Fig. 3h and Extended Data Fig. 7h**). As predicted, using allele specific qRT-PCR, we detected higher expression of the *CCND1* allele with an A at rs7177, when compared to the allele linked to the protective variant C at rs7948643 (new data in **Fig. 3j**). Importantly, PAX8 depletion in the RCC-JF cells led to strongly biased reduction in the expression of the two *CCND1* alleles, with stronger reduction seen for the allele linked to the risk variant (**Fig. 3k and Extended Data Fig. 7k**). These results confirm the contribution of PAX8 to the allele-specific activation of the *CCND1* locus.

“2. Does HIF2A activation, upon PAX8 depletion, no longer induce CCND1 expression?”

Reply 3.6. We agree that this is an important point. Two lines of evidence suggest that both PAX8 and HIF2A are needed to support *CCND1* expression by E11:69419. First, both PAX8 depletion and *VHL* restoration, which depletes HIF2A, lead to a similar reduction in *CCND1* expression (**Fig. 2j and Extended Data Fig. 6l**), and combining *VHL* restoration with PAX8

depletion does not further reduce *CCND1* expression (**Fig. 2j**). We have now confirmed that HIF2A, not HIF1A, regulates *CCND1* in ccRCC cells (new data in **Extended Data Fig. 6h-k**). Second, in reporter assays mutations in the PAX8 and HIF2A binding sites have a similar effect on the enhancer activity of E11:69419, and we do not observe further inhibition by combined mutations (**Fig. 3c**). The reporter activity is also specifically dependent on HIF2A (new data in **Extended Data Fig. 7a**). These data suggest that in the absence of PAX8 endogenous HIF2A does not induce *CCND1* expression in ccRCC cells. In addition, even though HIF2A expression in the PAX8 expressing HK-2 cells is capable of activating the E11:69419 sequence in an accessible DNA construct, this does not lead to E11:69419 accessibility in chromatinised DNA or *CCND1* induction (new data in **Extended Data Fig. 7d-g**). Collectively, these results support a model whereby PAX8 and HIF2A are both needed for E11:69419-dependent induction of *CCND1* expression.

“3. It is important to show the ChIP-seq (both HIF2A and PAX8) and ATAC-seq signals as bigWig tracks in the main figures for additional loci, such as PAX8 itself, and the rs7948643 locus all the way to CCND1.”

Reply 3.7. We agree that this is an important suggestion. We have added genome tracks for the indicated and other relevant loci in **Fig. 1g, 2c, 3a, 4e** and **Extended Data Fig. 4b**.

“4. The “hypothesis” that lineage transcription factors are essential determinants of oncogenic phenotypes (p3) is a bit over-stated. Lineage TFs are maintained in most cancer types, this is more the rule than the exception. Androgen receptor, estrogen receptor, SOX10/MITF in melanoma, many examples in leukemia, etc. Perhaps the authors can provide more nuance, discussing more examples, how this is a common feature in cancer, rather than “we are testing this novel hypothesis”. The novel question can be phrased more precisely? (e.g., how lineage TFs co-control cancer-related genes?).”

Reply 3.8. We agree that lineage factors are important in many cancers, including those listed by the referee. However, whether and how lineage factors interact with cancer type-specific oncogenic mutations, and how this may lead to the establishment of cancer type-specific oncogenic programmes remains poorly understood. In this work we have tested whether oncogenic signalling downstream of cancer-associated genetic alterations is dependent on tissue-specific lineage factors, not whether lineage factors as such are important. We have revised the text in the introduction to clarify this point as suggested.

“Also, knock-down of these lineage factors in other cancer types (e.g., depletion of SOX10 in melanoma cells) has been shown repeatedly to cause cell cycle arrest, and induction of EMT-like genes, thus presenting a highly similar process to the current study (depletion of the lineage TFs causes collapse of the lineage chromatin state, cell cycle arrest, and induction of a stress-like state). For this particular aspect, the current study is not that novel, and some nuance or context can be provided (more novelty lies in the characterisation of the SNP, and linking that with the lineage program). Perhaps the up-regulated genes can even be compared with the up-regulated genes upon SOX10 depletion in melanoma cells (or similar data sets for other cancer types with clear lineage TFs).”

Reply 3.9. We agree that SOX10 in melanoma provides an interesting comparison point for our present work, in particular because melanomas commonly depend on activating mutations in *BRAF*, which show relative specificity for melanomas, even if not as strongly as *VHL* mutations for ccRCCs. Indeed, melanoma proliferation and tumour formation have been linked to SOX10 in many studies (44–47). Interestingly, recent evidence suggests that the role of SOX10 in melanoma is to specify a developmental stage in the neural crest-melanoblast-melanocyte trajectory in which mutant *BRAF* is capable of cellular transformation and tumour initiation (48). These data support the hypothesis that the role of lineage factors in cancer is to establish a permissive cellular state for oncogenesis. However, this hypothesis does not

provide answers to the question of why a particular cellular state is required. In other words, SOX10 maintains a stem-like state which is compatible with melanoma-induction by *BRAF* activation, but why can mutant *BRAF* only induce tumours when SOX10 is present is still unclear. The fact that developmental programmes are determined by specific interaction between DNA and transcription factors with DNA binding-specificities, the answer to this question is likely to require an understanding of how the factors that establish a cellular state interact with oncogenic driver pathways at the level of DNA. Our data on the specific interaction between PAX8 and HIF2A at E11:69419, as well as the characterisation of the effects of rs7948643 on this enhancer provide detailed evidence of DNA-level interaction between lineage factors and oncogenic pathways. We agree that this represents the most interesting result from our work and we hope that the revised manuscript highlights it more clearly.

It remains unclear why a lineage chromatin state per se, beyond what is specifically needed for the activation of critical oncogenic programmes, would be needed for cancer. Thus, while lineage factor inhibition may, as suggested by the referee and demonstrated experimentally e.g. in melanoma (46), lead to a collapse in lineage transcriptional state and secondarily to cell cycle arrest, an alternative possibility is that the lineage factors, in addition to supporting a global transcriptional programme, maintain oncogenic programmes through specific genomic loci. Our data on the PAX8-dependent expression of two canonical oncogenes, *CCND1* and *MYC*, support the latter possibility. Moreover, while PAX8/HNF1B depletion reduces chromatin accessibility at specific enhancers, the effects are fairly subtle and do not lead to a whole-sale collapse of chromatin accessibility patterns (**Fig. 1g, 2c, 3a, 4e** and **Extended Data Fig. 4b**). Yet, we see very strong anti-tumour phenotypes upon PAX8/HNF1B inhibition (**Extended Data Fig. 1h**). Combined with the new data on E11:69419 transcriptional activation vs. accessibility at the endogenous locus in HK-2 cells (new data in **Extended Data Fig. 7d-g**), these results support the idea that the role of PAX8 and HNF1B in ccRCC is to facilitate the transcription of specific oncogenic target genes, such as *CCND1* and *MYC*. We have restructured and reworded the revised manuscript extensively to highlight these points. It should be noted that unlike what has been reported for melanoma, where SOX10 depletion can lead to widespread cell death (46), we do not observe significant cell death in ccRCC cells after PAX8 depletion. The role of lineage factors may thus vary between tumour types, an interesting avenue for work in future studies.

“5. Related to the previous point - would PAX8 also control CCND1 and MYC during development or maintenance of renal tubular epithelial cells, and does the expression in ccRCC cells constitute an up-regulation rather than a de novo activation?”

Reply 3.10. We agree, this is an interesting point. In the revised manuscript we present extensive data in support of the idea that PAX8 regulates *MYC* expression in normal proliferating renal epithelial cells, while PAX8-dependent *CCND1* expression seems to be ccRCC specific. First, new data show that PAX8 depletion reduces proliferation and *MYC* expression in the renal epithelial cell line HK-2 and organoids derived from normal human renal epithelial cells (new data in **Fig. 4i-j** and **Extended Data Fig. 10e-h**). Through CRISPRi-based proliferative screening, we have identified two distal *MYC* enhancers that support ccRCC proliferation, bind HNF1B, show reduced accessibility upon HNF1B depletion and contain the HNF1B motif (new data in **Fig. 4d-f**), and targeting these enhancers by CRISPRi reduces *MYC* expression in HK-2 and ccRCC cells (new data in **Fig. 4g** and **Extended Data Fig. 10j**). Finally, analysis of chromatin accessibility data from the Meuleman et al. DHS atlas (49) suggest that these two enhancers are active in cells of the renal epithelial lineage and RCCs (new analysis in **Fig. 4h**). This physiological PAX8-HNF1B-MYC regulatory axis is co-opted by cancer cells and further enhanced by ccRCC metastasis-associated 8q21.3-q24.3 amplicons (new data in **Extended Data Fig. 10a**) (21). On the other hand, PAX8 inhibition does not reduce *CCND1* expression in HK-2 cells or organoids (new data in **Extended Data Fig. 10i**) and E11:69419 is not active in cells from foetal or normal renal lineage (new analysis in **Fig. 2f**), including the HK-2 cells in which even prolonged expression of HIF2A, while

capable of activating E11:69419 from a reporter construct (new data in **Extended Data Fig. 7d-e**), is not able to induce *CCND1* expression or accessibility at the endogenous E11:69419 locus (new data in **Extended Data Fig. 7f-g**). ccRCC-specificity of PAX8-dependent *CCND1* expression, at least by the E11:69419-mediated mechanism described in this study, is also supported by the fact that E11:69419 activity and *CCND1* expression is dependent on the co-activity of PAX8 and HIF2A, which is normally not present in renal epithelial cells.

“6. Also related: line 121 concludes that PAX8 and HNF1B maintain “cancer-specific” chromatin accessibility patterns; I would argue that they maintain lineage specific chromatin accessibility, as shown by the data presented. Otherwise, more comparisons are needed to discriminate normal accessibility in renal cells, versus cancer-specific regions. The statement on line 139, that PAX8 and HNF1B regulate proximal tubule cells lineage (and that their induction can create such cells) should be moved to the very beginning, as this is essential information to interpret the first experiments/data.”

Reply 3.11. We agree, the term “cancer-specific” is not correct and it was not our intention to suggest cancer specificity in this context. We have revised the text accordingly. As discussed in detail under Reply 3.12 below, new analyses suggest that PAX8 and HNF1B support the activity of both renal lineage-specific and renal cancer-specific enhancers. We have moved the reference to renal reprogramming to the beginning of the results section as requested.

“7. PAX8 + HNF1B are argued to maintain a ccRCC chromatin state. However, it is more likely that they maintain a renal cell chromatin state - as the cancer state strongly reflects the cell type of origin (in many cancer types). How different is the chromatin state between normal renal tubular epithelial cells and ccRCC cells? Are the PAX8+HNF1B binding sites (accessible regions) the same or not? The recent Meuleman et al. DHS atlas may come to rescue here.”

Reply 3.12. This is an important point and the suggestion to use the Meuleman et al. DHS atlas (49) is excellent, thank you. The data set contains 733 DNase-seq samples from 439 cell types and states. The renal lineage is well-represented with 74 samples from foetal kidney, 14 from normal kidney and 10 from renal cancer. We find that the chromatin regions that show reduced accessibility upon PAX8 or HNF1B depletion in ccRCC cells have a statistically significant overlap with genomic regions that are characteristic of normal renal lineage ($P = 2.0 \times 10^{-117}$ and $P = 3.0 \times 10^{-10}$, for PAX8 and HNF1B, respectively), but this overlap is far less significant than what we observe for the renal cancer-specific regions in the same data set ($P < 1.0 \times 10^{-150}$ for both factors, new analysis in **Extended Data Fig. 3h**). In agreement, E11:69419 upstream of *CCND1* shows very limited evidence for activity outside the RCC context (new analyses in **Fig. 2f**), while E8:128132 and E8:128526 upstream of *MYC* show activation in both normal samples of the renal lineage and renal cancer (new analyses **Fig. 4h**). We have revised the text throughout to reflect these new results.

“8. Regions with increased accessibility after PAX8/HNF1B depletion show “motifs for other factors” : which motifs and for which factors? AP-1?”

Reply 3.13. We agree, this information should be included in the manuscript. The suggestion that motifs for AP-1 factors is interesting as it could reflect parallels of lineage factor inhibition in ccRCC and melanoma, where reduced SOX10 activity is associated with a cell state switch and drug resistance (45,46,50). However, we see no enrichment for AP-1 motifs in these genomic regions. The top motifs enriched in the regions that are upregulated upon PAX8 depletion are TEAD1, TEAD3 and GATA6, but the enrichment is far less significant than what we detected for the regions with less accessibility in PAX8 depleted cells (**Extended Data Fig. 3k**), making it difficult to interpret the significance of this result. It is of interest, however, that TEAD factors have been linked to reduced SOX10 activity in melanoma (45). The top motifs enriched in the regions with increased accessibility following HNF1B depletion are NFY, KLF1 and SP1, following closely with similar KLF and SP motifs (**Extended Data Fig. 3l**). These

regions are likely to represent indirect effects of PAX8/HNF1B inhibition, and while of great interest for future work, following up on these results is beyond the scope of this project.

“9. MYC target genes are strongly inhibited in the PAX8/HNF1B depletion - this is normal, since the cells go in arrest.”

Reply 3.14. We agree that MYC target gene expression is expected to go down in cells that stop proliferating. However, why developmental lineage factors, which are highly expressed in normal tissues, should be required for cancer cell proliferation is far less clear, especially since cancers often are poorly differentiated and do not resemble faithfully their tissue of origin in terms of their histological appearance or proliferative phenotype. Our data supports a model by which ccRCC cells maintain their lineage characteristics at the chromatin level at least partially because PAX8 is specifically required for the expression of two canonical oncogenes, *CCND1* and *MYC*. While cancer-specific genetic alterations are also important for *CCND1* and *MYC* activity (i.e. *VHL* mutations and 8q21.3-q24.3 amplifications, respectively), they are dependent on PAX8 for oncogenic signalling. Based on this model, maintaining the renal lineage in itself is not the critical reason for PAX8 being important for advanced ccRCC clones, but rather, PAX8 and HNF1B are contributing to oncogenic signalling through specific distal enhancer elements.

“10. Line 207 concludes that PAX8 and HNF1B were now identified as drivers of ccRCC formation - but this is not shown and seems like an incorrect conclusion. These are the drivers of the cell type of origin, and maintain their expression in the cancer cell.”

Reply 3.15. We agree that these factors are lineage factors the expression of which is maintained in ccRCC. However, they are not simple remnants of the original cellular lineage. They are specifically needed for oncogenic signalling downstream of ccRCC-associated genetic alterations. As discussed in our responses above, this point, together with significant new data sets, is now more clearly presented across the manuscript. We have also removed the specific sentence the referee is referring to from the revised version of the manuscript.

“11. 205 HIF2A dependent transcripts are found, with 175 HIF2A binding sites in 500kb. Can the authors randomly select 205 expressed genes, and calculate how many binding sites are found, and repeat that 100 times? This would give an empirical p-value. It seems quite likely to find this number of binding sites.”

Reply 3.16. We agree that this is an important point. We can confirm that there is a statistically significant enrichment of HIF2A binding sites in the vicinity of the HIF2A transcriptional targets identified *in vivo*. The number of HIF2A peaks within 500kb flanking the transcription start sites of 1,000 random sets of 205 genes ranged from 46-115, which is less than the 175 peaks observed for the HIF2A target gene set, giving us an empirical p-value < 0.001. This result is clarified in the revised text.

“12. Line 272: ‘several’ HIF2A bound enhancers were depleted in the dCas-KRAB assay: how many exactly, and is this significant? Is there a relationship between depletion and HIF2A sites or PAX8 sites?”

Reply 3.17. We agree that this is an important point. We have now calculated permutation-based empirical p-values for each target region interrogated in the *in vivo* screen. We used 10,000 permutations of the normalized sgRNA depletion scores (i.e. normalized counts in tumour/normalized counts in plasmid) for each tumour, followed by calculation of the average depletion based on the top two constructs for each target region across the whole tumour cohort. This analysis identified 21 enhancers with an empirical p-value < 0.01. All known essential genes also had an empirical p-value < 0.01. The p-values are now reported using colour coding in **Fig. 2b**. We find evidence of PAX8 binding at 148/175 target regions of the

screen and 16/21 of the significant hits of the screen. There is no statistical difference between these groups in terms of PAX8 binding ($P = 0.33$, Fisher's exact test). Given the high fraction of PAX8 positive sites within the target set, it is not surprising that we observe no association between sgRNA depletion and PAX8 binding within this set of enhancers.

"13. Does PAX8 knock-down affect CCND1 up-regulation by HIF2A, in an allele-specific manner (when the heterozygous line for rs7948643 is used)?"

Reply 3.18. Yes, it does. To phase the haplotypes at the *CCND1* locus, we have now used long-read whole genome sequencing in the heterozygous RCC-JF cells (new data in **Fig. 3h** and **Extended Data Fig. 7h**). As predicted, PAX8 depletion led to strongly biased reduction in the expression of *CCND1* from the risk allele (new data in **Fig. 3k**).

"14. Is CCND1 an endogenous target of PAX8 during development (most parsimonious explanation), and HIF2A hijacks this interaction to cause increased CCND1 expression?"

Reply 3.19. We agree that this is an interesting question. We find that PAX8 depletion does not reduce *CCND1* expression in HK-2 cells or organoids (new data in **Extended Data Fig. 10i**) and E11:69419 shows very limited accessibility in cells from foetal or normal renal lineage (new analysis in **Fig. 2f**), including the HK-2 cells in which even prolonged expression of HIF2A, while capable of activating E11:69419 from a reporter construct (new data in **Extended Data Fig. 7d-e**), is not able to induce *CCND1* expression or accessibility at the endogenous E11:69419 locus (new data in **Extended Data Fig. 7f-g**). ccRCC-specificity of PAX8-dependent *CCND1* expression, at least by the E11:69419-mediated mechanism described in this study, is also supported by the fact that E11:69419 activity and *CCND1* expression is dependent on the co-activity of PAX8 and HIF2A (**Fig. 2j** and **3c**), which is normally not present in renal epithelial cells. Thus, several lines of evidence suggest that PAX8-dependent *CCND1* expression may be specific to ccRCC, but we cannot rule out the possibility that under some circumstances during renal development E11:69419 may contribute to *CCND1* expression in a PAX8-dependent manner.

"Also, CCND1 might also be high in ccRCC that has no HIF2A activation. Is CCND1 overall higher in tcga samples with HIF2A activation? Or simply, are HIF2A and CCND1 correlated across tcga samples (and is that stronger than pax8-ccnd1 correlation)?"

Reply 3.20. We agree, this is an interesting point. As predicted, we can confirm that the expression of *HIF2A* and *CCND1* correlate strongly in the TCGA data set (Pearson's correlation coefficient (PCC) = 0.54; $P = 1.9 \times 10^{-37}$), whereas the correlation between *PAX8* and *CCND1* expression is significantly weaker (PCC = 0.12, $P = 0.005$). These results give additional clinical support to our model and they have been added into the revised manuscript (**Extended Data Fig. 6o**).

Referee #4 (Remarks to the Author):

"In the manuscript "Lineage factor-dependent signalling by kidney cancer-associated genetic alterations" Patel et al. identify PAX8 and HNF1b as lineage specific transcription factors that guide oncogenic signaling in clear cell renal cancer. They use a CRISPR Cas9 library to screen for transcription factors relevant for renal tumor growth in xenograft models. The two most kidney specific factors PAX8 and HNF1b are examined further by knock-out cell models and genome-wide screens using ATAC-seq. Through this and by RNAseq of PAX8 and HNF1b depleted RCC cells the authors show that sites positively regulated by PAX8/HNF1b are enriched in kidney cancer relevant open chromatin regions. They follow up HNF1b and MYC as downstream targets of PAX8. Having observed a hypoxia signature in PAX8 depleted cells, in a second set of experiments the authors test for regulatory elements important for

HIF2A signaling. RNA-seq from RCC xenografts with and without HIF2A and HIF2A ChIPseq from xenografts define sites regulated by HIF2A. By this the authors retrieve a known enhancer of CCND1 expression, which is harboring the most significant RCC GWAS hit so far and has been linked to regulation of CCND1 expression by various groups before. From PAX8 and HIF2A RIME analysis and ChIPseq data it appears that both signaling pathways seem to overlap. The authors can also define a hypoxia signature in the PAX8 depleted RCC cells suggesting that both pathways have common targets. This is evident at the CCND1 enhancer at which PAX8 DNA interaction might be dependent on the presence of the risk allele for rs7948643 possibly mediating CCND1 regulation in RCC.”

Reply 4.1. We agree with this overall summary of our experimental work that recognises the discovery of renal lineage factors as essential components of oncogenic signalling downstream of ccRCC-associated genetic alterations. Indeed, this discovery is novel and as described below, extensive new data sets in the revised manuscript further solidify our conclusion.

“The manuscript describes important findings for renal cancer biology. Lineage specific effects must contribute to the tissue specific development of cancers, but so far evidence for that especially in the kidney is lacking. Examples from other tissues exist such as TRPS1 for breast cancer (Witwicki et al. Cell Rep. 2018 Oct 30; 25(5): 1255–1267.e5. or SOX proteins in gastric cancer (Francis R. et al. Science Advances 11 Dec 2019:Vol. 5, no. 12, eaax8898 DOI: 10.1126/sciadv.aax8898). In addition, overexpression of PAX8 and PAX8 mediated chromatin changes have been linked to the development of various malignancies including ovarian cancer (Chaves-Moreira D. et al. Cancer Res. 2020 Dec 23;canres.3173.2020.doi: 10.1158/0008-5472.CAN-20-3173).”

Reply 4.2. We agree, lineage must contribute to the development of cancers. This is evident from multiple lines of evidence, including human cancer genetics, large-scale cancer cell line screens and specific examples such as those highlighted by the referee but also others. The goal of our work was therefore not to test whether lineage factors are also important in ccRCC, but to build on the previous data on lineage factors in cancer to test whether and how renal lineage factors influence oncogenic signalling by common ccRCC-associated genetic alterations. As described in our manuscript, ccRCC with its unique genetic composition provides an ideal cancer context for this work. Critically, none of the examples brought up by the referee addresses this fundamental question our work is focusing on, thus highlighting the novelty of our study. Indeed, we are also not aware of other studies from any cancer context that would have similarly tackled the question of interaction between cancer driver alterations and lineage factors. We hope that the revised manuscript with extensive new data sets more clearly highlights the conceptual advances made by our work.

“The authors use elegant experimental approaches to define PAX8 as a lineage specific factor potentially contributing to renal cancer development. In fact, PAX8 immunohistochemistry already is used in clinical routine to define renal tubular origin of tumors in the kidney, since it is a marker for clear cell and papillary tumors. The experimental set up to generate and validate the data appears to be adequate and precise. Though for some of the experiments only technical triplicates have been used, the experiments were conducted in many cell lines confirming the results. The manuscript is well written and data presentation is well intelligible. Selection of references is adequate and comprehensive.”

Reply 4.3. We appreciate the recognition of our experimental approach and rigorous validation of results in multiple experimental models. The revised manuscript contains significant new data sets which further strengthen the conclusion of our study.

“I have some comments on specific aspects of the manuscript:

ATAC analysis. PAX8 depletion and ATACseq: sites which show decreased accessibility upon PAX8 depletion are enriched in KIRC and KIRP specific ATAC sites (Figure 1e). For this analysis, were the PAX8 depleted sites derived from the renal cancer cells? If so, is it possible that the overlap between the two data sets is generated from the same tissue background rather than from specific PAX8 sites. How was this analysis done and what was the comparator?"

Reply 4.4. We agree that based on the analysis in **Fig. 1c** (previous Fig. 1e), which compared regions that showed reduced, but not eliminated, accessibility upon PAX8 depletion in ccRCC cells to those characteristically active in different tumour types in the TCGA data set, it is only possible to distinguish between different cancer types, not normal tissues. We have now expanded this analysis to contain also normal tissues from a collection of 733 DNase-seq samples from 439 cell types and states (49). The renal lineage is well-represented with 74 samples from foetal kidney, 14 from normal kidney and 10 from renal cancer. We find that the chromatin regions that show reduced accessibility upon PAX8 or HNF1B depletion in ccRCC cells have a statistically significant overlap with genomic regions that are characteristic of normal renal lineage ($P = 2.0 \times 10^{-117}$ and $P = 3.0 \times 10^{-10}$, for PAX8 and HNF1B, respectively), but this overlap is far less significant than what we observe for the renal cancer-specific regions in the same data set ($P < 1.0 \times 10^{-150}$ for both factors, new analysis in **Extended Data Fig. 3h**). In agreement, E11:69419 upstream of *CCND1* shows very limited evidence for activity outside the RCC context (**Fig. 2f**), while E8:128132 and E8:128526 upstream of *MYC* show activation in both normal samples of the renal lineage and renal cancer (**Fig. 4h**). We conclude that PAX8 and HNF1B operate partially through distal enhancer elements that characterise the renal lineage, but also through cancer specific regions that show limited accessibility in normal renal epithelium.

*"If PAX8 is necessary to promote oncogenic signaling in renal tumors why is it downregulated in tumors when compared to normal tissue? An overall reduction of PAX8 would also compromise accessibility to tumor promoting sites which is in contrast to the hypothesis presented by the authors and in contrast to the proposed effects of PAX8 in other tissues(compare *Chaves-Moreira D. et al. Cancer Res. 2020 Dec 23;canres.3173.2020.doi: 10.1158/0008-5472.CAN-20-3173*)."*

Reply 4.5. The clinical and experimental data show robust PAX8 expression in ccRCCs (**Extended Data Fig. 2a-c**) and PAX8 inhibition in ccRCC cells leads to consistent and widespread phenotypic changes (**Extended Data Fig. 1a-b,h**). Moreover, PAX8 inhibition in ccRCC reduces, but does not eliminate, chromatin accessibility at ccRCC enhancers, such as E11:69419 (**Fig. 1g, 3a** and **Extended Data Fig. 3c-d**), and reduced expression of several pro-tumorigenic target genes such as *MYC* and *CCND1* (**Fig. 2j** and **Extended Data Fig. 4h,9b**). It is therefore clear that ccRCC cells have sufficient levels of PAX8 for tumour-promoting functions. However, PAX8 is also a lineage factor that enforces the identity of renal epithelial cells, the full effect of which might not be optimal for tumorigenesis. Recent data suggest that in ccRCC genetic mechanisms may reduce, but not eliminate, PAX8 activity, and that this might be a requirement for ccRCC formation (51). This is perfectly in line with the slight reduction, but not elimination, of PAX8 expression in ccRCCs, as well as the significant amount of data in our study that demonstrates specific tumour-promoting functions for PAX8 in ccRCC. It is unclear why the referee thinks these observations, and the suggested functions of PAX8 in other contexts cited by the referee (52), would contradict our hypothesis.

"MYC regulation. The observation that the PAX8/HNF1b axis regulates MYC is of great interest, but the work lacks more mechanistic insights into this regulation. Does MYC expression (RNA or Protein) correlate with PAX8/HNF1b expression in ccRCCs? Is the regulation observed mediated via direct promoter binding? Does an RCC specific enhancer module exist which binds PAX8 or HNF1b?"

Reply 4.6. We agree that the precise mechanism through which the PAX8-HNF1B axis regulates *MYC* expression is of interest. As HNF1B expression can rescue *MYC* expression (new data in **Fig. 4c**) and the *in vitro* proliferative phenotype upon PAX8 depletion (**Fig. 4a** and **Extended Data Fig. 8f**), our data suggest that HNF1B supports *MYC* expression downstream of PAX8. In order to test this model, we have now conducted a proliferation based CRISPRi screen to identify relevant *MYC* enhancers (new data in **Fig. 4d**). Combining these data with chromatin accessibility and ChIP-seq data we have identified two enhancer elements that (i) are needed for ccRCC proliferation (new data in **Fig. 4d**), (ii) show reduced accessibility upon HNF1B and PAX8 depletion by RNAi (new analysis in **Fig. 4f** and **Extended Data Fig. 10d**), (iii) show evidence of HNF1B binding by ChIP-seq (new analysis in **Fig. 4e**), and (iv) contain the predicted HNF1B binding motif. Independent CRISPRi-based targeting of these two enhancers led to reduced *MYC* expression (new data in **Fig. 4g**). Importantly, we find that these two enhancers also support *MYC* expression in the non-cancerous HK-2 renal epithelial cells (new data in **Extended Data Fig. 10j**). Moreover, comprehensive whole genome sequencing analysis has identified large 8q21.3-q24.3 amplifications in association with ccRCC metastasis (21). This region covers the *MYC* coding region but also the regulatory regions now identified by us as HNF1B-dependent *MYC* enhancers. Interestingly, our new FISH data show that the metastatic 786-M1A cells carry six copies of the *MYC* locus (new data in **Extended Data Fig. 10a**), but they are still dependent on the PAX8-HNF1B axis for *MYC* expression (**Fig. 4c** and **Extended Data Fig. 9b**). We do not see significant PAX8 or HNF1B binding at the *MYC* promoter. The mechanism described here is conserved between normal renal epithelial cells and cancer, but several other factors are also known to contribute to *MYC* expression in cancer (53), including genetic amplifications. Our model thus does not suggest that a strong correlation should exist between *PAX8/HNF1B* and *MYC* expression in ccRCC. Indeed, no significant correlation was observed between *MYC* and *PAX8* or *HNF1B* mRNA levels ($P = 0.15$ and $P = 0.19$, respectively) in the TCGA RNA-seq data set. These data demonstrate that the ccRCC metastasis-associated genetic 8q21.3-q24.3 amplifications are dependent on the renal lineage factor module for *MYC* expression and proliferation, further highlighting the general concept arising from our work, i.e. that tissue-specific lineage factors are required for oncogenic signalling by cancer-associated genetic alterations. We have revised the text extensively to include these new data.

“CCND1 enhancer and CCND1 expression. KIRC-56592 is an established enhancer of CCND1 expression that has been detected in HIF ChIPseq experiments and comprehensive mapping of renal cancer associated enhancers (histone marks, FAIRE-seq) by various groups (Schodel et al. Nat Gen 2012 REF28, Yao et al. Cancer Discovery 2017 REF29). It has been physically linked to the CCND1 promoter using different chromatin capture technology (Schodel et al Nat Gen 2012, Platt et al. EMBO Rep. 2016 Oct;17(10):1410-1421.doi: 10.15252/embr.201642198) and tissue-specific HIF effects on CCND1 expression were described extensively in earlier reports. The SNPs described in the manuscript and identified by Purdue et al. (Nature Gen 2011 REF5) have been linked to HIF and CCND1 by Schodel et al. (REF28). This part of the current manuscript therefore confirms existing data, which is acknowledged in the manuscript. The observation that PAX8 contributes to CCND1 regulation is novel and may yield some interesting aspects for this enhancer.”

Reply 4.7. Here the referee lists important previous studies that have indeed partially inspired the present work. However, the remark fails to recognise that until our discovery the mechanism through which rs7948643 contributes to ccRCC risk had not been proposed. Moreover, previous work has also not functionally linked E11:69419 (updated nomenclature for KIRC-56592 for consistency) to *CCND1* expression and ccRCC growth, nor provided any clues to the question of how tissue-specificity of this locus is regulated. While we have reproduced some of the previous results as control experiments, an essential element of reproducible science, the findings of our study are novel, and they provide critical new understanding into the role of lineage factors as determinants of the oncogenic phenotypes

that arise downstream of cancer-associated genetic alterations such as rs7948643, VHL inactivation and common 8q21.3-q24.3 amplifications.

We appreciate that more evidence was needed to solidify the findings introduced in our previous submission regarding E11:69419 function. As described in detail in our responses to the specific comments below, the revised manuscript contains renal cancer subtype specific human GWAS data showing very significant association between rs7948643 and clear cell RCC (ccRCC), but not papillary RCC, providing strong genetic support for our model (new data in **Fig. 3I**). We also provide extensive new analyses of E11:69419, including long-read DNA sequencing-based haplotyping of the *CCND1* locus, which facilitated the demonstration of rs7948643 risk allele-specific effects of PAX8 on transcriptional activity (new data in **Fig. 3h-k** and **Extended Data Fig. 7h**). These data solidify the functional evidence for our model of lineage-factor dependent signalling at E11:69419 by the two most common ccRCC-causing genetic alterations in humans. Furthermore, we characterise the role of PAX8-HIF2A interaction as a tissue-specific activator of accessible E11:69419, where PAX8 supports HIF2A binding but not vice versa (new data in **Extended Data Fig. 7i-j**), not a pioneer function (new data in **Extended Data Fig. 7c-g**), thus establishing a model for epigenetically layered regulation of tissue-specific interaction between lineage factors and cancer driver mutations at E11:69419. We hope that the novel findings and conceptual advancement are more clearly presented in the revised manuscript.

“However, I wonder why relevant differences in CCND1 expression did not appear in PAX8 knock-out experiments in cells as reported in the comparison with the dependency scores (compare Figure 4a and 2e).”

Reply 4.8. The plots in **Fig. 4b** and **Extended Data Fig. 6a** (previous Fig. 2e and Fig. 4a, respectively) show gene dependency data, not gene expression. *CCND1* is a dependency in ccRCC cell lines, as is shown in **Extended Data Fig. 6a**. However, HNF1B does not regulate *CCND1* expression, hence it is not included in the plot shown in **Fig. 4b**, which is focusing on shared genes regulated by PAX8 and HNF1B. We have amended the relevant text to clarify this point.

“Though in the supplementary data there is some repression of CCND1 expression upon PAX8 depletion in RCC cells (suppl. Table 2), this is in contrast to the prominent changes observed by HIF2A manipulation (suppl. Table 6) and the effects observed in the reporter assays which suggest that PAX8 is relevant for HIF signaling at this site (Figure 4f).”

Reply 4.9. Based on qRT-PCR data, the effect of VHL restoration and PAX8 inhibition on *CCND1* expression in both ccRCC cell lines tested is very similar, and combined VHL restoration and PAX8 inhibition does not further reduce *CCND1* expression (**Fig. 2j**). Similarly, the effect of mutating the PAX8 and HIF2A sites within the E11:69419 sequence has a similar effect in reporter assays, and combining these mutations does not further reduce reporter activity (**Fig. 3c**). The effect of the HIF2A inhibitor is also similar (new data in **Extended Data Fig. 7a**). While the relationship between enhancer activation level and gene expression is not necessarily linear, our observation of similar effect sizes in both qRT-PCR and reporter assays are nonetheless compatible with each other. It is therefore unclear what the referee means by suggesting that our reporter and gene expression results are in contradiction. Directly comparing the RNA-seq data set on HIF2A withdrawal *in vivo* (**Supplementary Table 5**) to the RNA-seq data set on PAX8 depletion *in vitro* (**Supplementary Table 3**), as the referee has done, is not correct as the experimental conditions, cell lines, and data analyses approaches are different between those two data sets. In sum, comparable experiments show very similar effect sizes for PAX8 and HIF2A inhibition on *CCND1* expression and E11:69419 activity.

“Does loss of PAX8 influence HIF DNA interactions at this site? Do they directly interact at this site? This would complement the very nice data derived from the RIME experiments.”

Reply 4.10. We agree that direct protein-protein interaction is a possible mechanism through which two transcription factors can jointly mediate transcriptional activation. However, functionally relevant transcription factor interaction can take place at multiple different levels. First, transcription factors may be dependent on heterodimerization, as is the case with HIF2A and HIF1B/ARNT, both of which are required for the formation of a functional DNA binding domain (22). In such cases the interaction is typically strong and mediated by significant interactive surfaces on the two proteins (22). Second, two transcription factors can bind DNA co-operatively through a DNA-facilitated manner in which case only some amino acids in the collaborating factors interact directly (23). Third, transcription factors may bind DNA in a completely DNA-mediated yet co-operative manner, in which subtle transcription factor binding-induced alterations in the nearby DNA structure are thought to enhance the binding of the second factor (24,25). Systematic high throughput analysis suggests that the vast majority of co-operative DNA binding by transcription factors falls in this category and is not dependent on direct protein-protein interaction, even when the DNA motifs are overlapping (23). Finally, transcription factors may co-operate at a functional level within an enhancer by independently binding DNA but collaboratively regulating the activation of the enhancer. In contrast to co-operative binding, which in the vast majority of cases depends on the DNA motifs being <5bp from each other (23), this kind of functional enhancer level interaction between transcription factors can occur over any distance within the enhancer DNA sequence.

The RIME experiment, which is based on two steps of cross-linking, can detect interaction that is mediated through all these different mechanisms. Similarly, ChIP-seq experiments can detect any enhancer-level interaction as co-localization within chromatin. This is the benefit of RIME-based approaches and transcription factor co-mapping validation, since it reveals functional connections that are genuine, but might be missed by less sensitive traditional approaches. On the other hand, co-IP experiments without cross-linking are likely to only detect strong protein-protein interactions, such as that between HIF2A and HIF1B/ARNT (22). In fact, we know that oestrogen receptor (ER) activity in breast cancer (work from one of the co-authors, Prof. Jason Carroll) involves the pioneer factor FOXA1 in mediating ER-enhancer interactions. This is now a well validated functional interaction and the paradigm of nuclear receptor activity (26–29). Despite the fact that FOXA1 was discovered as an ER-associated protein 15 years ago, via ChIP-seq co-binding and (more recently via) RIME-based discovery (30), the Carroll lab is yet to successfully show an interaction by co-IP, attesting to the limitations of traditional biochemical approaches.

Given that most known co-operative DNA-binding interactions between transcription factors are dependent on close proximity and defined distances between their individual motifs on DNA (23), the distribution of the distances between PAX8 and HIF2A motifs within the PAX8/HIF2A co-bound enhancers can be used to infer the type of interaction they are likely to be engaged in. In other words, if PAX8/HIF2A co-bound enhancers depend on a specific orientation determined by PAX8 and HIF2A structures, we would expect there to be an enrichment of PAX8 and HIF2A motifs in the specific orientation and distance that would allow this interaction to happen. On the other hand, if we see no enrichment for a particular configuration, the interaction is likely to be mediated by DNA and shared protein complexes, but not through direct protein-protein interaction. New analysis in **Extended Data Fig. 4f** shows that the distance between PAX8 and HIF2A motifs in ccRCC enhancers ranges between 0-500 nucleotides with a continuum over the whole range and different orientations, the only exception being the HIF2A-PAX8 motif sequence with an 18bp distance, which is repeatedly observed in the LTR sequence of the common endogenous retrovirus ERV1. Interestingly, ERV1 expression has been recently linked to ccRCC progression (31,32). The sequence level data thus suggest that PAX8 and HIF2A are unlikely to interact directly at the protein level, and that their co-operative effects on transcription (**Fig. 2a,j** and **3c**) is most likely mediated by enhancer level or enhancer cluster level (2) interaction through DNA and shared protein complexes. In line with this, and in contrast to PAX8 interaction with transcriptional co-factors that do not directly bind DNA, such as PRDM3 that is recruited to chromatin by PAX8 in ovarian cancer (33), we did not detect direct interaction between PAX8 and HIF2A in ccRCC

cells by co-IP, while HIF2A showed interaction with HIF1B/ARNT, as expected (new data in **Extended Data Fig. 4g**).

The centre-to-centre distance between the functional PAX8 and HIF2A binding sites within E11:69419 is 18bp (**Fig. 3d**), but in a different orientation than what is seen for the frequent ERV1 site (**Extended Data Fig. 4f**). This suggests that PAX8 and HIF2A do not directly interact on E11:69419. However, we find that PAX8 depletion reduces HIF2A binding at E11:69419, but not vice versa (new data in **Extended Data Fig. 7i-j**). Overall our data suggest a model whereby the pro-tumorigenic PAX8-HIF2A interaction takes place at the level of shared chromatin complexes and transcriptional activation. We have clarified this point in the revised manuscript.

“Regarding CCND1 expression it would be of interest whether PAX8 regulates CCND1 expression in cell lines derived from proximal tubules with and without HIF (e.g. HK-2). To my knowledge HK-2 cells (and other proximal tubular derived cell lines) do not regulate CCND1 via HIF, despite the expression of HIF2A (and the expression of PAX8). If the hypothesis of the authors that PAX8 maintains chromatin accessibility at this site is correct, than PAX8 and HIF2A binding should be observed at this site already in non-cancer derived cell lines (with HIF2A expression).”

Reply 4.11. We agree that this is an interesting point. First, it is important to clarify that even though PAX8 inhibition leads to reduced chromatin accessibility at PAX8-bound enhancers, complete abrogation of accessibility is not typical, especially at loci with high accessibility (**Fig. 1g, 2c, 3a, 4e** and **Extended Data Fig. 3c-d**), as is the case with E11:69419. This suggests that the initial accessibility at E11:69419 and other highly accessible PAX8-bound ccRCC enhancers may be controlled independently of PAX8, the role of PAX8 being critical for transcriptional activation. To further explore this, we have generated ATAC-seq data from the rs7948643 heterozygous ccRCC cell line RCC-JF. As expected, E11:69419 is highly accessible, but we do not observe biased accessibility for the two alleles (new data **Extended Data Fig. 7c,f**). Importantly, this result was confirmed in human ccRCCs, in which three out of four heterozygous cases showed balanced accessibility (new analysis **Extended Data Fig. 7c**). Despite balanced accessibility, we detected biased PAX8 binding towards the risk allele at rs7948643 (new data in **Fig. 3i**), and critically, PAX8 depletion led to strong reduction in the expression of the *CCND1* allele linked to the risk allele at rs7948643 (new data in **Fig. 3k** and **Extended Data Fig. 7k**). These data suggest that activation, rather than accessibility, is the level at which PAX8 controls E11:69419.

To further explore the function of E11:69419 in non-ccRCC cell lines of the renal lineage, we have used the renal epithelial cell line HK-2, as suggested by the referee. These cells express endogenous PAX8, but do not stabilize HIF2A under normoxic conditions, because they have an intact *VHL* gene. We transduced HK-2 cells with an exogenous transgene that encodes a VHL-insensitive HIF2A protein. With HIF2A, but not without, these cells activated E11:69419 in reporter assays (new data in **Extended Data Fig. 7d-e**), indicating that with HIF2A these cells have what is needed for transcriptional activation of the E11:69419 sequence from accessible DNA. However, even after 6 weeks and 40 population doublings, the HIF2A-expressing cells did not show increased chromatin accessibility at the endogenous E11:69419 locus (new data in **Extended Data Fig. 7f**). E11:69419 also shows limited accessibility in cells from foetal or normal renal lineage (new analysis in **Fig. 2f**).

Based on these data, the prediction is that PAX8 or HIF2A do not regulate *CCND1* expression in HK-2 cells via E11:69419. Indeed, our experimental data show that PAX8 depletion does not reduce *CCND1* expression in HK-2 cells or normal human renal epithelial organoids (new data in **Extended Data Fig. 10i**). Also, expression of the VHL insensitive HIF2A mutant does not induce *CCND1* expression in HK-2 cells (new data in **Extended Data Fig. 7g**). Collectively these results suggest that PAX8 and HIF2A, while necessary for E11:69419 transcriptional activation, are not sufficient to make E11:69419 initially accessible. This is an important clarification and it highlights the layered nature of epigenetic regulation that underlies tissue-specificity in carcinogenesis.

“In addition, papillary renal cell carcinomas with FH mutations overexpress HIF2A (Pollard PJ et al. Cancer Cell. 2007 Apr;11(4):311-9.doi: 10.1016/j.ccr.2007.02.005.), but enhancer activity appears to be reduced in these tumors at this site when compared to ccRCC (Figure 3h). The KIRP ATAC data set appears to have a strong (maybe even stronger than KIRC) PAX8 signature (Figure 1e). Does PAX8 bind to this region in papillary RCCs?”

Reply 4.12. The TCGA KIRP data set (54) contains five samples with *FH* mutations, none of which were included in the TCGA ATAC-seq cohort. *FH* mutant papillary RCCs represent a rare subtype of papillary RCCs with unique molecular features. It is therefore not possible to draw conclusions on the role of *FH*, the possible consequent activation of HIF2A, and the regulation of E11:69419 in *FH*-deficient RCCs based on this data set. However, we have now used cell lines derived from papillary RCCs to test PAX8 binding at E11:69419. In agreement with the ATAC-seq data on papillary RCCs, which shows low accessibility at E11:69419 when compared to ccRCCs (**Fig. 2e**), we do not observe PAX8 binding at this locus in these cell lines (new data in **Extended Data Fig. 7I**). In line with the results on HK-2 cells discussed in the previous point, this suggests that PAX8 alone is not sufficient to make E11:69419 accessible, but rather, that the primary role of PAX8 at E11:69419 is to facilitate transcriptional activation in collaboration with HIF2A. To further test our model, which predicts ccRCC-specificity of the E11:69419 function we describe, we have now performed RCC subtype-specific association analysis in a large human RCC GWAS data set (12) that comprises 6,211 RCC cases for which the subtype is known as well as ~15,000 healthy controls. As predicted, we observe a very strong association between rs7948643 and ccRCC ($P = 2.4 \times 10^{-10}$), but no evidence of association for papillary RCC ($P = 0.9$). These data provide strong human genetic support for our model (new data in **Fig. 3I**).

“The authors point out that the enhancer might regulate the expression of genes other than CCND1. In fact, the enhancer has been shown to physically interact with the promoter of MYEOV (Platt J. et al Embo rep 2016) which is upstream of CCND1 and the enhancer and the second most downregulated gene in the HIF2A depleted xenografts (suppl. Table 6). Suggesting that regulation of this gene could be involved in (or even responsible for) the generation of the observed effects as well. Does PAX8 regulate the expression of MYEOV in a genotype specific manner?”

Reply 4.13. We agree, *MYEOV* is also a likely target gene of E11:69419 as its expression is dependent on both PAX8 and HIF2A. We have explored the role of *MYEOV* further in the revised manuscript. First, we find that E11:69419 inhibition reduces *MYEOV* expression (new data in **Fig. 2h**). Unlike *CCND1*, which has a heterozygous variant in the 3'UTR, *MYEOV* does not have suitable variants in the transcript, preventing the analysis of allele-specific expression in the heterozygous RCC-JF cells. However, exploration of CRISPR/Cas9-based screen data show that *MYEOV*, unlike *CCND1*, is not required for ccRCC cell proliferation *in vitro* (new analysis in **Extended Data Fig. 6a**). Furthermore, new data show that *MYEOV* depletion by lentivirally introduced stable RNAi has no effect on ccRCC growth *in vivo*, whereas *CCND1* inhibition has a strong effect on tumour formation (new data in **Fig. 2i** and **Extended Data Fig. 6b**). We conclude that while *MYEOV* is a PAX8 and HIF2A-dependent transcriptional target of E11:69419, it is not a major contributor to the pro-tumorigenic functions of these factors.

“Allele specific effects (Figure 4g-i). The authors use reporter assays and EMSA to examine binding of PAX8 to the different alleles of rs7948643. Though the data fit to the hypothesis that PAX8 preferentially binds to the risk allele, data from chromatinized DNA should be included, i.e. ChIP for PAX8 in cell lines heterozygous for the SNP. The minor allele frequency of this SNP will be low, but this experiment would strengthen the results a lot.”

Reply 4.14. We agree that this is an important point. We have now used RCC-JF cells which are heterozygous for rs7948643 to demonstrate by ChIP-qPCR that PAX8 favours the risk allele in the chromatin context (new data in **Fig. 3i**). We have also expanded our EMSA experiments with more replicates and quantification (new data in **Fig. 3g**). Moreover, we have used long-read whole genome sequencing to phase the *CCND1* locus haplotypes in RCC-JF cells (new data in **Fig. 3h** and **Extended Data Fig. 7h**) and tested allele-specific effects of PAX8 inhibition on *CCND1* expression. As predicted, PAX8 depletion led to biased reduction in the expression of the two *CCND1* alleles, with stronger reduction seen for the allele linked to the risk variant (new data in **Fig. 3k** and **Extended Data Fig. 7k**). These results confirm the contribution of PAX8 to allele-specific pro-tumorigenic effects of the rs7948643 locus at the chromatin and transcriptional levels.

1. Bleu M. et al. *Nat Commun* **10**, 3739 (2019)
2. Rodrigues P. et al. *Cancer Discov* **8**, 850–65 (2018)
3. Vanharanta S. et al. *Nat Med* **19**, 50–6 (2013)
4. Jacob L.S. et al. *Cancer Res* **75**, 3713–9 (2015)
5. Syafruddin S.E. et al. *Nat Commun* **10** (2019)
6. Turajlic S. et al. *Cell* **173**, 581-594.e12 (2018)
7. Patel S.A. et al. *Br J Cancer* **124**, 3–12 (2021)
8. Cho H. et al. *Nature* **539**, 107–11 (2016)
9. Choueiri T.K. et al. *Nat Med* **27**, 802–5 (2021)
10. Kaelin W.G. *Annu Rev Pathol Mech Dis* **2**, 145–73 (2007)
11. Shen C. et al. *Cancer Discov* **1**, 222–35 (2011)
12. Scelo G. et al. *Nat Commun* **8**, 15724 (2017)
13. Sun M. et al. *Cancer Res* **77**, 5313–26 (2017)
14. Gossage L. et al. *Nat Rev Cancer* **15**, 55–64 (2015)
15. Speisky D. et al. *Clin Cancer Res* **18**, 2838–49 (2012)
16. Lorenzo P.I. et al. *Histochem Cell Biol* **136**, 595–607 (2011)
17. Gijtenbeek J.M.M. et al. *J Neurooncol* **74**, 261–6 (2005)
18. Carney E.M. et al. *Am J Surg Pathol* **35**, 262–7 (2011)
19. Mitchell T.J. et al. *Cell* **173**, 611–23 (2018)
20. Turajlic S. et al. *Cell* **173**, 595–610 (2018)
21. Watkins T.B.K. et al. *Nature* **587**, 126–32 (2020)
22. Wu D. et al. *Nature* **524**, 303–8 (2015)
23. Jolma A. et al. *Nature* **527**, 384–8 (2015)
24. Klemm J.D. et al. *Genes Dev* **10**, 27–36 (1996)
25. Panne D. et al. *EMBO J* **23**, 4384–93 (2004)
26. Glont S.-E. et al. *Cell Rep* **26**, 2558-2565.e3 (2019)
27. Hurtado A. et al. *Nat Genet* **43**, 27–33 (2011)
28. Carroll J.S. et al. *Cell* **122**, 33–43 (2005)
29. Arruabarrena-Aristorena A. et al. *Cancer Cell* **38**, 534-550.e9 (2020)
30. Papachristou E.K. et al. *Nat Commun* **9**, 2311 (2018)
31. Smith C.C. et al. *J Clin Invest* **128**, 4804–20 (2018)
32. Zapatka M. et al. *Nat Genet* **52**, 320–30 (2020)
33. Bleu M. et al. *Nat Commun* **12**, 2442 (2021)
34. Heinz S. et al. *Nat Rev Mol Cell Biol* **16**, 144–54 (2015)
35. Schick S. et al. *Nat Genet* **53**, 269–78 (2021)
36. Vannam R. et al. *Cell Chem Biol* **28**, 503-514.e12 (2021)
37. Carico Z.M. et al. *PLoS Genet* **17**, e1009435 (2021)
38. Ochi Y. et al. *Cancer Discov* **10**, 836–53 (2020)
39. Platt J.L. et al. *EMBO Rep* **17**, 1410–21 (2016)
40. Smith J.C. et al. *bioRxiv*, 2021.06.01.446243 (2021)
41. Cerami E. et al. *Cancer Discov* **2**, 401–4 (2012)
42. Ricketts C.J. et al. *Cell Rep* **23**, 313-326.e5 (2018)
43. Kaminski M.M. et al. *Nat Cell Biol* **18**, 1269–80 (2016)

44. Shakhova O. et al. *Nat Cell Biol* **14**, 882–90 (2012)
45. Verfaillie A. et al. *Nat Commun* **6**, 6683 (2015)
46. Wouters J. et al. *Nat Cell Biol* **22**, 986–98 (2020)
47. Kaufman C.K. et al. *Science* **351**, aad2197 (2016)
48. Baggiolini A. et al. *Science* **373**, eabc1048 (2021)
49. Meuleman W. et al. *Nature* **584**, 244–51 (2020)
50. Shaffer S.M. et al. *Nature* **546**, 431–5 (2017)
51. Gu X. et al. *Cell Rep* **36**, 109747 (2021)
52. Chaves-Moreira D. et al. *Cancer Res* **81**, 806–10 (2021)
53. Meyer N. et al. *Nat Rev Cancer* **8**, 976–90 (2008)
54. Cancer Genome Atlas Research Network et al. *N Engl J Med* **374**, 135–45 (2016)

Reviewer Reports on the First Revision:

Referees' comments:

Referee #1 (Remarks to the Author):

I have reviewed the revised manuscript. In my opinion the authors have gone above and beyond in addressing the comments. This work represents a significant advance in the mechanism of renal tumorigenesis.

Referee #2 (Remarks to the Author):

The authors have answered all my comments thoroughly.

Referee #3 (Remarks to the Author):

The authors provide an extensive and high-quality revision that addresses most of our comments.

There is one statement that I disagree with:

"We find that the chromatin regions that show reduced accessibility upon PAX8 or HNF1B depletion in ccRCC cells have a statistically significant overlap with genomic regions that are characteristic of normal renal lineage ($P = 2.0 \times 10^{-117}$ and $P = 3.0 \times 10^{-10}$, for PAX8 and HNF1B, respectively), but this overlap is far less significant than what we observe for the renal cancer-specific regions in the same data set ($P < 1.0 \times 10^{-150}$ for both factors, new analysis in Extended Data Fig. 3h)."

The "far less significant" is strongly exaggerated. It doesn't seem possible to me to compare 10^{-117} with 10^{-150} .

Referee #4 (Remarks to the Author):

This is a revised version of a previous manuscript. The authors now include more experimental and genetic data to support the hypothesis that lineage specific factors co-operate with genetic alterations in the development of clear cell renal cell carcinoma. They exemplify this for the MYC locus and the HIF-associated CCND1 enhancer. The authors have restructured the manuscript and increased clarity.

I have only minor comments:

Line 844: Is this the correct reference for the GWAS?

Figure legends: some of the error bars are not defined (or this reviewer is unable to find it). E.g. Figure 1c. Figure 2 h,i,j. 3b,c.

Referee #5 (Remarks to the Author):

Patel and colleagues report that the lineage transcription factor PAX8 is required for oncogenic signaling by two common ccRCC-predisposing genetic alterations: the common germline variant rs7948643, and somatic inactivation of VHL. I have been asked to review the newly added evidence from the analysis of the human GWAS data, and my comments therefore pertain only to this aspect and the related inference. Overall, the support by human genetic evidence clearly represents an important addition to the manuscript. My main concern relates to the data presented for rs7177 in the 3' UTR of CCND1 and its interpretation, as detailed below. The methods and evidence for rs7105934, rs7948643 and rs7177 should also be presented differently, as detailed below.

1. Methods about human GWAS data in the manuscript: the current reference in lines 841 – 844 is incorrect, it should be reference 5 instead of 25. The Editors supplied a separate document with more detail than currently included in the manuscript (agree this is necessary). This additional document contains a description of what was done in the original publication by Scelo et al. Rather than describing this again, it would be more informative instead to know what the authors of this manuscript did with this published data. Specifically, please include whether this data is publicly available anywhere (e.g., download link or dbGaP application number) or whether these data are not in the public domain at all and can only be obtained from Drs. Purdue and Chanock? If the latter, then please present all summary statistics underlying Figure 3I as source data (variant identifier, reference and coded allele, coded allele frequency, odds ratio or effect estimate, standard error or 95% confidence interval, p-value, sample size, I2 statistic, imputation quality).

2. Please generate two regional association plots, one for ccRCC and one for papillary RCC, spanning the entire region including rs7105934, rs7948643 and rs7177. This will show the contrast for genetic risk for ccRCC and papillary RCC much better than comparing p-values for individual SNPs. It will also show where the signal is centered, and that the association does not extend into the CCND1 region. Of course, if the mechanism is that rs7105934 and rs7948643 alter the binding of a TF controlling expression of CCND1 in target tissue, then CCND1 can still be the effector gene.

3. It is not my expertise to assess the evidence supporting the selection of CCND1 as the likely target gene for ccRCC over MYEOV (Fig. 2i and Extended 199 Data Fig. 6a-b) or other genes in this region. My comment therefore only relates to the presentation of evidence supporting CCND1: in the largest human eQTL database, the GTEx Project, neither rs7948643 nor rs7105934 is associated with transcript levels of CCND1 in any of the 50 tissues studied. This could be explained by the lack of target tissue or the right conditions. The authors show with long-range sequencing and follow-up experiments that PAX8 preferentially supports CCND1 expression from the risk allele (as evidenced by an altered allelic ratio of a common CCND1 variant), but this does not show differences in overall levels of CCND1 abundance. Also, the authors only look at CCND1 and not any of the other genes in the locus, for which the RCC-JF cell line should also contain heterozygous variants detectable with long-range sequencing. This does not take away from the conclusion that rs7948643 affects PAX8 binding in the chromatin context, but it should at least be discussed that only CCND1 was assessed as a readout using the long-range sequencing data.

Author Rebuttals to First Revision:

Point-by-point response to referee comments

We would like to thank the referees for taking the time to review our work. We really appreciate the constructive input that has significantly improved our study. We have now addressed the final points in a revised manuscript. Our point-by-point responses are below. The main changes are:

1. Clarification of the GWAS method and data availability sections in the text and inclusion of GWAS source data.
2. Improved presentation of the GWAS signal at the E11:69419-CCND1 locus.
3. Adjustment of the conclusions about CCND1, PAX8 and rs7948643 by acknowledging the possibility that other pro-tumorigenic targets of E11:69419 may also be affected by the rs7948643 variants.

Referees' comments:

Referee #1 (Remarks to the Author):

"I have reviewed the revised manuscript. In my opinion the authors have gone above and beyond in addressing the comments. This work represents a significant advance in the mechanism of renal tumorigenesis."

Reply: We thank the referee for the positive assessment of our revised manuscript.

Referee #2 (Remarks to the Author):

"The authors have answered all my comments thoroughly."

Reply: We thank the referee for the positive assessment of our revised manuscript.

Referee #3 (Remarks to the Author):

"The authors provide an extensive and high-quality revision that addresses most of our comments."

Reply: We thank the referee for the positive assessment of our revised manuscript.

"There is one statement that I disagree with:

"We find that the chromatin regions that show reduced accessibility upon PAX8 or HNF1B depletion in ccRCC cells have a statistically significant overlap with genomic regions that are characteristic of normal renal lineage ($P = 2.0 \times 10^{-117}$ and $P = 3.0 \times 10^{-10}$, for PAX8 and HNF1B, respectively), but this overlap is far less significant than

what we observe for the renal cancer-specific regions in the same data set ($P < 1.0 \times 10^{-150}$ for both factors, new analysis in Extended Data Fig. 3h)."

The "far less significant" is strongly exaggerated. It doesn't seem possible to me to compare 10-117 with 10-150."

Reply: We agree, this statement, which is not from the manuscript but from Reply 3.12 in our point-by-point response to the original Referee #3 comments, is not correct. The text related to **Extended Data Fig. 3h** does not make this point, but rather states that "PAX8 and HNF1B inhibition reduced ... chromatin accessibility at genomic loci ... characteristic of the renal origin", taking into account significant overlap with both RCC and normal renal epithelium-associated chromatin regions.

Referee #4 (Remarks to the Author):

"This is a revised version of a previous manuscript. The authors now include more experimental and genetic data to support the hypothesis that lineage specific factors co-operate with genetic alterations in the development of clear cell renal cell carcinoma. They exemplify this for the MYC locus and the HIF-associated CCND1 enhancer. The authors have restructured the manuscript and increased clarity."

Reply: We thank the referee for the positive assessment of our revised manuscript.

"I have only minor comments:

Line 844: Is this the correct reference for the GWAS?"

Reply: Thank you for pointing this out, we have corrected the references in the revised section on GWAS methods.

"Figure legends: some of the error bars are not defined (or this reviewer is unable to find it). E.g. Figure 1c. Figure 2 h,i,j. 3b,c."

Reply: Thank you for pointing this out. We have revised the figure legends throughout the manuscript and added error bar and other relevant information.

Referee #5 (Remarks to the Author):

"Patel and colleagues report that the lineage transcription factor PAX8 is required for oncogenic signaling by two common ccRCC-predisposing genetic alterations: the common germline variant rs7948643, and somatic inactivation of VHL. I have been asked to review the newly added evidence from the analysis of the human GWAS data, and my comments therefore pertain only to this aspect and the related inference. Overall, the support by human genetic evidence clearly represents an important addition to the manuscript. My main concern relates to the data presented for rs7177 in the 3' UTR of CCND1 and its interpretation, as detailed below. The methods and evidence for rs7105934, rs7948643 and rs7177 should also be presented differently, as detailed below."

Reply: We appreciate the referee's positive assessment of the human genetic data in our study. We also agree that the methods section on GWAS and the presentation of the GWAS data required improvements, which we have now conducted as described below.

"1. Methods about human GWAS data in the manuscript: the current reference in lines 841 – 844 is incorrect, it should be reference 5 instead of 25."

Reply: Thank you for pointing this out, we have corrected the references in the revised section on GWAS methods.

"The Editors supplied a separate document with more detail than currently included in the manuscript (agree this is necessary). This additional document contains a description of what was done in the original publication by Scelo et al. Rather than describing this again, it would be more informative instead to know what the authors of this manuscript did with this published data."

Reply: We agree that this is an important clarification. The revised manuscript contains an expanded methods section for the GWAS analysis, describing the previously published RCC GWAS meta-analysis data set for reference and explicitly stating that for the current work RCC subtype-specific data was extracted to determine ccRCC and papillary RCC-specific risks associated with the variants of interest.

"Specifically, please include whether this data is publicly available anywhere (e.g., download link or dbGaP application number) or whether these data are not in the public domain at all and can only be obtained from Drs. Purdue and Chanock? If the latter, then please present all summary statistics underlying Figure 3I as source data (variant identifier, reference and coded allele, coded allele frequency, odds ratio or effect estimate, standard error or 95% confidence interval, p-value, sample size, I2 statistic, imputation quality)."

Reply: We agree that these are important clarifications. We have revised the Data Availability section to clarify that the complete RCC GWAS meta-analysis data set is available from Drs. Purdue and Chanock upon request and the original scans are available either from dbGaP or from the corresponding authors of the original data sets. We have also included a new data table that provides the source data for **Fig. 3I** (new **Extended Data Table 1**) as well as **Supplementary Tables 7-8** that contain the source data for the new regional association plots in **Extended Data Fig. 8**.

"2. Please generate two regional association plots, one for ccRCC and one for papillary RCC, spanning the entire region including rs7105934, rs7948643 and rs7177. This will show the contrast for genetic risk for ccRCC and papillary RCC much better than comparing p-values for individual SNPs. It will also show where the signal is centered, and that the association does not extend into the CCND1 region. Of course, if the mechanism is that rs7105934 and rs7948643 alter the binding of a TF controlling expression of CCND1 in target tissue, then CCND1 can still be the effector gene."

Reply: Thank you for the suggestion. We have added the two regional association plots in **Extended Data Fig. 8** with the source data available in **Supplementary**

Tables 7-8. The peak of the signal for ccRCC is at rs11263655, and the linked region contains both rs7948643 and rs7105934. The signal does not extend to *CCND1*. In line with our previous analyses shown in **Fig. 3i**, no signal is observed for papillary RCC.

*“3. It is not my expertise to assess the evidence supporting the selection of *CCND1* as the likely target gene for ccRCC over *MYEOV* (Fig. 2i and Extended 199 Data Fig. 6a-b) or other genes in this region. My comment therefore only relates to the presentation of evidence supporting *CCND1*: in the largest human eQTL database, the GTEx Project, neither rs7948643 nor rs7105934 is associated with transcript levels of *CCND1* in any of the 50 tissues studied. This could be explained by the lack of target tissue or the right conditions.”*

Reply: We agree with the referee’s assessment related to the GTEx data set. Based on our functional analyses, rs7948643 influences E11:69419 activity in the presence of both PAX8 and HIF2A in cells in which the E11:69419 locus is accessible. As the PAX8 and HIF2A proteins are expected to have restricted expression patterns across most normal tissues, and the E11:69419 locus is not universally accessible even in cells originating from the renal epithelium, it is unsurprising that the GTEx eQTL database does not show evidence for association between rs7948643 or rs7105934 and *CCND1* expression in the 50 tissues examined. As the referee suggests, the lack of correlation between rs7948643 genotypes and *CCND1* expression in the GTEx database is therefore likely to reflect the lack of right cell types and conditions for the association to be detectable. We have added the following statement in the Discussion to highlight this.

Line 473 now reads: “In line with the tissue-restricted accessibility of E11:69419 and expression of PAX8 and HIF2A, the rs7948643 genotype does not correlate with *CCND1* expression in most normal tissues⁴⁸.”

*“The authors show with long-range sequencing and follow-up experiments that PAX8 preferentially supports *CCND1* expression from the risk allele (as evidenced by an altered allelic ratio of a common *CCND1* variant), but this does not show differences in overall levels of *CCND1* abundance.”*

Reply: We agree, our data show preferential expression of *CCND1* from the ccRCC risk allele (**Fig. 3j**). Combined with the data showing preferential binding of PAX8 to the risk allele *in vitro* (**Fig. 3f-g**) and in the chromatin context (**Fig. 3h**), and the demonstration of preferential reduction of *CCND1* expression from the risk allele upon PAX8 depletion (**Fig. 3k**), our data support the idea that a PAX8-dependent mechanism leads to higher expression of *CCND1* from the risk allele. These results are well controlled as the expression of both alleles is assessed under identical cellular conditions. We also agree that testing whether the higher level of E11:69419 activity in the risk allele translates into overall higher *CCND1* expression would be of interest. Such analyses are, however, complicated by the uneven distribution of the different alleles at rs7948643, with the minor allele frequency being only 0.0737. As homozygous individuals for the minor allele are thus expected to represent only ~0.5% of the population, the currently available data sets, such as the TCGA ccRCC data set, are too small for robust eQTL analysis in the context of rs7948643 and *CCND1*,

in particular because the analysis would have to be performed in cancer samples, which display a lot of variability in transcript levels.

“Also, the authors only look at CCND1 and not any of the other genes in the locus, for which the RCC-JF cell line should also contain heterozygous variants detectable with long-range sequencing. This does not take away from the conclusion that rs7948643 affects PAX8 binding in the chromatin context, but it should at least be discussed that only CCND1 was assessed as a readout using the long-range sequencing data.”

Reply: We agree that *CCND1* may not be the only relevant target downstream of E11:69419. Chromatin conformation analysis has previously shown that of the known genes, E11:69419 interacts most strongly with the promoter proximal regions of *MYEOV* and *CCND1* (1). In addition, of the genes within the 1Mb of DNA flanking E11:69419, only *MYEOV* and *CCND1* are downregulated upon PAX8 and HIF2A inhibition in our data (**Supplementary Table 3** and **Supplementary Table 5**). We show that both *MYEOV* and *CCND1* are downregulated upon E11:69419 inhibition by CRISPRi (**Fig. 2h**). Furthermore, we show that PAX8 inhibition leads to allele specific downregulation of *CCND1* (**Fig. 3k**). We did not evaluate allele specific effects on *MYEOV* as we did not identify suitable variants in the *MYEOV* transcript. However, it is likely, based on our data, that *MYEOV* expression would also be regulated in an allele specific manner. We therefore fully agree with the referee that E11:69419 and rs7948643 are likely to have other target genes beyond *CCND1*. *MYEOV* is likely to be one, but there may be others.

We did not find evidence that *MYEOV* expression would be required for ccRCC formation in our experimental system *in vivo*, while *CCND1* inhibition strongly suppressed tumorigenesis (**Fig. 2i**). Similarly, *CCND1* knockout by CRISPR-Cas9 inhibits the proliferation of ccRCC cells, while no effect is detected for *MYEOV* (**Extended Data Fig. 6a**). We agree, however, that other experimental setups could reveal a pro-tumorigenic role for *MYEOV* or other yet unidentified E11:69419 target genes in ccRCC. To reflect this, we have revised the text as follows:

Line 261 now reads: “...*CCND1* is required for ccRCC proliferation and tumour formation *in vivo* (Fig. 2i and Extended Data Fig. 6a-b), making it a likely functionally important target gene of E11:69419 in ccRCC.”

Line 338 now reads: “...the ccRCC protective allele C at rs7948643 inhibits PAX8 binding, consequently reducing the activity of E11:69419 upstream of the oncogenic driver *CCND1* and other possible pro-tumorigenic mediators.”

Line 484 now reads: “...functional evidence that PAX8, HIF2A and E11:69419 regulate *CCND1* expression, possibly also other pro-tumorigenic targets,...

Reviewer Reports on the Second Revision:

Referees' comments:

Referee #5 (Remarks to the Author):

Patel and colleagues have worked on all of my comments. However, based on the newly added, very helpful regional association plots depicting the association of SNPs in the region with ccRCC and with papillary RCC, there is an additional observation that merits further clarification. While, as the authors state, there is no association between the ccRCC candidate SNPs with papillary RCC, the plot for papillary RCC shows genome-wide significant SNPs in the CCND1 region. Thus, based on the current plot, one would conclude that the region in fact contains different risk SNPs, some for ccRCC via the mechanism described in this manuscript, and some for papillary RCC more upstream. If so, I believe it would be interesting for readers to know that while the authors convincingly demonstrate the mechanism related to ccRCC, other germline variants in the region may relate to papillary RCC through different mechanisms. Please check with Dr. Chanock first about the papillary RCC data, as I did not find mention of the CCND1 locus for papillary RCC in the original publication, as one would expect for a genome-wide significant locus. Just to make sure the correct file was transferred for papillary RCC, and/or whether the transferred file needs further filtering. Also, please also label the ccRCC-related SNPs in the panel showing results for papillary RCC.

Author Rebuttals to Second Revision:

Referees' comments:

Referee #5 (Remarks to the Author)

“Patel and colleagues have worked on all of my comments. However, based on the newly added, very helpful regional association plots depicting the association of SNPs in the region with ccRCC and with papillary RCC, there is an additional observation that merits further clarification. While, as the authors state, there is no association between the ccRCC candidate SNPs with papillary RCC, the plot for papillary RCC shows genome-wide significant SNPs in the CCND1 region. Thus, based on the current plot, one would conclude that the region in fact contains different risk SNPs, some for ccRCC via the mechanism described in this manuscript, and some for papillary RCC more upstream. If so, I believe it would be interesting for readers to know that while the authors convincingly demonstrate the mechanism related to ccRCC, other germline variants in the region may relate to papillary RCC through different mechanisms.”

Reply: We agree that these are important points, thank you for highlighting them. After further investigation, we found that the apparent genome-wide significant SNPs for papillary RCC represent rare variants with limited data, the case and control numbers being far smaller than for the rs7948643-rs7105934 locus, which is the focus of this study. This makes the asymptotic test statistics for these variants unreliable, and we therefore feel it would be more prudent to restrict the regional plots and tables of meta-analysis summary results to include only results for SNPs of minor allele frequencies ≥ 0.01 , which are statistically robust. Given the uncertainty related to the relatively low numbers, it would also be safer to wait for the completion of larger papillary RCC GWAS projects before highlighting possible positive papillary RCC-specific associations in the CCND1 locus, in particular because they are only peripherally related to the main finding of our current study. We have revised **Extended Data Fig. 8** and **Supplementary Tables 7-8** to reflect the new analysis with filtered data.

“Please check with Dr. Chanock first about the papillary RCC data, as I did not find mention of the CCND1 locus for papillary RCC in the original publication, as one would expect for a genome-wide significant locus.”

Reply: The GWAS meta-analysis (Scelo et al. Nat Commun. 2017) focused on overall RCC risk and the RCC subtype-specific associations were not examined further nor reported at that point. The majority of samples in the study were ccRCCs, which allowed ccRCC-specific loci, such as rs7105934, to reach significance. However, the first GWAS meta-analysis was not expected to identify papillary RCC-specific loci.

“Just to make sure the correct file was transferred for papillary RCC, and/or whether the transferred file needs further filtering.”

Reply: Thank you for highlighting this. As discussed above, we have now filtered the data to only include SNPs with minor allele frequencies ≥ 0.01 . The revised plots and source data are provided in **Extended Data Fig. 8** and **Supplementary Tables 7-8**.

“Also, please also label the ccRCC-related SNPs in the panel showing results for papillary RCC.”

Reply: Thank you for the suggestion, we have labelled the ccRCC-associated SNPs in the revised **Extended Data Fig. 8**.